# Regret Bounds for Noise-Free Cascaded Kernelized Bandits

**Zihan Li**  *lizihan@u.nus.edu*
*National University of Singapore*

**Jonathan Scarlett**  *scarlett@comp.nus.edu.sg*
*National University of Singapore*

**Reviewed on OpenReview:** *https://openreview.net/forum?id=oCfamUtecN*

## Abstract

We consider optimizing a function network in the noise-free grey-box setting with RKHS function classes, where the exact intermediate results are observable. We assume that the structure of the network is known (but not the underlying functions comprising it), and we study three types of structures: (1) chain: a cascade of scalar-valued functions, (2) multi-output chain: a cascade of vector-valued functions, and (3) feed-forward network: a fully connected feed-forward network of scalar-valued functions. We propose a sequential upper confidence bound based algorithm GPN-UCB along with a general theoretical upper bound on the cumulative regret. In addition, we propose a non-adaptive sampling based method along with its theoretical upper bound on the simple regret for the Matérn kernel. We also provide algorithm-independent lower bounds on the simple regret and cumulative regret. Our regret bounds for GPN-UCB have the same dependence on the time horizon as the best known in the vanilla black-box setting, as well as near-optimal dependencies on other parameters (e.g., RKHS norm and network length).

## 1 Introduction

Black-box optimization of an expensive-to-evaluate function based on point queries is a ubiquitous problem in machine learning. Bayesian optimization (or Gaussian process optimization) refers to a class of methods using Gaussian processes (GPs), whose main idea is to place a prior over the unknown function and update the posterior according to point query results. Bayesian optimization has a wide range of applications including parameter tuning (Snoek et al., 2012), experimental design (Griffiths and Hernández-Lobato, 2020), and robotics (Lizotte et al., 2007). While function evaluations are noisy in most applications, there are also scenarios where noise-free modeling can be suitable, such as simulation (Nguyen et al., 2016), goal-driven dynamics learning (Bansal et al., 2017), and density map alignment (Singer and Yang, 2023).

In the literature on Bayesian optimization, the problem of optimizing a real-valued black-box function is usually studied under two settings: (1) Bayesian setting: the target function is sampled from a known GP prior, and (2) non-Bayesian setting: the target function has a low norm in reproducing kernel Hilbert space (RKHS).

In this work, we consider a setting falling "in between" the white-box setting (where the full definition of the target function is known) and the black-box setting, namely, a grey-box setting in which the algorithms can leverage partial internal information of the target function beyond merely the final outputs or even slightly modify the target function (Astudillo and Frazier, 2021b). Existing grey-box optimization methods exploit internal information such as observations of intermediate outputs for composite functions (Astudillo and Frazier, 2019) and lower fidelity but faster approximation of the final output for modifiable functions (Huang et al., 2006). Numerical experiments show that the grey-box methods significantly outperform standard black-box methods (Astudillo and Frazier, 2021b).

Though any real-valued network can be treated as a single black-box function and solved with classical Bayesian optimization methods, we explore the benefits offered by utilizing the network structure information and the exact intermediate results under the noise-free grey-box setting. Several practical applications of the cascaded setting (e.g., alloy heat treatment and simulation) are highlighted in Appendix A.

## 1.1 Related Work

Numerous works have proposed Bayesian optimization algorithms for optimizing a single real-valued black-box function under the RKHS setting. For the noisy setting, (Srinivas et al., 2010; Chowdhury and Gopalan, 2017; Gupta et al., 2022) provided a typical cumulative regret $\widetilde{O}(\sqrt{T}\gamma_T)$, where $T$ is the time horizon, $\gamma_T$ is the maximum information gain associated to the underlying kernel, and the $\widetilde{O}(\cdot)$ notation hides the poly-logarithmic factors. Recently, (Camilleri et al., 2021; Salgia et al., 2021; Li and Scarlett, 2022) achieved cumulative regret $\widetilde{O}(\sqrt{T\gamma_T})$, which nearly matches algorithm-independent lower bounds for the squared exponential and Matérn kernels (Scarlett et al., 2017; Cai and Scarlett, 2021). For the noise-free setting, (Bull, 2011) achieved a nearly optimal simple regret $O(T^{-\nu/d})$ for the Matérn kernel with smoothness $\nu$. This result implies a two-batch algorithm that uniformly selects $T^{\frac{d}{\nu+d}}$ points in the first batch and repeatedly picks the returned point in the second batch, has cumulative regret $O(T^{\frac{d}{\nu+d}})$. In addition, (Lyu et al., 2019) provided deterministic cumulative regret $O(\sqrt{T\gamma_T})$, with the rough idea being to substitute zero noise into the analysis of GP-UCB (Srinivas et al., 2010). Recently, (Salgia et al., 2023) proposed a batch algorithm based on random sampling, attaining cumulative regret $\widetilde{O}(T^{1-\nu/d})$ when $\nu < d$ and $O(\mathrm{poly}(\log T))$ when $\nu \geq d$.

Meanwhile, several Bayesian optimization algorithms for optimizing a composition of multiple functions under the noise-free setting have been proposed. (Nguyen et al., 2016) provides a method for cascade Bayesian optimization; (Astudillo and Frazier, 2019) studies optimizing a composition of a black-box function and a known cheap-to-evaluate function; and (Astudillo and Frazier, 2021a) studies optimizing a network of functions sampled from a GP prior under the grey-box setting. Both (Astudillo and Frazier, 2019) and (Astudillo and Frazier, 2021a) prove the asymptotic consistency of their expected improvement sampling based methods.

The most related works to ours are (Kusakawa et al., 2022) and (Sussex et al., 2023). (Kusakawa et al., 2022) introduces two confidence bound based algorithms along with their regret guarantees for both noise-free and noisy settings, as well as an expected improvement based algorithm without theory. (Sussex et al., 2023) considers directed grey-box networks representing a causal structure, and proposes an expected improvement based method with regret guarantee. A detailed comparison regarding problem setup and theoretical performance is provided in Appendix K. In short, we significantly improve certain dependencies in their regret for a UCB-type approach, and we study two new directions – non-adaptive sampling and algorithm-independent lower bounds – that were not considered therein (summarized in Section 1.2).

For function networks with $m$ layers, our cumulative regret bounds are expressed in terms of

$$\Sigma_T = \max_{i \in [m]} \max_{\mathbf{z}_1, \ldots, \mathbf{z}_T \in \mathcal{X}^{(i)}} \sum_{t=1}^{T} \sigma_{t-1}^{(i)}(\mathbf{z}_t),$$

where $\mathcal{X}^{(i)}$ and $\sigma_{t-1}^{(i)}(\cdot)$ denote the domain and posterior standard deviation of the $i$-th layer respectively. This is a term for general layer-composed networks associated to domains $\mathcal{X}^{(1)}, \ldots, \mathcal{X}^{(m)}$ and kernel $k$. When $m = 1$, the term $\max_{\mathbf{x}_1, \ldots, \mathbf{x}_T \in \mathcal{X}} \sum_{t=1}^{T} \sigma_{t-1}(\mathbf{x}_t)$ often appears in the cumulative regret analysis of classic black-box optimization (Srinivas et al., 2010; Lyu et al., 2019; Vakili, 2022). Explicit upper bounds on $\Sigma_T$ will be discussed in Section 3.4.

## 1.2 Contributions

We study the problem of optimizing an $m$-layer function network in the noise-free grey-box setting, where the exact intermediate results are observable. We focus on three types of network structures: (1) chain:

| Lower bound | $\Omega(B(cL)^{m-1}T^{1-\nu/d})$ | |
|---|---|---|
| Upper bound (chains) | $O(2^m BL^{m-1}\Sigma_T)$ | $\stackrel{(c)}{=} O(2^m BL^{m-1}T^{1-\nu/d})$ |
| Upper bound (multi-output chains) | $O(5^m BL^{m-1}\Sigma_T)$ | $\stackrel{(c)}{=} O(5^m BL^{m-1}T^{1-\nu/d_{\max}})$ |
| Upper bound (feed-forward networks) | $O(2^m \sqrt{D_{2,m}}BL^{m-1}\Sigma_T)$ | $\stackrel{(c)}{=} O(2^m \sqrt{D_{2,m}}BL^{m-1}T^{1-\nu/d_{\max}})$ |

Table 1: Summary of cumulative regret bounds for the Matérn kernel when $d \geq \nu \geq 1$ and $T = \Omega\big((B(cL)^{m-1})^{d/\nu}\big)$ for some $c = \Theta(1)$. Here $\stackrel{(c)}{=}$ indicates the behavior when a conjecture of (Vakili, 2022) on the black-box setting holds.

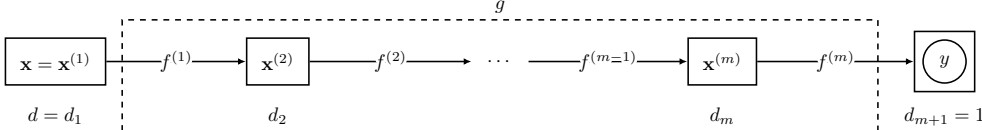

Figure 1: A function network $g$ of $m$ layers with input $\mathbf{x}$ and output $y$.

a cascade of scalar-valued functions, (2) multi-output chain: a cascade of vector-valued functions, and (3) feed-forward network: a fully connected feed-forward network of scalar-valued functions. Then:

- We propose a fully sequential upper confidence bound based algorithm GPN-UCB along with its upper bound on cumulative regret for each network structure. Our regret bound significantly reduces certain dependencies compared to (Kusakawa et al., 2022), in particular showing that their dependence on a "posterior standard deviation Lipschitz constant" can be completely removed.

- We introduce a non-adaptive sampling based method, and provide its theoretical upper bound on the simple regret for the Matérn kernel.

- We provide algorithm-independent lower bounds on the simple and cumulative regret for an arbitrary algorithm optimizing any chain, multi-output chain, or feed-forward network associated to the Matérn kernel. In broad regimes of interest, these provide evidence or even proof that our upper bounds are near-optimal.

- While the goals of this paper are essentially entirely theoretical, we show in Appendix L that (slight variations of) our algorithms can be effective in at least simple experimental scenarios.

Let $d$ denote the dimension of the domain, $d_{\max}$ denote the maximum dimension among all the $m$ layers, and $D_{2,m}$ denote the product of dimensions from the second layer to the last layer. With $B > 0$ restricting the magnitude and smoothness of each layer and $L > 1$ restricting the slope of each layer, a partial summary of the proposed cumulative regret bounds for the Matérn kernel with smoothness $\nu \geq 1$ when $d \geq \nu$ and $T = \Omega\big((B(cL)^{m-1})^{d/\nu}\big)$ for $c = \Theta(1)$ is displayed in Table 1. From this table, we note the following:

- The upper and lower bounds share the same $BL^{m-1}T^{1-\nu/d}$ dependence when $d = d_{\max}$ and simultaneously a conjecture of (Vakili, 2022) holds.

- Even without such a conjecture, the dependence on $\Sigma_T$ is precisely that given in state-of-the-art bounds for the vanilla black-box setting (Vakili, 2022), and rigorous upper bounds on it are known.

See Section 3.4 for further details on the conjecture and rigorous bounds.

To our knowledge, we are the first to attain provably near-optimal scaling (in broad cases of interest), and doing so requires both improving the existing upper bounds and attaining novel lower bounds. A full summary of our theoretical results is provided in Appendix J. Perhaps our most restrictive assumption is noise-free observations, but we believe this is a crucial stepping stone towards the noisy setting (as was the case with regular black-box optimization, e.g., (Bull, 2011)).

## 2 Problem Setup

We consider optimizing a real-valued grey-box function $g$ on $\mathcal{X} = [0,1]^d$ based on noise-free point queries. As shown in Figure 1, the target function $g$ is known to be a network of $m$ unknown layers $f^{(i)}$ with $i \in [m]$. In general, for any input $\mathbf{x} \in \mathcal{X}$, the network $g$ has $\mathbf{x}^{(1)} = \mathbf{x}$ and

$$\mathbf{x}^{(i+1)} = f^{(i)}(\mathbf{x}^{(i)}) \qquad \text{for } i \in [m-1],$$
$$y = g(\mathbf{x}) = f^{(m)}(\mathbf{x}^{(m)}) \quad \in \mathbb{R},$$

where $\mathbf{x}^{(i)}$ has dimension $d_i$ for each $i \in [m]$. The domain of $f^{(i)}$ is $\mathcal{X}^{(i)}$, and the range of $f^{(i)}$ is $\mathcal{X}^{(i+1)}$.[1] For any $\mathbf{z} \in \mathcal{X}^{(i)}$, there exists $\mathbf{x} \in \mathcal{X}$ such that $\mathbf{x}^{(i)} = \mathbf{z}$.

We aim to find $\mathbf{x}^* = \arg\max_{\mathbf{x} \in \mathcal{X}} g(\mathbf{x})$ based on a sequence of point queries up to time horizon $T$. When we query $g$ with input $\mathbf{x}_t$ at time step $t$, the intermediate noise-free results $\mathbf{x}_t^{(2)}, \ldots, \mathbf{x}_t^{(m)}$ and the final noise-free output $y_t$ are accessible. We measure the performance as follows:

- **Simple regret**: With $\mathbf{x}_T^*$ being the additional point returned after $T$ rounds, the simple regret is defined as $r_T^* = g(\mathbf{x}^*) - g(\mathbf{x}_T^*)$;

- **Cumulative regret**: The cumulative regret incurred over $T$ rounds is defined as $R_T = \sum_{i=1}^{T} r_t$ with $r_t = g(\mathbf{x}^*) - g(\mathbf{x}_t)$.

### 2.1 Kernelized Bandits

We assume $g$ is a composition of multiple constituent functions, for which we consider both scalar-valued functions and vector-valued functions based on a given kernel. For a scalar-valued kernel $k$ and a known constant $B > 0$, we consider scalar-valued functions that lie in $\mathcal{H}_k(B)$, the reproducing kernel Hilbert space (RKHS) associated to $k$, with norm at most $B$. In this work, we focus on the Matérn kernel $k_{\text{Matérn}}$ with smoothness $\nu > 0$. Similarly, for an operator-valued kernel $\Gamma$ and a known constant $B > 0$, we consider vector-valued functions in $\mathcal{H}_\Gamma(B)$, the RKHS corresponding to $\Gamma$ with norm at most $B$. More details on RKHS and $k_{\text{Matérn}}$ are given in Appendix B.

### 2.2 Surrogate GP Model

As is common in kernelized bandit problems, our algorithms employ a surrogate Bayesian GP model for $f \in \mathcal{H}_k(B)$. For prior with zero mean and kernel $k$, given a sequence of points $(\mathbf{x}_1, \ldots, \mathbf{x}_t)$ and their noise-free observations $\mathbf{y}_t = (y_1, \ldots, y_t)$ up to time $t$, the posterior distribution of the function is a GP with mean and variance given by (Rasmussen and Williams, 2006)

$$\mu_t(\mathbf{x}) = \mathbf{k}_t(\mathbf{x})^T \mathbf{K}_t^{-1} \mathbf{y}_t \tag{1}$$
$$\sigma_t(\mathbf{x})^2 = k(\mathbf{x}, \mathbf{x}) - \mathbf{k}_t(\mathbf{x})^T \mathbf{K}_t^{-1} \mathbf{k}_t(\mathbf{x}), \tag{2}$$

where $\mathbf{k}_t(\mathbf{x}) = [k(\mathbf{x}, \mathbf{x}_i)]_{i=1}^t \in \mathbb{R}^{t \times 1}$ and $\mathbf{K}_t = [k(\mathbf{x}_i, \mathbf{x}_j)]_{i,j=1}^t \in \mathbb{R}^{t \times t}$. The following lemma shows that the posterior confidence region defined with parameter $B$ is always deterministically valid.

**Lemma 1.** (Kanagawa et al., 2018, Corollary 3.11) *For $f \in \mathcal{H}_k(B)$, let $\mu_t(\mathbf{x})$ and $\sigma_t(\mathbf{x})^2$ denote the posterior mean and variance based on $t$ points $(\mathbf{x}_1, \ldots, \mathbf{x}_t)$ and their noise-free observations $(y_1, \ldots, y_t)$ using (1) and (2). Then, it holds for all $\mathbf{x} \in \mathcal{X}$ that*

$$|f(\mathbf{x}) - \mu_t(\mathbf{x})| \leq B \sigma_t(\mathbf{x}).$$

We also impose a surrogate GP model for functions in $\mathcal{H}_\Gamma(B)$. The posterior mean and covariance matrix based on $(\mathbf{x}_1, \ldots, \mathbf{x}_t)$ and the noise-free observations $Y_t = (\mathbf{y}_1, \ldots, \mathbf{y}_t)$ are (Chowdhury and Gopalan, 2021)

$$\mu_t(\mathbf{x}) = G_t(\mathbf{x})^T G_t^{-1} Y_t, \tag{3}$$
$$\Gamma_t(\mathbf{x}, \mathbf{x}) = \Gamma(\mathbf{x}, \mathbf{x}) - G_t(\mathbf{x})^T G_t^{-1} G_t(\mathbf{x}), \tag{4}$$

---

[1] Note the equivalent notation $\mathcal{X}^{(1)} = \mathcal{X}$ and $d_1 = d$.

---

**Algorithm 1** GPN-UCB (Gaussian Process Network - Upper Confidence Bound)

---

1: **for** $t \leftarrow 1, 2, \dots, T$ **do**
2:     Select $\mathbf{x}_t \leftarrow \arg\max_{\mathbf{x} \in \mathcal{X}} \mathrm{UCB}_{t-1}(\mathbf{x})$
3:     Obtain observations $\mathbf{x}_t^{(2)}, \dots, \mathbf{x}_t^{(m)}$, and $y_t$.
4:     Compute $\mathrm{UCB}_t$ using (11), (15), or (19) based on $\{\mathbf{x}_s^{(1)}, \dots, \mathbf{x}_s^{(m)}, y_s\}_{s=1}^t$.

---

where $G_t(\mathbf{x}) = [\Gamma(\mathbf{x}, \mathbf{x}_i)]_{i=1}^t \in \mathbb{R}^{nt \times n}$, $G_t = [\Gamma(\mathbf{x}_i, \mathbf{x}_j)]_{i,j=1}^t \in \mathbb{R}^{nt \times nt}$, and $Y_t = [\mathbf{y}_i]_{i=1}^t \in \mathbb{R}^{nt \times 1}$. With $\|\cdot\|_2$ denoting the spectral norm, the following lemma provides a deterministic confidence region.

**Lemma 2.** *For $f \in \mathcal{H}_\Gamma(B)$, let $\mu_t(\mathbf{x})$ and $\Gamma_t(\mathbf{x}, \mathbf{x})$ denote the posterior mean and variance based on $t$ points $(\mathbf{x}_1, \dots, \mathbf{x}_t)$ and their noise-free observations $(\mathbf{y}_1, \dots, \mathbf{y}_t)$ using (3) and (4). Then, it holds for all $\mathbf{x} \in \mathcal{X}$ that*

$$\|f(\mathbf{x}) - \mu_t(\mathbf{x})\|_2 \leq B\|\Gamma_t(\mathbf{x}, \mathbf{x})\|_2^{1/2}.$$

The proof is given in Appendix D.

### 2.3 Lipschitz Continuity

We also assume that each constituent function in the network $g$ is Lipschitz continuous. For a constant $L > 1$, we denote by $\mathcal{F}(L)$ the set of functions such that

$$\mathcal{F}(L) = \{f : \|f(\mathbf{x}) - f(\mathbf{x}')\|_2 \leq L\|\mathbf{x} - \mathbf{x}'\|_2, \forall \mathbf{x}, \mathbf{x}'\},$$

where $L$ is called the Lipschitz constant. This is a mild assumption, as (Lee et al., 2022) has shown that Lipschitz continuity is a guarantee for functions in $\mathcal{H}_k(B)$ for the commonly-used squared exponential kernel and Matérn kernel with smoothness $\nu > 1$.

### 2.4 Network Structures

In this work, we consider three types of network structure; example figures are included in Appendix C:

- **Chain**: For a scalar-valued kernel $k$, a chain is a cascade of scalar-valued functions. Specifically, $d_1 \geq 1$, $d_2 = d_3 = \cdots = d_m = 1$, and $f^{(i)} \in \mathcal{H}_k(B) \cap \mathcal{F}(L)$ for each $i \in [m]$.

- **Multi-output chain**: For an operator-valued kernel $\Gamma$, a multi-output chain is a cascade of vector-valued functions. Specifically, $d_i \geq 1$ and $f^{(i)} \in \mathcal{H}_\Gamma(B) \cap \mathcal{F}(L)$ for each $i \in [m]$.

- **Feed-forward network**: For a scalar-valued kernel $k$, a feed-forward network is a fully-connected feed-forward network of scalar-valued functions: $d_i \geq 1$ and $f^{(i)}(\mathbf{z}) = [f^{(i,j)}(\mathbf{z})]_{j=1}^{d_{i+1}}$ with $f^{(i,j)} \in \mathcal{H}_k(B) \cap \mathcal{F}(L)$ for each $i \in [m], j \in [d_{i+1}]$.

In each case, the network $g$ is scalar-valued with the dimension of the final output $y$ being $d_{m+1} = 1$.

## 3 GPN-UCB Algorithm and Regret Bounds

In this section, we propose a fully sequential algorithm GPN-UCB (see Algorithm 1) for chains, multi-output chains, and feed-forward networks. The algorithm works with structure-specific upper confidence bounds. Similar to GP-UCB for scalar-valued functions (Srinivas et al., 2010), the proposed algorithm repeatedly queries the point with the highest posterior upper confidence bound, while the posterior upper confidence bound $\mathrm{UCB}_{t-1}$ used here is computed based on not only the historical final outputs $\{y_s\}_{s=1}^{t-1}$ but also the intermediate results $\{\mathbf{x}_s^{(2)}, \dots, \mathbf{x}_s^{(m)}\}_{s=1}^{t-1}$.

### 3.1 GPN-UCB for Chains

A chain is a cascade of scalar-valued functions. For each $i \in [m]$, we denote by $\mu_t^{(i)}$ and $\sigma_t^{(i)}$ the posterior mean and standard deviation of $f^{(i)}$ computed using (1) and (2) based on $\{\mathbf{x}_s^{(i)}, \mathbf{x}_s^{(i+1)}\}_{s=1}^t$.[2] Then, based on Lemma 1, the upper confidence bound and lower confidence bound of $f^{(i)}(\mathbf{z})$ based on $t$ exact observations are defined as follows:

$$\text{UCB}_t^{(i)}(\mathbf{z}) = \mu_t^{(i)}(\mathbf{z}) + B\sigma_t^{(i)}(\mathbf{z}), \tag{5}$$

$$\text{LCB}_t^{(i)}(\mathbf{z}) = \mu_t^{(i)}(\mathbf{z}) - B\sigma_t^{(i)}(\mathbf{z}). \tag{6}$$

Since $f^{(i)} \in \mathcal{F}(L)$, we have for any $\mathbf{z}, \mathbf{z}'$ that

$$\text{UCB}_t^{(i)}(\mathbf{z}') + L\|\mathbf{z} - \mathbf{z}'\|_2 \geq f^{(i)}(\mathbf{z}') + L\|\mathbf{z} - \mathbf{z}'\|_2 \geq f^{(i)}(\mathbf{z}), \tag{7}$$

$$\text{LCB}_t^{(i)}(\mathbf{z}') - L\|\mathbf{z} - \mathbf{z}'\|_2 \leq f^{(i)}(\mathbf{z}') - L\|\mathbf{z} - \mathbf{z}'\|_2 \leq f^{(i)}(\mathbf{z}). \tag{8}$$

It follows that

$$\overline{\text{UCB}}_t^{(i)}(\mathbf{z}) := \min_{\mathbf{z}'} \left( \text{UCB}_t^{(i)}(\mathbf{z}') + L\|\mathbf{z} - \mathbf{z}'\|_2 \right), \tag{9}$$

$$\overline{\text{LCB}}_t^{(i)}(\mathbf{z}) := \max_{\mathbf{z}'} \left( \text{LCB}_t^{(i)}(\mathbf{z}') - L\|\mathbf{z} - \mathbf{z}'\|_2 \right) \tag{10}$$

are also valid confidence bounds for $f^{(i)}(\mathbf{z})$. $\overline{\text{UCB}}_t^{(i)}(\mathbf{z})$ is the lower envelope of a collection of upper bounds for $f^{(i)}(\mathbf{z})$, which can be obtained by considering multiple values of $\mathbf{z}'$ in (7). Then, since $g$ is a cascade of $f^{(i)}$'s, for any input $\mathbf{x}$, we can recursively construct a confidence region of $\mathbf{x}^{(i+1)}$ based on the confidence region of $\mathbf{x}^{(i)}$, and the following UCB for $g(\mathbf{x})$ is valid:

$$\text{UCB}_t(\mathbf{x}) = \max_{\mathbf{z} \in \Delta_t^{(m)}(\mathbf{x})} \overline{\text{UCB}}_t^{(m)}(\mathbf{z}), \tag{11}$$

where $\Delta_t^{(i)}(\mathbf{x})$ denotes the confidence region of $\mathbf{x}^{(i)}$:

$$\Delta_t^{(1)}(\mathbf{x}) = \{\mathbf{x}\}$$

$$\Delta_t^{(i+1)}(\mathbf{x}) = \left[ \min_{\mathbf{z} \in \Delta_t^{(i)}(\mathbf{x})} \overline{\text{LCB}}_t^{(i)}(\mathbf{z}), \max_{\mathbf{z} \in \Delta_t^{(i)}(\mathbf{x})} \overline{\text{UCB}}_t^{(i)}(\mathbf{z}) \right]$$

for $i \in [m-1]$. The theoretical performance of Algorithm 1 for chains using the upper confidence bound in (11) is provided in the following theorem.

**Theorem 1** (GPN-UCB for chains). *Under the setup of Section 2, given $B > 0$ and $L > 1$, a scalar-valued kernel $k$, and a chain $g = f^{(m)} \circ f^{(m-1)} \circ \cdots \circ f^{(1)}$ with $f^{(i)} \in \mathcal{H}_k(B) \cap \mathcal{F}(L)$ for each $i \in [m]$, Algorithm 1 achieves*

$$R_T \leq 2^{m+1} B L^{m-1} \Sigma_T,$$

*where $\Sigma_T = \max_{i \in [m]} \max_{\mathbf{z}_1, \ldots, \mathbf{z}_T \in \mathcal{X}^{(i)}} \sum_{t=1}^T \sigma_{t-1}^{(i)}(\mathbf{z}_t)$.* [3]

The proof is given in Appendix F.1, and upper bounds on $\Sigma_T$ will be discussed in Section 3.4. Regardless of such upper bounds, we note that $B\Sigma_T$ serves as a noise-free regret bound for standard GP optimization (Vakili, 2022), and thus, the key distinction here is the multiplication by $L^{m-1}$. See Section 5 for a study of the extent to which this dependence is unavoidable.

---

[2]Note the equivalent notation $\mathbf{x}^{(1)} = \mathbf{x}$ and $\mathbf{x}^{(m+1)} = y$.

[3]In this definition and analogous definitions below, $\sigma_{t-1}^{(i)}$ is defined according to the hypothetical sampled points $\mathbf{x}_\tau^{(i)} = \mathbf{z}_\tau$ for $\tau = 1, \ldots, t-1$.

We note that GPN-UCB may be difficult to implement *exactly* in practice; in particular: (i) Since $\mathcal{X}^{(2)}, \ldots, \mathcal{X}^{(m)}$ are not known, (9) and (10) are computed based on all $\mathbf{z}' \in \mathbb{R}^{d_i}$; (ii) Recursively computing (11) is also resource consuming. However, these problems can be alleviated by (i) only considering $\mathbf{z}'$ sufficiently close to $\mathbf{z}$ (since distant ones should have no impact) and (ii) replacing each confidence region by its intersection with a fixed *discrete* domain (e.g., a finite grid). In Appendix L, we show that such a practical variant can be effective, at least in simple experimental scenarios.

## 3.2 GPN-UCB for Multi-Output Chains

A multi-output chain is a cascade of vector-valued functions. For any input $\mathbf{z}$ of the multi-output function $f^{(i)}$, we define the confidence region of $f^{(i)}(\mathbf{z})$ as

$$\bar{\mathcal{C}}_t^{(i)}(\mathbf{z}) = \bigcap_{\mathbf{z}'} \mathcal{C}_t^{(i)}(\mathbf{z}, \mathbf{z}'), \tag{12}$$

where

$$\mathcal{C}_t^{(i)}(\mathbf{z}') = \{\mu_t^{(i)}(\mathbf{z}') + \mathbf{u} : \mathbf{u} \in \mathbb{R}^{d_{i+1}}, \|\mathbf{u}\|_2 \le B\|\Gamma_t^{(i)}(\mathbf{z}, \mathbf{z})\|_2^{1/2}\}, \tag{13}$$

$$\mathcal{C}_t^{(i)}(\mathbf{z}, \mathbf{z}') = \{\mathbf{v} + \mathbf{w} : \mathbf{v} \in \mathcal{C}_t^{(i)}(\mathbf{z}'), \|\mathbf{w}\|_2 \le L\|\mathbf{z} - \mathbf{z}'\|_2\}. \tag{14}$$

Lemma 2 shows that $\mathcal{C}_t^{(i)}(\mathbf{z}')$ is a valid deterministic confidence region for $f^{(i)}(\mathbf{z}')$. Assuming $f^{(i)} \in \mathcal{F}(L)$, $\{f^{(i)}(\mathbf{z}') + \mathbf{w} : \mathbf{w} \in \mathbb{R}^{d_{i+1}}, \|\mathbf{w}\|_2 \le L\|\mathbf{z} - \mathbf{z}'\|_2\}$ containing all the points satisfying the Lipschitz property is a valid confidence region for $f^{(i)}(\mathbf{z})$, and therefore its superset $\mathcal{C}_t^{(i)}(\mathbf{z}, \mathbf{z}')$ is also a valid confidence region for $f^{(i)}(\mathbf{z})$. Since $f^{(i)}(\mathbf{z})$ must belong to the intersection of all its confidence regions, $\bar{\mathcal{C}}_t^{(i)}(\mathbf{z})$ is again a valid deterministic confidence region for $f^{(i)}(\mathbf{z})$. Hence, noting that $\bar{\mathcal{C}}_t^{(m)}$ is a subset of $\mathbb{R}$ (unlike the vector-valued layers), the upper confidence bound for $g(\mathbf{x})$ for any input $\mathbf{x}$ based on $t$ observations is

$$\mathrm{UCB}_t(\mathbf{x}) = \max_{\mathbf{z} \in \Delta_t^{(m)}(\mathbf{x})} \bar{\mathcal{C}}_t^{(m)}(\mathbf{z}), \tag{15}$$

where $\Delta_t^{(i)}(\mathbf{x})$ denotes the confidence region of $\mathbf{x}^{(i)}$ based on $t$ observations such that

$$\begin{aligned} \Delta_t^{(1)}(\mathbf{x}) &= \{\mathbf{x}\} \\ \Delta_t^{(i+1)}(\mathbf{x}) &= \bigcup_{\mathbf{z} \in \Delta_t^{(i)}(\mathbf{x})} \bar{\mathcal{C}}_t^{(i)}(\mathbf{z}) \qquad \text{for } i \in [m-1]. \end{aligned} \tag{16}$$

The cumulative regret achieved by Algorithm 1 for multi-output chains using the upper confidence bound in (15) is provided in the following theorem.

**Theorem 2** (GPN-UCB for multi-output chains). *Under the setup of Section 2, given $B > 0$ and $L > 1$, an operator-valued kernel $\Gamma$, and a multi-output chain $g = f^{(m)} \circ f^{(m-1)} \circ \cdots \circ f^{(1)}$ with $f^{(i)} \in \mathcal{H}_\Gamma(B) \cap \mathcal{F}(L)$ for each $i \in [m]$, Algorithm 1 achieves*

$$R_T \le 5^m B L^{m-1} \Sigma_T^\Gamma,$$

*where $\Sigma_T^\Gamma = \max_{i \in [m]} \max_{\mathbf{z}_1, \ldots, \mathbf{z}_T \in \mathcal{X}^{(i)}} \sum_{t=1}^T \|\Gamma_{t-1}^{(i)}(\mathbf{z}_t, \mathbf{z}_t)\|_2^{1/2}.$*

The proof is given in Appendix F.2.

**Remark 1.** Fix $\mathcal{X}^{(1)}, \mathcal{X}^{(2)}, \ldots, \mathcal{X}^{(m)}$ with dimension $d_1, d_2, \ldots, d_m \ge 1$ respectively. For $\Gamma(\cdot, \cdot) = k(\cdot, \cdot)\mathbf{I}$ with $k$ being a scalar-valued kernel and $\mathbf{I}$ being the identity matrix of size $d_{i+1}$, it follows from Lemma 4 (see Appendix E) that $\Sigma_T^\Gamma = \Sigma_T$.

The upper bound on $\Sigma_T^\Gamma$ for general operator-valued kernels will be discussed in Section 3.4.

### 3.3 GPN-UCB for Feed-Forward Networks

In the feed-forward network structure, $f^{(i)}(\mathbf{z}) = [f^{(i,j)}(\mathbf{z})]_{j=1}^{d_{i+1}}$ and each $f^{(i,j)} \in \mathcal{H}_k(B) \cap \mathcal{F}(L)$ is a scalar-valued function. Similar to (9) and (10), with $\mu_t^{(i,j)}(\mathbf{z})$ and $\sigma_t^{(i,j)}(\mathbf{z})^2$ denoting the posterior mean and variance of $f^{(i,j)}(\mathbf{z})$ using (1) and (2), the following confidence bounds on $f^{(i,j)}(\mathbf{z})$ based on $\{(\mathbf{x}_s^{(i)}, \mathbf{x}_s^{(i+1,j)})\}_{s=1}^t$ are valid:

$$\overline{\mathrm{UCB}}_t^{(i,j)}(\mathbf{z}) = \min_{\mathbf{z}'}(\mathrm{UCB}_t^{(i,j)}(\mathbf{z}') + L\|\mathbf{z} - \mathbf{z}'\|_2),$$
$$\overline{\mathrm{LCB}}_t^{(i,j)}(\mathbf{z}) = \max_{\mathbf{z}'}(\mathrm{LCB}_t^{(i,j)}(\mathbf{z}') - L\|\mathbf{z} - \mathbf{z}'\|_2), \tag{17}$$

where

$$\mathrm{UCB}_t^{(i,j)}(\mathbf{z}) = \mu_t^{(i,j)}(\mathbf{z}) + B\sigma_t^{(i,j)}(\mathbf{z}),$$
$$\mathrm{LCB}_t^{(i,j)}(\mathbf{z}) = \mu_t^{(i,j)}(\mathbf{z}) - B\sigma_t^{(i,j)}(\mathbf{z}). \tag{18}$$

Then, the upper confidence bound of $g(\mathbf{x})$ based on $t$ observations is

$$\mathrm{UCB}_t(\mathbf{x}) = \max_{\mathbf{z} \in \Delta_t^{(m)}(\mathbf{x})} \overline{\mathrm{UCB}}_t^{(m,1)}(\mathbf{z}), \tag{19}$$

where

$$\Delta_t^{(1)}(\mathbf{x}) = \{\mathbf{x}\},$$
$$\Delta_t^{(i+1,j)}(\mathbf{x}) = \left[\min_{\mathbf{z} \in \Delta_t^{(i)}(\mathbf{x})} \overline{\mathrm{LCB}}_t^{(i,j)}(\mathbf{z}), \max_{\mathbf{z} \in \Delta_t^{(i)}(\mathbf{x})} \overline{\mathrm{UCB}}_t^{(i,j)}(\mathbf{z})\right] \qquad \text{for } i \in [m-1], j \in [d_{i+1}],$$
$$\Delta_t^{(i)}(\mathbf{x}) = \Delta_t^{(i,1)}(\mathbf{x}) \times \cdots \times \Delta_t^{(i,d_i)}(\mathbf{x}) \qquad \text{for } i \in [m]. \tag{20}$$

The following theorem provides the theoretical performance of Algorithm 1 for feed-forward networks using the upper confidence bound in (19).

**Theorem 3** (GPN-UCB for feed-forward networks). *Under the setup of Section 2, given $B > 0$ and $L > 1$, a scalar-valued kernel $k$, and a feed-forward network $g = f^{(m)} \circ f^{(m-1)} \circ \cdots \circ f^{(1)}$ with $f^{(i)}(\mathbf{z}) = [f^{(i,j)}(\mathbf{z})]_{j=1}^{d_{i+1}}$ and $f^{(i,j)} \in \mathcal{H}_k(B) \cap \mathcal{F}(L)$ for each $i \in [m], j \in [d_{i+1}]$, Algorithm 1 achieves*

$$R_T \le 2^{m+1}\sqrt{D_{2,m}} BL^{m-1}\Sigma_T,$$

*where $D_{2,m} = \prod_{i=2}^m d_i$ and $\Sigma_T = \max_{i \in [m]} \max_{\mathbf{z}_1, \ldots, \mathbf{z}_T \in \mathcal{X}^{(i)}} \sum_{t=1}^T \sigma_{t-1}^{(i,1)}(\mathbf{z}_t)$.*

The proof is given in Appendix F.3.

Since $\sigma^{(i,1)}$ in the feed-forward network setting is computed using (2), which is exactly the same as how $\sigma^{(i)}$ is computed in the chain setting, despite the slightly different superscripts, the $\Sigma_T$ term in Theorem 3 represents the same quantity as in Theorem 1 (depending on $\mathcal{X}^{(1)}, \mathcal{X}^{(2)}, \ldots, \mathcal{X}^{(m)}$). In particular, when $d_2 = \cdots = d_m = 1$, Theorem 3 recovers Theorem 1.

### 3.4 Upper Bounds on $\Sigma_T$ and $\Sigma_T^\Gamma$

A simple way to establish an upper bound on the $\Sigma_T$ term in Theorem 1, Remark 1, and Theorem 3 is to essentially set the noise term to be zero in a known result for the noisy setting. For the scalar-valued function (GP bandit) optimization problem under the noisy setting, most existing upper bounds on cumulative regret are expressed in terms of the maximum information gain corresponding to the kernel defined as $\gamma_t = \max_{\mathbf{x}_1, \ldots, \mathbf{x}_t} \frac{1}{2}\log\det(\mathbf{I}_t + \lambda^{-1}\mathbf{K}_t)$ for a free parameter $\lambda > 0$ (Srinivas et al., 2010), and (Srinivas et al., 2010) has shown that the sum of posterior variances in the noisy setting satisfies $\sum_{t=1}^T \sigma_{t-1}'(\mathbf{x}_t)^2 = O(\gamma_T)$,

---

**Algorithm 2** Non-Adaptive Sampling Based Method

---

1: Choosing $\{\mathbf{x}_s\}_{s=1}^T$ such that $\delta_T = O(T^{-\frac{1}{d}})$.
2: Obtain observations $\{\mathbf{x}_s^{(2)}, \ldots, \mathbf{x}_s^{(m)}, y_s\}_{s=1}^T$.
3: Compute $\mu_T^g$ based on $\{\mathbf{x}_s^{(2)}, \ldots, \mathbf{x}_s^{(m)}, y_s\}_{s=1}^T$.
**Output:** $\mathbf{x}_T^* = \arg\max_{\mathbf{x} \in \mathcal{X}} \mu_T^g(\mathbf{x})$.

---

where $\sigma_t'(\mathbf{x})^2 = k(\mathbf{x}, \mathbf{x}) - \mathbf{k}_t(\mathbf{x})^T(\mathbf{K}_t + \lambda\mathbf{I}_t)^{-1}\mathbf{k}_t(\mathbf{x})$. Using $\sigma_t(\mathbf{x}) \leq \sigma_t'(\mathbf{x})$ and the Cauchy-Schwartz inequality, we have $\Sigma_T = O(\sqrt{T\gamma_T})$. An existing upper bound on $\gamma_T$ for the Matérn kernel with dimension $d$ and smoothness $\nu$ on a fixed compact domain is $\gamma_T^{\text{Matérn}} = \widetilde{O}(T^{\frac{d}{2\nu+d}})$ (Vakili et al., 2021). In our setting, a simple sufficient condition for this bound to apply is that $d_{\max} = \max_{i \in [m]} d_i$, $L$, and $m$ are constant, since then the Lipschitz assumption implies that each domain $\mathcal{X}^{(1)}, \mathcal{X}^{(2)}, \ldots, \mathcal{X}^{(m)}$ is also compact/bounded. More generally, we believe that uniformly bounded domains is a mild assumption, and when it holds, the bound $\Sigma_T = O(\sqrt{T\gamma_T})$ simplifies to $\Sigma_T = O(T^{\frac{\nu+d_{\max}}{2\nu+d_{\max}}})$.

In addition, (Vakili, 2022) provides the following conjecture on the upper bound on $\Sigma_T$ for the Matérn kernel[4]

$$
\Sigma_T^{\text{Matérn}} = \begin{cases} O(T^{1-\nu/d_{\max}}) & \text{when } d_{\max} > \nu, \\ O(\log T) & \text{when } d_{\max} = \nu, \\ O(1) & \text{when } d_{\max} < \nu. \end{cases}
$$

We will discuss in Section 6 how if this conjecture is true, we can deduce the near-optimality of GPN-UCB for the Matérn kernel. We note that (Vakili, 2022) primarily conjectured on vanilla noise-free cumulative regret by conjecturing an upper bound on $\Sigma_T$. Recently, (Salgia et al., 2023) used a random sampling algorithm with elimination to attain the conjectured cumulative regret, while leaving open the conjecture on $\Sigma_T$ and whether GP-UCB attains the same regret (though arguably further increasing its plausibility).

For an arbitrary operator-valued kernel $\Gamma : \mathcal{X} \times \mathcal{X} \to \mathbb{R}^{n \times n}$ and a free parameter $\lambda$, the maximum information gain is defined as $\gamma_t^\Gamma = \max_{\mathbf{x}_1, \ldots, \mathbf{x}_t} \frac{1}{2} \log\det(\mathbf{I}_{nt} + \lambda^{-1}G_t)$ (Chowdhury and Gopalan, 2021). (Chowdhury and Gopalan, 2021) has shown that $\sum_{t=1}^T \|\Gamma_{t-1}(\mathbf{x}_t, \mathbf{x}_t)\|_2 = O(\gamma_T^\Gamma)$, and therefore $\Sigma_T^\Gamma = O(\sqrt{T\gamma_T^\Gamma})$ by similar reasoning to above.

## 4 Non-Adaptive Sampling Based Method

In this section, we propose a simple non-adaptive sampling based method (see Algorithm 2) for each structure, and provide the corresponding theoretical simple regret for the Matérn kernel. For a set of $T$ sampled points $\{\mathbf{x}_s\}_{s=1}^T$, its fill distance is defined as the largest distance from a point in the domain to the closest sampled point $\delta_T = \max_{\mathbf{x} \in \mathcal{X}} \min_{s \in [T]} \|\mathbf{x} - \mathbf{x}_s\|_2$ (Wendland, 2004). Algorithm 2 samples $T$ points with $\delta_T = O(T^{-1/d})$. For $\mathcal{X} = [0,1]^d$, a simple way to construct such a sample is to use a uniform $d$-dimensional grid with step size $T^{-1/d}$. The algorithm observes the selected points in parallel, computes a structure-specific "composite mean" $\mu_T^g$ (to be defined shortly) for the overall network $g$, and returns the point that maximizes $\mu_T^g$.

The composite posterior mean of $g(\mathbf{x})$ with chain structure is defined as

$$
\mu_T^g(\mathbf{x}) = (\mu_T^{(m)} \circ \mu_T^{(m-1)} \circ \cdots \circ \mu_T^{(1)})(\mathbf{x}), \tag{21}
$$

where $\mu_T^{(i)}$ denotes the posterior mean of $f^{(i)}$ computed using (1) based on $\{(\mathbf{x}_s^{(i)}, \mathbf{x}_s^{(i+1)})\}_{s=1}^T$ for each $i \in [m]$. Then, the following theorem provides the theoretical upper bound on the simple regret of Algorithm 2

---

[4]In more detail, (Vakili, 2022) shows that an analysis of GP-UCB gives rise to the quantity $\Theta_T^* = \max_{x_1, \ldots, x_T} \sum_{t=1}^T \sigma_{t-1}(x_t)$, where the maximum is over an arbitrary sequence of points (not necessarily those of GP-UCB). For $(x_1^*, \ldots, x_T^*)$ where the maximum is achieved, (Vakili, 2022) conjectures that $(x_1^*, \ldots, x_T^*)$ are roughly uniformly distributed across the domain. The desired upper bound on $\Theta_T^*$ (and, in turn, our $\Sigma_T$) is derived by assuming that this conjecture holds.

using (21). Note that the notation $\widetilde{O}(\cdot)$ hides poly-logarithmic factors *with respect to the argument*, e.g., $\widetilde{O}(\sqrt{T}) = O(\sqrt{T} \cdot (\log T)^{O(1)})$ and $\widetilde{O}(2^n) = O(2^n \cdot n^{O(1)})$.

**Theorem 4** (Non-adaptive sampling method for chains). *Under the setup of Section 2, given $B = \Theta(L)$, $k = k_{Matérn}$ with smoothness $\nu$, and a chain $g = f^{(m)} \circ f^{(m-1)} \circ \cdots \circ f^{(1)}$ with $f^{(i)} \in \mathcal{H}_k(B) \cap \mathcal{F}(L)$ for each $i \in [m]$, we have*

- *When $\nu \leq 1$, Algorithm 2 achieves*

$$r_T^* = \widetilde{O}(BL^{m-1}T^{-\nu^m/d});$$

- *When $\nu > 1$, Algorithm 2 achieves*

$$r_T^* = \widetilde{O}\big( \max \big\{ BL^{(m-1)\nu}T^{-\nu/d}, B^{1+\nu+\nu^2+\cdots+\nu^{m-2}}L^{\nu^{m-1}}T^{-\nu^2/d} \big\}\big).$$

The proof is given in Appendix G.1, and the optimality will be discussed in Section 6. When $\nu > 1$, the simple regret upper bound takes the maximum of two terms. The first term has a smaller constant factor, while the second term has a smaller $T$-dependent factor. By taking the highest-order constant factor and the highest-order $T$-dependent factor, we can deduce the weaker but simpler bound $r_T^* = O(B^{1+\nu+\nu^2+\cdots+\nu^{m-2}}L^{\nu^{m-1}}T^{-\nu/d})$.

We also consider two more restrictive cases, where we remove the assumption of $B = \Theta(L)$, but have additional assumptions on $g$ as follows:

- **Case 1**: We additionally assume that $\mu_T^{(i)} \circ \cdots \circ \mu_T^{(1)}(\mathbf{x}^*) \in \mathcal{X}^{(i+1)}$ and $\mu_T^{(i)} \circ \cdots \circ \mu_T^{(1)}(\mathbf{x}_T^*) \in \mathcal{X}^{(i+1)}$ for all $i \in [m-1]$.

- **Case 2**: We additionally assume that all the domains $\mathcal{X}^{(i)}$ are known. Defining

$$\widetilde{\mu}_T^{(i)}(\mathbf{z}) = \underset{\mathbf{z}' \in \mathcal{X}^{(i+1)}}{\arg\min} |\mu_T^{(i)}(\mathbf{z}) - \mathbf{z}'|,$$

we slightly modify the algorithm to return

$$\mathbf{x}_T^* = \underset{\mathbf{x} \in \mathcal{X}}{\arg\max}(\widetilde{\mu}_T^{(m)} \circ \cdots \circ \widetilde{\mu}_T^{(1)})(\mathbf{x}).$$

**Remark 2.** Under the assumptions of either Case 1 or Case 2, Algorithm 2 achieves for chains that

$$r_T^* = \begin{cases} O(BL^{m-1}T^{-\nu/d}) & \text{when } \nu \leq 1, \\ O(BL^{(m-1)\nu}T^{-\nu/d}) & \text{when } \nu > 1. \end{cases} \tag{22}$$

The proof is given in Appendix G.2.

The composite posterior means and simple regret upper bounds for multi-output chains and feed-forward networks are provided in Appendix G.3 and Appendix G.4 respectively, where the simple regret upper bounds are stated only for the case that the domain of each layer is a hyperrectangle. Removing this restrictive assumption is left for future work.

## 5 Algorithm-Independent Lower Bounds

In this section, we provide algorithm-independent lower bounds on the simple regret and cumulative regret for any algorithm optimizing chains, multi-output chains, or feed-forward networks for the scalar-valued kernel $k_{Matérn}$ or the operator-valued $\Gamma_{Matérn}(\cdot, \cdot) = k_{Matérn}(\cdot, \cdot)\mathbf{I}$ with smoothness $\nu$.

**Theorem 5** (Lower bound on simple regret). *Fix $\epsilon \in (0, \frac{1}{2}]$, sufficiently large $B > 0$, $k = k_{Matérn}$, and $\Gamma = \Gamma_{Matérn}$ with smoothness $\nu \geq 1$. Suppose that there exists an algorithm (possibly randomized) that*

*achieves average simple regret $\mathbb{E}[r_T^*] \leq \epsilon$ after $T$ rounds for any m-layer chain, multi-output chain, or feed-forward network on $[0,1]^d$ with some $L = \Theta(B)$. Then, provided that $\frac{\epsilon}{B}$ is sufficiently small, it is necessary that*

$$T = \Omega\left(\left(\frac{B(cL)^{m-1}}{\epsilon}\right)^{d/\nu}\right)$$

*for some $c = \Theta(1)$.*

The proof is given in Appendix H, and the high-level steps are similar to (Bull, 2011), but the main differences are significant. For each structure, we consider a collection of $M$ hard functions $\mathcal{G} = \{g_1, \ldots, g_M\}$, where each $g_j$ is obtained by shifting a base function $\bar{g}$ of the specified structure and cropping the shifted function into $[0,1]^d$. Then, we show that there exists a worst-case function in $\mathcal{G}$ with the provided lower bound. Different from (Bull, 2011), the hard functions we construct here are function networks. We define the first layer as a "needle" function with much smaller height and width than (Bull, 2011). For subsequent layers, we construct a function with corresponding RKHS norm such that the output is always larger than the input. As a consequence, the "needle" function gets higher when being fed into subsequent layers, and the composite function is a "needle" function with some specified height but a much smaller width.

**Remark 3.** It will be evident from the proof that the constant $c$ is always strictly less than 1. Ideally, we would like it to be close to 1 so that $(cL)^m$ is similar to $L^m$, with the latter quantity appearing in our upper bounds. It turns out that $c$ can indeed be arbitrarily close to 1 in most cases. Specifically, we will show in Appendix H.5 that when $\nu > 1$ and $\epsilon$ is small enough, $c$ simply becomes the ratio of the minimum slope to the maximum slope of the kernel function (as a function of the Euclidean distance $\|\mathbf{x} - \mathbf{x}'\|$) on $[u - \widetilde{u}, u + \widetilde{u}]$, where $u$ and $\widetilde{u}$ can be arbitrarily small. Since the squared exponential (SE) and Matérn kernels have no sharp changes as a function of $\|\mathbf{x} - \mathbf{x}'\|$, this ensures that $c$ can be arbitrarily close to one when $\nu > 1$ and $\epsilon$ is small. In Appendix H.5, we will also demonstrate cases where $c$ is not too small (e.g., $c > 0.93$) even when the above-mentioned quantities $(u, \widetilde{u})$ are moderate (e.g., $(u, \widetilde{u}) = (0.5, 0.3)$).

The lower bound on simple regret readily implies the following lower bound on cumulative regret.

**Theorem 6** (Lower bound on cumulative regret). *Fix sufficiently large $B > 0$, $k = k_{Matérn}$, and $\Gamma = \Gamma_{Matérn}$ with smoothness $\nu \geq 1$. Suppose that there exists an algorithm (possibly randomized) that achieves average cumulative regret $\mathbb{E}[R_T]$ after $T$ rounds for any m-layer chain, multi-output chain, or feed-forward network on $[0,1]^d$ with some $L = \Theta(B)$. Then, it is necessary that*

$$\mathbb{E}[R_T] = \begin{cases} \Omega\big(\min\{T, B(cL)^{m-1}T^{1-\nu/d}\}\big) & \text{when } d > \nu, \\ \Omega\big(\min\{T, \big(B(cL)^{m-1}\big)^{d/\nu}\}\big) & \text{when } d \leq \nu, \end{cases}$$

*for some $c = \Theta(1)$.*

The proof is given in Appendix I.

## 6  Comparison of Bounds

In this section, we compare the algorithmic upper bounds of GPN-UCB (Algorithm 1) and non-adaptive sampling (Algorithm 2) to the algorithmic-independent lower bounds in Section 5. We present our discussion conditioned on the conjecture of (Vakili, 2022) being true, but we re-iterate that even without this, any $\Sigma_T$ dependence still matches the vanilla setting, and has known rigorous upper bounds as detailed in Section 3.4. A table summarizing the regret bounds for the Matérn kernel is provided in Appendix J.

For GPN-UCB, the cumulative regret upper bound for chains (Theorem 1) matches the lower bound (Theorem 6) up to a $2^m$ factor when $d \geq \nu \geq 1$ and $T = \Omega\big((B(cL)^{m-1})^{d/\nu}\big)$. The upper bound for multi-output chains (Theorem 2) is similarly optimal (up to a $5^m$ term) when $d_{\max} = d \geq \nu \geq 1$, while there is always an $O(\sqrt{D_{2,m}})$ gap for the $T$-independent factor of feed-forward networks (Theorem 3). When $d < \nu$ and $T = \Omega\big((B(cL)^{m-1})^{d/\nu}\big)$, the cumulative regret lower bound for all the three structures is $\Omega\big((B(cL)^{m-1})^{d/\nu}\big)$,

while the upper bound always contains an $O(BL^{m-1})$ factor; hence, the terms behave similarly but some gaps still remain.

We expect that the discrepancies for multi-output chains and feed-forward networks are due to the looseness of the proposed lower bound. Since the hard functions $\mathcal{G}$ used in analysis always produce a single-entry vector output for intermediate layers, for a fixed value of $B$, there might exist a worse hard function network with more nonzero entries for intermediate outputs and a probably higher final regret.

For non-adaptive sampling, when $\nu = 1$, the upper bound for chains (Theorem 4) matches the lower bound (Theorem 5) up to a $c^{m-1}$ factor. When $\nu > 1$, Theorem 4 shows that the simple regret upper bound takes the maximum of two terms, where the first term has a matched $T$-dependent factor. However, both terms have a larger $T$-independent factor than the lower bound when $\nu > 1$; this arises due to magnifying the uncertainty from each layer to the next.

## 7 Conclusion

We have proposed an upper confidence bound based method GPN-UCB and a non-adaptive sampling based method for optimizing chains, multi-output chains, and feed-forward networks in the noise-free grey-box setting. Our regret bounds significantly improve certain dependencies compared to previous works, and we provide lower bounds that are near-matching in broad cases of interest. An immediate direction for future work is to explore noisy extensions of our algorithms (as well as lower bounds), ideally attaining analogous improvements over existing works (Kusakawa et al., 2022; Sussex et al., 2023) as those that we attained in the noiseless setting (as discussed in Appendix K).

**Acknowledgment**

This work was supported by the Singapore Ministry of Education Academic Research Fund Tier 1 under grant number A-8000872-00-00.

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
