# Appendix

## A  Applications of the Cascaded Setting

Optimization of composite functions (cascaded functions, or function networks) has wide applications in optimizing multi-stage processes, where the output of the current stage is the input of the next stage. For example, a real-world application of grey-box composite function optimization in material science is alloy heat treatment, which consists of multiple heat treatment steps and the resulting hardness after each step is available. The objective is to find the starting concentration and heat treatment (temperature and time) that maximize the hardness of the final product (Nguyen et al., 2016). Note that to find the best heat treatment for all the steps, the algorithms are expected to support additional input for intermediate layers, and we will show in Appendix K that our algorithms and theories can be easily adapted to support this.

Similarly, as an application in simulation, solar cell simulators can also utilize the technique of composite function optimization to maximize the power generation efficiency (Kusakawa et al., 2022). As discussed by (Astudillo and Frazier, 2021a), composite functions also arise in numerous areas e.g., engineering design, material design, system design, reinforcement learning, and Markov decision processes. Many analogous studies for white-box settings also exist (Drusvyatskiy and Paquette, 2019; Wang et al., 2017) with various roles highlighted in learning tasks; black-box variants then become indispensable when gradients are unavailable.

## B  Reproducing Kernel Hilbert Space (RKHS)

### B.1  Scalar-Valued Functions

For a given scalar-valued kernel $k : \mathcal{X} \times \mathcal{X} \to \mathbb{R}$, consider a function space $\mathcal{S}_k := \{f(\cdot) = \sum_{i=1}^{n_0} a_i k(\cdot, \mathbf{x}_i) : n_0 \in \mathbb{N}, a_i \in \mathbb{R}, \mathbf{x}_i \in \mathcal{X}\}$. Then, the reproducing kernel Hilbert space (RKHS) corresponding to kernel $k$, denoted by $\mathcal{H}_k$, can be obtained by forming the completion of $\mathcal{S}_k$, and the elements in $\mathcal{H}_k$ are called scalar-valued kernelized bandits or GP bandits. $\mathcal{H}_k$ is equipped with the inner product

$$\langle f, f' \rangle_k = \sum_{i=1}^{n_1} \sum_{j=1}^{n_2} a_i b_j k(\mathbf{x}_i, \mathbf{x}'_j) \tag{23}$$

for $f = \sum_{i=1}^{n_1} a_i k(\cdot, \mathbf{x}_i)$ and $f' = \sum_{j=1}^{n_2} b_j k(\cdot, \mathbf{x}'_j)$. This inner product satisfies the reproducing property, such that $\langle f(\cdot), k(\cdot, \mathbf{x}) \rangle_k = f(\mathbf{x}), \forall \mathbf{x} \in \mathcal{X}, \forall f \in \mathcal{H}_k$. The RKHS norm of $f$ is $\|f\|_k = \sqrt{\langle f, f \rangle_k}$, and we use $\mathcal{H}_k(B) := \{f \in \mathcal{H}_k : \|f\|_k \leq B\}$ to denote the set of functions whose RKHS norm is upper bounded by some known constant $B > 0$. In this work, we mainly focus on the Matérn kernel:

$$k_{\text{Matérn}}(\mathbf{x}, \mathbf{x}') = \frac{2^{1-\nu}}{\Gamma(\nu)} \left( \frac{\sqrt{2\nu} d_{\mathbf{x}, \mathbf{x}'}}{l} \right)^\nu B_\nu \left( \frac{\sqrt{2\nu} d_{\mathbf{x}, \mathbf{x}'}}{l} \right),$$

where $d_{\mathbf{x}, \mathbf{x}'} = \|\mathbf{x} - \mathbf{x}'\|_2$, $l > 0$ denotes the length-scale, $\nu > 0$ is a smoothness parameter, $\Gamma$ is the Gamma function, and $B_\nu$ is the modified Bessel function.

### B.2  Vector-Valued Functions

An operator-valued kernel $\Gamma : \mathcal{X} \times \mathcal{X} \to \mathbb{R}^{n \times n}$ is called a multi-task kernel on $\mathcal{X}$ if $\Gamma(\cdot, \cdot)$ is symmetric positive definite. Moreover, a single-task kernel with $n = 1$ recovers a scalar-valued kernel. For a given multi-task kernel $\Gamma$, similarly to the scalar-valued kernels, there exists an RKHS of vector-valued functions $\mathcal{H}_\Gamma$, which is the completion of $\mathcal{S}_\Gamma := \{f(\cdot) = \sum_{i=1}^{n_0} \Gamma(\cdot, \mathbf{x}_i) \mathbf{a}_i : n_0 \in \mathbb{N}, \mathbf{a}_i \in \mathbb{R}^n, \mathbf{x}_i \in \mathcal{X}\}$. The elements in $\mathcal{H}_\Gamma$ are called vector-valued kernelized bandits, and $\mathcal{H}_\Gamma$ is equipped with the inner product (Carmeli et al., 2006)

$$\langle f, f' \rangle_\Gamma = \sum_{i=1}^{n_1} \sum_{j=1}^{n_2} \langle \Gamma(\mathbf{x}_i, \mathbf{x}'_j) \mathbf{a}_i, \mathbf{b}_j \rangle \tag{24}$$



Figure 2: A chain with $m = 3$, where $\{f^{(i)}\}_{i=1}^{3}$ are scalar-valued functions.

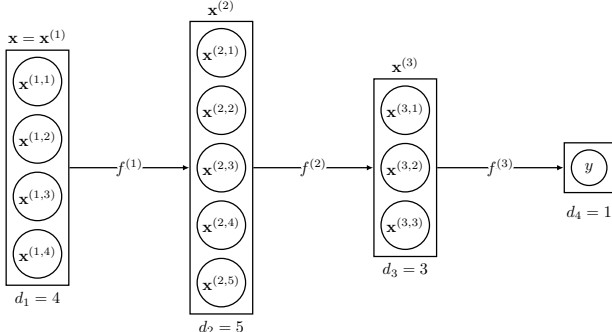

Figure 3: A multi-output chain with $m = 3$, where $\{f^{(i)}\}_{i=1}^{3}$ are vector-valued functions.

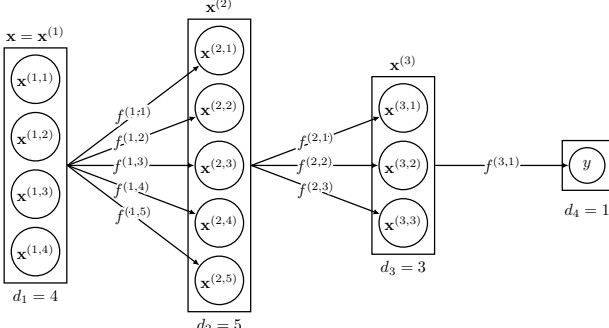

Figure 4: A feed-forward network with $m = 3$, where $f^{(i,j)}$ is a scalar-valued function for each $i \in [3], j \in [d_{i+1}]$.

for $f = \sum_{i=1}^{n_1} \Gamma(\cdot, \mathbf{x}_i)\mathbf{a}_i$ and $f' = \sum_{j=1}^{n_2} \Gamma(\cdot, \mathbf{x}'_j)\mathbf{b}_i$. This inner product satisfies the reproducing property, such that $\langle f(\cdot), \Gamma(\cdot, \mathbf{x})\mathbf{v}\rangle_\Gamma = \langle f(\mathbf{x}), \mathbf{v}\rangle_2, \forall \mathbf{x} \in \mathcal{X}, \forall \mathbf{v} \in \mathbb{R}^n, \forall f \in \mathcal{H}_\Gamma$. The RKHS norm of $f \in \mathcal{H}_\Gamma$ is $\|f\|_\Gamma = \sqrt{\langle f, f\rangle_\Gamma}$, and we focus on $\mathcal{H}_\Gamma(B) := \{f \in \mathcal{H}_\Gamma : \|f\|_\Gamma \leq B\}$ containing functions with norm at most $B > 0$. We will often pay particular attention to the Matérn kernel $\Gamma_{\text{Matérn}}(\cdot, \cdot) = k_{\text{Matérn}}(\cdot, \cdot)\mathbf{I}$, where $k_{\text{Matérn}}$ is the scalar-valued Matérn kernel and $\mathbf{I}$ is the $n \times n$ identity matrix.

## C  Figures Illustrating the Network Structures

We depict the chain structure in Figure 2, the multi-output chain structure in Figure 3, and the feed-forward network structure in Figure 4.

## D  Confidence Region for Vector-Valued Functions (Proof of Lemma 2)

In this section, we prove Lemma 2, which is restated as follows.

**Lemma 2.** *For $f \in \mathcal{H}_\Gamma(B)$, let $\mu_t(\mathbf{x})$ and $\Gamma_t(\mathbf{x}, \mathbf{x})$ denote the posterior mean and variance based on $t$ points $(\mathbf{x}_1, \ldots, \mathbf{x}_t)$ and their noise-free observations $(\mathbf{y}_1, \ldots, \mathbf{y}_t)$ using (3) and (4). Then, it holds for all $\mathbf{x} \in \mathcal{X}$ that*

$$\|f(\mathbf{x}) - \mu_t(\mathbf{x})\|_2 \leq B\|\Gamma_t(\mathbf{x}, \mathbf{x})\|_2^{1/2}.$$

*Proof.* We first review the GP posterior and confidence region for operator-valued kernel $\Gamma : \mathcal{X} \times \mathcal{X} \to \mathbb{R}^{n \times n}$ in the noisy setting.

**Lemma 3.** *(Chowdhury and Gopalan, 2021, Theorem 1)* *For $f \in \mathcal{H}_\Gamma(B)$, given a sequence of points $(\mathbf{x}_1, \ldots, \mathbf{x}_t)$ and their noisy observations $Y'_t = (\mathbf{y}'_1, \ldots, \mathbf{y}'_t)$, where $\mathbf{y}'_i = f(\mathbf{x}_i) + \epsilon_i$ with $\epsilon_i$ being i.i.d. $\sigma$-sub-Gaussian for each $i \in [t]$ for some $\sigma > 0$, let $\mu'_t(\mathbf{x})$ and $\Gamma'_t(\mathbf{x}, \mathbf{x})$ denote the posterior mean and variance computed using*

$$\mu'_t(\mathbf{x}) = G_t(\mathbf{x})^T (G_t + \lambda \mathbf{I}_{nt})^{-1} Y'_t, \tag{25}$$

$$\Gamma'_t(\mathbf{x}, \mathbf{x}') = \Gamma(\mathbf{x}, \mathbf{x}') - G_t(\mathbf{x})^T (G_t + \lambda \mathbf{I}_{nt})^{-1} G_t(\mathbf{x}'), \tag{26}$$

*where $G_t(\mathbf{x}) = [\Gamma(\mathbf{x}, \mathbf{x}_i)]^t_{i=1} \in \mathbb{R}^{nt \times n}$, $G_t = [\Gamma(\mathbf{x}_i, \mathbf{x}_j)]^t_{i,j=1} \in \mathbb{R}^{nt \times nt}$, $Y'_t = [\mathbf{y}'_i]^t_{i=1} \in \mathbb{R}^{nt \times 1}$, and $\lambda > 0$ is a regularization parameter. Then, for any $\lambda > 0$ and $\delta \in (0, 1]$, with probability at least $1 - \delta$, it holds for all $\mathbf{x} \in \mathcal{X}$ that*

$$\|f(\mathbf{x}) - \mu'_t(\mathbf{x})\|_2 \le \alpha_t \|\Gamma'_t(\mathbf{x}, \mathbf{x})\|_2^{1/2} \tag{27}$$

*with $\alpha_t = B + \frac{\sigma}{\sqrt{n}} \sqrt{2 \log(1/\delta) + \log \det(I_{nt} + \lambda^{-1} G_t)}$.*

Since zero noise is $\sigma$-sub-Gaussian for any $\sigma > 0$, by setting $\sigma = \lambda = \frac{1}{a}$ and then taking $a \to \infty$, (25) and (26) converge to

$$\mu_t(\mathbf{x}) = G_t(\mathbf{x})^T G_t^{-1} Y_t, \tag{28}$$

$$\Gamma_t(\mathbf{x}, \mathbf{x}') = \Gamma(\mathbf{x}, \mathbf{x}') - G_t(\mathbf{x})^T G_t^{-1} G_t(\mathbf{x}'), \tag{29}$$

with $Y_t = (\mathbf{y}_1, \ldots, \mathbf{y}_t)$, thus yielding the posterior mean and variance based on $(\mathbf{x}_1, \ldots, \mathbf{x}_t)$ and their noise-free observations $Y_t$.

Then, by setting $\sigma = \delta = \lambda = \frac{1}{a}$ for $a \to \infty$, with $\{\lambda_i\}_{i=1}^{nt}$ being the eigenvalues of $G_t$, we obtain

$$\lim_{a \to \infty} \alpha_t = B + \frac{1}{\sqrt{n}} \cdot \lim_{a \to \infty} \sqrt{\frac{2 \log a}{a^2} + \frac{\sum_{i=1}^{nt} \log(a\lambda_i + 1)}{a^2}} = B. \tag{30}$$

Hence, we obtain Lemma 2 for the deterministic confidence region based on noise-free observations. $\qquad\square$

# E  Posterior Variance for Vector-Valued Functions

In this section, we state and prove Lemma 4.

**Lemma 4.** *For a scalar-valued kernel $k$, define $\Gamma(\mathbf{x}, \mathbf{x}') = k(\mathbf{x}, \mathbf{x}') \mathbf{I}_n$ with $\mathbf{I}_n$ being the $n \times n$ identity matrix. For $f_1 \in \mathcal{H}_k(B)$ and $f_2 \in \mathcal{H}_\Gamma(B)$ with domain $\mathcal{X}$ and constant $B > 0$, let $\sigma_t(\mathbf{x})^2$ and $\Gamma_t(\mathbf{x}, \mathbf{x})$ denote the posterior variance (matrix) for $f_1$ and $f_2$ based on $t$ points $(\mathbf{x}_1, \ldots, \mathbf{x}_t)$ in the noise-free setting computed using (2) and (4). Then, it holds for all $\mathbf{x} \in \mathcal{X}$ that*

$$\Gamma_t(\mathbf{x}, \mathbf{x}) = \sigma_t(\mathbf{x})^2 \mathbf{I}_n. \tag{31}$$

*Proof.* Let $\sigma'_t(\mathbf{x})^2$ and $\Gamma'_t(\mathbf{x}, \mathbf{x})$ denote the posterior variance (matrix) for $f_1$ and $f_2$ based on $t$ points $(\mathbf{x}_1, \ldots, \mathbf{x}_t)$ in the noisy setting. For any $\lambda > 0$, we have

$$\sigma'_t(\mathbf{x})^2 = k(\mathbf{x}, \mathbf{x}) - \mathbf{k}_t(\mathbf{x})^T (\mathbf{K}_t + \lambda \mathbf{I}_t)^{-1} \mathbf{k}_t(\mathbf{x}), \tag{32}$$

$$\Gamma'_t(\mathbf{x}, \mathbf{x}) = \Gamma(\mathbf{x}, \mathbf{x}) - G_t(\mathbf{x})^T (G_t + \lambda \mathbf{I}_{nt})^{-1} G_t(\mathbf{x}'), \tag{33}$$

where $\mathbf{k}_t(\mathbf{x}) = [k(\mathbf{x}, \mathbf{x}_i)]^t_{i=1} \in \mathbb{R}^{t \times 1}$, $\mathbf{K}_t = [k(\mathbf{x}_i, \mathbf{x}_j)]^t_{i,j=1} \in \mathbb{R}^{t \times t}$, $G_t(\mathbf{x}) = [\Gamma(\mathbf{x}, \mathbf{x}_i)]^t_{i=1} \in \mathbb{R}^{nt \times n}$, and $G_t = [\Gamma(\mathbf{x}_i, \mathbf{x}_j)]^t_{i,j=1} \in \mathbb{R}^{nt \times nt}$. With $\otimes$ denoting the Kronecker product, we have $G_t(\mathbf{x}) = \mathbf{k}_t(\mathbf{x}) \otimes \mathbf{I}_n$ and

$G_t = \mathbf{K}_t \otimes \mathbf{I}_n$. Then, it follows from (33) that

$$\Gamma'_t(\mathbf{x}, \mathbf{x}) = \Gamma(\mathbf{x}, \mathbf{x}) - G_t(\mathbf{x})^T (G_t + \lambda \mathbf{I}_{nt})^{-1} G_t(\mathbf{x}) \tag{34}$$

$$= k(\mathbf{x}, \mathbf{x})\mathbf{I}_n - \big(\mathbf{k}_t(\mathbf{x})^T \otimes \mathbf{I}_n\big)(\mathbf{K}_t \otimes \mathbf{I}_n + \lambda \mathbf{I}_{nt})^{-1}\big(\mathbf{k}_t(\mathbf{x}) \otimes \mathbf{I}_n\big) \tag{35}$$

$$= k(\mathbf{x}, \mathbf{x})\mathbf{I}_n - \big(\mathbf{k}_t(\mathbf{x})^T \otimes \mathbf{I}_n\big)\big((\mathbf{K}_t + \lambda \mathbf{I}_t) \otimes \mathbf{I}_n\big)^{-1}\big(\mathbf{k}_t(\mathbf{x}) \otimes \mathbf{I}_n\big) \tag{36}$$

$$= k(\mathbf{x}, \mathbf{x})\mathbf{I}_n - \big(\mathbf{k}_t(\mathbf{x})^T \otimes \mathbf{I}_n\big)\big((\mathbf{K}_t + \lambda \mathbf{I}_t)^{-1} \otimes \mathbf{I}_n\big)\big(\mathbf{k}_t(\mathbf{x}) \otimes \mathbf{I}_n\big) \tag{37}$$

$$= k(\mathbf{x}, \mathbf{x})\mathbf{I}_n - \big(\mathbf{k}_t(\mathbf{x})^T (\mathbf{K}_t + \lambda \mathbf{I}_t)^{-1}\mathbf{k}_t(\mathbf{x})\big) \otimes \mathbf{I}_n \tag{38}$$

$$= k(\mathbf{x}, \mathbf{x})\mathbf{I}_n - \big(\mathbf{k}_t(\mathbf{x})^T (\mathbf{K}_t + \lambda \mathbf{I}_t)^{-1}\mathbf{k}_t(\mathbf{x})\big)\mathbf{I}_n \tag{39}$$

$$= \big(k(\mathbf{x}, \mathbf{x}) - (\mathbf{k}_t(\mathbf{x})^T (\mathbf{K}_t + \lambda \mathbf{I}_t)^{-1}\mathbf{k}_t(\mathbf{x}))\big)\mathbf{I}_n \tag{40}$$

$$= \sigma'_t(\mathbf{x})^2 \mathbf{I}_n \tag{41}$$

Taking $\lambda \to 0$, we obtain in the noise-free setting that

$$\Gamma_t(\mathbf{x}, \mathbf{x}) = \sigma_t(\mathbf{x})^2 \mathbf{I}_n. \tag{42}$$

$\square$

# F   Analysis of GPN-UCB (Algorithm 1)

## F.1   Proof of Theorem 1 (Chains)

In this section, we prove Theorem 1, which is restated as follows.

**Theorem 1** (GPN-UCB for chains). *Under the setup of Section 2, given $B > 0$ and $L > 1$, a scalar-valued kernel $k$, and a chain $g = f^{(m)} \circ f^{(m-1)} \circ \cdots \circ f^{(1)}$ with $f^{(i)} \in \mathcal{H}_k(B) \cap \mathcal{F}(L)$ for each $i \in [m]$, Algorithm 1 achieves*

$$R_T \leq 2^{m+1} B L^{m-1} \Sigma_T,$$

*where $\Sigma_T = \max\limits_{i \in [m]} \max\limits_{\mathbf{z}_1, \ldots, \mathbf{z}_T \in \mathcal{X}^{(i)}} \sum_{t=1}^T \sigma_{t-1}^{(i)}(\mathbf{z}_t).$* [5]

*Proof.* With $\bar{\mathbf{x}}_t^{(i)} = \arg\max_{\mathbf{z} \in \Delta_{t-1}^{(i)}(\mathbf{x}_t)} \overline{\mathrm{UCB}}^{(i)}(\mathbf{z})$ and $\widetilde{\mathbf{x}}_t^{(i)} = \arg\min_{\mathbf{z} \in \Delta_{t-1}^{(i)}(\mathbf{x}_t)} \overline{\mathrm{LCB}}^{(i)}(\mathbf{z})$, the simple regret is upper bounded as follows:

$$r_t = g(\mathbf{x}^*) - g(\mathbf{x}_t) \tag{43}$$

$$\leq \mathrm{UCB}_{t-1}(\mathbf{x}_t) - \overline{\mathrm{LCB}}_{t-1}^{(m)}(\widetilde{\mathbf{x}}_t^{(m)}) \tag{44}$$

$$= \overline{\mathrm{UCB}}_{t-1}^{(m)}(\bar{\mathbf{x}}_t^{(m)}) - \overline{\mathrm{LCB}}_{t-1}^{(m)}(\widetilde{\mathbf{x}}_t^{(m)}) \tag{45}$$

$$\leq \mathrm{UCB}_{t-1}^{(m)}(\mathbf{x}_t^{(m)}) + L\|\bar{\mathbf{x}}_t^{(m)} - \mathbf{x}_t^{(m)}\|_2 - \mathrm{LCB}_{t-1}^{(m)}(\mathbf{x}_t^{(m)}) + L\|\mathbf{x}_t^{(m)} - \widetilde{\mathbf{x}}_t^{(m)}\|_2 \tag{46}$$

$$\leq 2B\sigma_{t-1}^{(m)}(\mathbf{x}_t^{(m)}) + 2L \cdot \mathrm{diam}\big(\Delta_{t-1}^{(m)}(\mathbf{x}_t)\big) \tag{47}$$

$$\leq 2B\Big(\sigma_{t-1}^{(m)}(\mathbf{x}_t^{(m)}) + (2L)\sigma_{t-1}^{(m-1)}(\mathbf{x}_t^{(m-1)}) + \cdots + (2L)^{m-1}\sigma_{t-1}^{(1)}(\mathbf{x}_t^{(1)})\Big) \tag{48}$$

$$= 2B \sum_{i=1}^m (2L)^{m-i} \sigma_{t-1}^{(i)}(\mathbf{x}_t^{(i)}), \tag{49}$$

where:

- (44) follows since $g(\mathbf{x}^*) \leq \mathrm{UCB}_{t-1}(\mathbf{x}^*) \leq \mathrm{UCB}_{t-1}(\mathbf{x}_t)$ (due to the algorithm maximizing the UCB score) and $g(\mathbf{x}_t) \geq \overline{\mathrm{LCB}}_{t-1}^{(m)}(\mathbf{x}_t^{(m)}) \geq \overline{\mathrm{LCB}}_{t-1}^{(m)}(\widetilde{\mathbf{x}}_t^{(m)})$;

---

[5] In this definition and analogous definitions below, $\sigma_{t-1}^{(i)}$ is defined according to the hypothetical sampled points $\mathbf{x}_\tau^{(i)} = \mathbf{z}_\tau$ for $\tau = 1, \ldots, t-1$.

- (45) follows from (11) and the definition of $\overline{\mathbf{x}}_t^{(m)}$;

- (46) follows from (9) and (10);

- (47) follows by defining $\text{diam}\big(\Delta_{t-1}^{(i)}(\mathbf{x}_t)\big)$ as the diameter of confidence region $\Delta_{t-1}^{(i)}(\mathbf{x}_t)$;

- (48) follows from the following recursion:

$$\text{diam}\big(\Delta_{t-1}^{(i+1)}(\mathbf{x}_t)\big) = \overline{\text{UCB}}_{t-1}^{(i)}(\overline{\mathbf{x}}_t^{(i)}) - \overline{\text{LCB}}_{t-1}^{(i)}(\widetilde{\mathbf{x}}_t^{(i)}) \tag{50}$$

$$\leq 2B\sigma_{t-1}^{(i)}(\mathbf{x}_t^{(i)}) + 2L \cdot \text{diam}\big(\Delta_{t-1}^{(i)}(\mathbf{x}_t)\big), \tag{51}$$

where the inequality is obtained by following (45)-(47) with $m$ replaced by $i$.

Then, the cumulative regret is

$$R_T = \sum_{t=1}^{T} r_t \tag{52}$$

$$\leq 2B \sum_{t=1}^{T} \sum_{i=1}^{m} (2L)^{m-i} \sigma_{t-1}^{(i)}(\mathbf{x}_t^{(i)}) \tag{53}$$

$$= 2B \frac{(2L)^m - 1}{2L - 1} \Sigma_T \tag{54}$$

$$\leq 2^{m+1} B L^{m-1} \Sigma_T, \tag{55}$$

where $\Sigma_T = \max\limits_{i \in [m]} \max\limits_{\mathbf{z}_1,\dots,\mathbf{z}_T \in \mathcal{X}^{(i)}} \sum_{t=1}^{T} \sigma_{t-1}^{(i)}(\mathbf{z}_t)$. $\qquad\square$

### F.2  Proof of Theorem 2 (Multi-Output Chains)

In this section, we prove Theorem 2, which is restated as follows.

**Theorem 2** (GPN-UCB for multi-output chains). *Under the setup of Section 2, given $B > 0$ and $L > 1$, an operator-valued kernel $\Gamma$, and a multi-output chain $g = f^{(m)} \circ f^{(m-1)} \circ \cdots \circ f^{(1)}$ with $f^{(i)} \in \mathcal{H}_\Gamma(B) \cap \mathcal{F}(L)$ for each $i \in [m]$, Algorithm 1 achieves*

$$R_T \leq 5^m B L^{m-1} \Sigma_T^\Gamma,$$

*where $\Sigma_T^\Gamma = \max\limits_{i \in [m]} \max\limits_{\mathbf{z}_1,\dots,\mathbf{z}_T \in \mathcal{X}^{(i)}} \sum_{t=1}^{T} \|\Gamma_{t-1}^{(i)}(\mathbf{z}_t, \mathbf{z}_t)\|_2^{1/2}.$*

*Proof.* For each $i \in [m-1]$, there must exist $\overline{\mathbf{x}}_t^{(i)}, \widetilde{\mathbf{x}}_t^{(i)} \in \Delta_{t-1}^{(i)}(\mathbf{x}_t)$ such that the following upper bound on $\text{diam}\big(\Delta_{t-1}^{(i+1)}(\mathbf{x}_t)\big)$ in terms of $\text{diam}\big(\Delta_{t-1}^{(i)}(\mathbf{x}_t)\big)$ holds:

$$\text{diam}\big(\Delta_{t-1}^{(i+1)}(\mathbf{x}_t)\big) \leq \|f^{(i)}(\overline{\mathbf{x}}_t^{(i)}) - f^{(i)}(\widetilde{\mathbf{x}}_t^{(i)})\|_2 + \text{diam}\big(\overline{\mathcal{C}}_{t-1}^{(i)}(\overline{\mathbf{x}}_t^{(i)})\big) + \text{diam}\big(\overline{\mathcal{C}}_{t-1}^{(i)}(\widetilde{\mathbf{x}}_t^{(i)})\big) \tag{56}$$

$$\leq L \cdot \text{diam}\big(\Delta_{t-1}^{(i)}(\mathbf{x}_t)\big) + \text{diam}\big(\mathcal{C}_{t-1}^{(i)}(\overline{\mathbf{x}}_t^{(i)}, \mathbf{x}_t^{(i)})\big) + \text{diam}\big(\mathcal{C}_{t-1}^{(i)}(\widetilde{\mathbf{x}}_t^{(i)}, \mathbf{x}_t^{(i)})\big) \tag{57}$$

$$\leq L \cdot \text{diam}\big(\Delta_{t-1}^{(i)}(\mathbf{x}_t)\big) + 2 \cdot \text{diam}\big(\mathcal{C}_{t-1}^{(i)}(\mathbf{x}_t^{(i)})\big) + 2L\|\overline{\mathbf{x}}_t^{(i)} - \mathbf{x}_t^{(i)}\|_2 + 2L\|\widetilde{\mathbf{x}}_t^{(i)} - \mathbf{x}_t^{(i)}\|_2 \tag{58}$$

$$\leq 4B\|\Gamma_{t-1}^{(i)}(\mathbf{x}_t^{(i)}, \mathbf{x}_t^{(i)})\|_2^{1/2} + 5L \cdot \text{diam}\big(\Delta_{t-1}^{(i)}(\mathbf{x}_t)\big), \tag{59}$$

where:

- (56) holds since there must exist $\overline{\mathbf{x}}_t^{(i)}, \widetilde{\mathbf{x}}_t^{(i)} \in \Delta_{t-1}^{(i)}(\mathbf{x}_t)$ such that $\text{diam}\big(\Delta_{t-1}^{(i+1)}(\mathbf{x}_t)\big) = \|\overline{\mathbf{z}} - \widetilde{\mathbf{z}}\|_2$ for some $\overline{\mathbf{z}} \in \overline{\mathcal{C}}_{t-1}^{(i)}(\overline{\mathbf{x}}_t^{(i)})$ and some $\widetilde{\mathbf{z}} \in \overline{\mathcal{C}}_{t-1}^{(i)}(\widetilde{\mathbf{x}}_t^{(i)})$ (see (16)), and the right hand side follows from the triangle inequality along with $f^{(i)}(\overline{\mathbf{x}}_t^{(i)}) \in \overline{\mathcal{C}}_{t-1}^{(i)}(\overline{\mathbf{x}}_t^{(i)})$ and $f^{(i)}(\widetilde{\mathbf{x}}_t^{(i)}) \in \overline{\mathcal{C}}_{t-1}^{(i)}(\widetilde{\mathbf{x}}_t^{(i)})$ (by Lemma 2). Here $\text{diam}\big(\overline{\mathcal{C}}_{t-1}^{(i)}(\cdot)\big)$ denotes the diameter of $\overline{\mathcal{C}}_{t-1}^{(i)}(\cdot)$ (i.e., the Euclidean distance between the most distant pair of points in $\overline{\mathcal{C}}_{t-1}^{(i)}(\cdot)$);

- (57) holds since $f^{(i)}$ has Lipschitz constant $L$, and since $\overline{\mathbf{x}}_t^{(i)}, \widetilde{\mathbf{x}}_t^{(i)} \in \Delta_{t-1}^{(i)}(\mathbf{x}_t)$ and $\overline{\mathcal{C}}_{t-1}^{(i)}(\mathbf{z}) \subseteq \mathcal{C}_{t-1}^{(i)}(\mathbf{z}, \mathbf{x}_t^{(i)})$ for all $\mathbf{z} \in \Delta_{t-1}^{(i)}(\mathbf{x}_t)$ by the definition in (12);

- (58) holds since $\mathrm{diam}\big(\mathcal{C}_{t-1}^{(i)}(\mathbf{z}, \mathbf{x}_t^{(i)})\big) \leq \mathrm{diam}\big(\mathcal{C}_{t-1}^{(i)}(\mathbf{x}_t^{(i)})\big) + 2L\|\mathbf{z} - \mathbf{x}_t^{(i)}\|_2$ by the definition in (14);

- (59) holds since $\mathrm{diam}\big(\mathcal{C}_{t-1}^{(i)}(\mathbf{x}_t^{(i)})\big) = 2B\|\Gamma_{t-1}^{(i)}(\mathbf{x}_t^{(i)}, \mathbf{x}_t^{(i)})\|_2^{1/2}$ (see (13)) and $\mathbf{x}_t^{(i)}, \overline{\mathbf{x}}_t^{(i)}, \widetilde{\mathbf{x}}_t^{(i)} \in \Delta_{t-1}^{(i)}(\mathbf{x}_t)$.

Analogous to the UCB, we define $\mathrm{LCB}_t(\mathbf{x}) = \min_{\mathbf{z} \in \Delta_t^{(m)}(\mathbf{x})} \overline{\mathcal{C}}_t^{(m)}(\mathbf{z})$. Moreover, we define $\overline{\mathbf{x}}_t^{(m)} = \arg\max_{\mathbf{z} \in \Delta_{t-1}^{(m)}(\mathbf{x}_t)} \big(\max \overline{\mathcal{C}}_{t-1}^{(m)}(\mathbf{z})\big)$, and $\widetilde{\mathbf{x}}_t^{(m)} = \arg\min_{\mathbf{z} \in \Delta_{t-1}^{(m)}(\mathbf{x}_t)} \big(\min \overline{\mathcal{C}}_{t-1}^{(m)}(\mathbf{z})\big)$. Then, we have

$$r_t = g(\mathbf{x}^*) - g(\mathbf{x}_t) \tag{60}$$
$$\leq \mathrm{UCB}_{t-1}(\mathbf{x}^*) - \mathrm{LCB}_{t-1}(\mathbf{x}_t) \tag{61}$$
$$\leq \mathrm{UCB}_{t-1}(\mathbf{x}_t) - \mathrm{LCB}_{t-1}(\mathbf{x}_t) \tag{62}$$
$$= \max \overline{\mathcal{C}}_{t-1}^{(m)}(\overline{\mathbf{x}}_t^{(m)}) - \min \overline{\mathcal{C}}_{t-1}^{(m)}(\widetilde{\mathbf{x}}_t^{(m)}) \tag{63}$$
$$\leq \Big( \mu_{t-1}^{(m)}(\mathbf{x}_t^{(m)}) + B\|\Gamma_{t-1}^{(m)}(\mathbf{x}_t^{(m)}, \mathbf{x}_t^{(m)})\|_2^{1/2} + L\|\overline{\mathbf{x}}_t^{(m)} - \mathbf{x}_t^{(m)}\|_2 \Big)$$
$$\quad - \Big( \mu_{t-1}^{(m)}(\mathbf{x}_t^{(m)}) - B\|\Gamma_{t-1}^{(m)}(\mathbf{x}_t^{(m)}, \mathbf{x}_t^{(m)})\|_2^{1/2} - L\|\widetilde{\mathbf{x}}_t^{(m)} - \mathbf{x}_t^{(m)}\|_2 \Big) \tag{64}$$
$$\leq 2B\|\Gamma_{t-1}^{(m)}(\mathbf{x}_t^{(m)}, \mathbf{x}_t^{(m)})\|_2^{1/2} + 2L \cdot \mathrm{diam}\big(\Delta_{t-1}^{(m)}(\mathbf{x}_t)\big) \tag{65}$$
$$\leq 2B\|\Gamma_{t-1}^{(m)}(\mathbf{x}_t^{(m)}, \mathbf{x}_t^{(m)})\|_2^{1/2} + 2L\Big( 4B\|\Gamma_{t-1}^{(m-1)}(\mathbf{x}_t^{(m-1)}, \mathbf{x}_t^{(m-1)})\|_2^{1/2} + 5L \cdot \mathrm{diam}\big(\Delta_{t-1}^{(m-1)}(\mathbf{x}_t)\big)\Big) \tag{66}$$
$$\leq 2B\|\Gamma_{t-1}^{(m)}(\mathbf{x}_t^{(m)}, \mathbf{x}_t^{(m)})\|_2^{1/2} + \sum_{i=1}^{m-1} 2L \cdot (5L)^{m-1-i} \cdot 4B\|\Gamma_{t-1}^{(i)}(\mathbf{x}_t^{(i)}, \mathbf{x}_t^{(i)})\|_2^{1/2}, \tag{67}$$

where:

- (64) holds since

$$\max \overline{\mathcal{C}}_{t-1}^{(m)}(\overline{\mathbf{x}}_t^{(m)}) \leq \max \mathcal{C}_{t-1}^{(m)}(\overline{\mathbf{x}}_t^{(m)}, \mathbf{x}_t^{(m)}) \qquad \text{(by (12))} \tag{68}$$
$$\leq \max \mathcal{C}_{t-1}^{(m)}(\mathbf{x}_t^{(m)}) + L\|\overline{\mathbf{x}}_t^{(m)} - \mathbf{x}_t^{(m)}\|_2 \qquad \text{(by (14))} \tag{69}$$
$$= \mu_{t-1}^{(m)}(\mathbf{x}_t^{(m)}) + B\|\Gamma_{t-1}^{(m)}(\mathbf{x}_t^{(m)}, \mathbf{x}_t^{(m)})\|_2^{1/2} + L\|\overline{\mathbf{x}}_t^{(m)} - \mathbf{x}_t^{(m)}\|_2 \qquad \text{(by (13))}. \tag{70}$$

  Similarly, we have

$$\min \overline{\mathcal{C}}_{t-1}^{(m)}(\widetilde{\mathbf{x}}_t^{(m)}) \geq \min \mathcal{C}_{t-1}^{(m)}(\widetilde{\mathbf{x}}_t^{(m)}, \mathbf{x}_t^{(m)}) \tag{71}$$
$$\geq \min \mathcal{C}_{t-1}^{(m)}(\mathbf{x}_t^{(m)}) - L\|\widetilde{\mathbf{x}}_t^{(m)} - \mathbf{x}_t^{(m)}\|_2 \tag{72}$$
$$= \mu_{t-1}^{(m)}(\mathbf{x}_t^{(m)}) - B\|\Gamma_{t-1}^{(m)}(\mathbf{x}_t^{(m)}, \mathbf{x}_t^{(m)})\|_2^{1/2} - L\|\widetilde{\mathbf{x}}_t^{(m)} - \mathbf{x}_t^{(m)}\|_2. \tag{73}$$

- (65) holds since all of the $\mathbf{x}$ vectors in (65) lie in $\Delta_{t-1}^{(m)}(\mathbf{x}_t)$;

- (66) and (67) follow from the recursive relation in (59).

Then, the cumulative regret is

$$R_T = \sum_{t=1}^{T} r_t \tag{74}$$

$$\leq 2B \sum_{t=1}^{T} \|\Gamma_{t-1}^{(m)}(\mathbf{x}_t^{(m)}, \mathbf{x}_t^{(m)})\|_2^{1/2} + 2L \cdot 4B \sum_{t=1}^{T} \|\Gamma_{t-1}^{(m-1)}(\mathbf{x}_t^{(m-1)}, \mathbf{x}_t^{(m-1)})\|_2^{1/2}$$

$$+ \sum_{i=1}^{m-2} 2L \cdot (5L)^{m-1-i} \cdot 4B \sum_{t=1}^{T} \|\Gamma_{t-1}^{(i)}(\mathbf{x}_t^{(i)}, \mathbf{x}_t^{(i)})\|_2^{1/2} \tag{75}$$

$$\leq 4B \frac{(5L)^m - 1}{5L - 1} \Sigma_T^{\Gamma} \tag{76}$$

$$\leq 5^m B L^{m-1} \Sigma_T^{\Gamma}, \tag{77}$$

where $\Sigma_T^{\Gamma} = \max_{i \in [m]} \max_{\mathbf{z}_1,\ldots,\mathbf{z}_T \in \mathcal{X}^{(i)}} \sum_{t=1}^{T} \|\Gamma_{t-1}^{(i)}(\mathbf{z}_t, \mathbf{z}_t)\|_2^{1/2}$. $\qquad\square$

### F.3 Proof of Theorem 3 (Feed-Forward Networks)

In this section, we prove Theorem 3, which is restated as follows.

**Theorem 3** (GPN-UCB for feed-forward networks). *Under the setup of Section 2, given $B > 0$ and $L > 1$, a scalar-valued kernel $k$, and a feed-forward network $g = f^{(m)} \circ f^{(m-1)} \circ \cdots \circ f^{(1)}$ with $f^{(i)}(\mathbf{z}) = [f^{(i,j)}(\mathbf{z})]_{j=1}^{d_{i+1}}$ and $f^{(i,j)} \in \mathcal{H}_k(B) \cap \mathcal{F}(L)$ for each $i \in [m], j \in [d_{i+1}]$, Algorithm 1 achieves*

$$R_T \leq 2^{m+1} \sqrt{D_{2,m}} B L^{m-1} \Sigma_T,$$

*where $D_{2,m} = \prod_{i=2}^{m} d_i$ and $\Sigma_T = \max_{i \in [m]} \max_{\mathbf{z}_1,\ldots,\mathbf{z}_T \in \mathcal{X}^{(i)}} \sum_{t=1}^{T} \sigma_{t-1}^{(i,1)}(\mathbf{z}_t)$.*

*Proof.* Recall the UCB and LCB definitions in (17)–(18). For a fixed input $\mathbf{x} \in \mathcal{X}$, defining $\overline{\mathbf{x}}^{(i,j)} = \arg\max_{\mathbf{z} \in \Delta_t^{(i)}(\mathbf{x})} \overline{\mathrm{UCB}}_t^{(i,j)}(\mathbf{z})$ and $\widetilde{\mathbf{x}}^{(i,j)} = \arg\min_{\mathbf{z} \in \Delta_t^{(i)}(\mathbf{x})} \overline{\mathrm{LCB}}_t^{(i,j)}(\mathbf{z})$, the diameter of the confidence region of $\mathbf{x}^{(i+1,j)}$ is

$$\mathrm{diam}\big(\Delta_t^{(i+1,j)}(\mathbf{x})\big) = \max_{\mathbf{z} \in \Delta_t^{(i)}(\mathbf{x})} \overline{\mathrm{UCB}}_t^{(i,j)}(\mathbf{z}) - \min_{\mathbf{z} \in \Delta_t^{(i)}(\mathbf{x})} \overline{\mathrm{LCB}}_t^{(i,j)}(\mathbf{z}) \tag{78}$$

$$= \overline{\mathrm{UCB}}_t^{(i,j)}(\overline{\mathbf{x}}^{(i,j)}) - \overline{\mathrm{LCB}}_t^{(i,j)}(\widetilde{\mathbf{x}}^{(i,j)}) \tag{79}$$

$$\leq \mathrm{UCB}_t^{(i,j)}(\mathbf{x}^{(i)}) + L\|\overline{\mathbf{x}}^{(i,j)} - \mathbf{x}^{(i)}\|_2 - \mathrm{LCB}_t^{(i,j)}(\mathbf{x}^{(i)}) + L\|\mathbf{x}^{(i)} - \widetilde{\mathbf{x}}^{(i,j)}\|_2 \tag{80}$$

$$\leq 2B\sigma_t^{(i,j)}(\mathbf{x}^{(i)}) + 2L \cdot \mathrm{diam}\big(\Delta_t^{(i)}(\mathbf{x})\big), \tag{81}$$

and therefore, the squared diameter of the confidence region of $\mathbf{x}^{(i+1)} = [\mathbf{x}^{(i+1,1)}, \ldots, \mathbf{x}^{(i+1,d_{i+1})}]^T$ is

$$\mathrm{diam}\big(\Delta_t^{(i+1)}(\mathbf{x})\big)^2 = \sum_{j=1}^{d_{i+1}} \mathrm{diam}\big(\Delta_t^{(i+1,j)}(\mathbf{x})\big)^2 \tag{82}$$

$$\leq \sum_{j=1}^{d_{i+1}} \Big(2B\sigma_t^{(i,j)}(\mathbf{x}^{(i)}) + 2L \cdot \mathrm{diam}\big(\Delta_t^{(i)}(\mathbf{x})\big)\Big)^2 \tag{83}$$

$$= d_{i+1} \Big(2B\sigma_t^{(i,1)}(\mathbf{x}^{(i)}) + 2L \cdot \mathrm{diam}\big(\Delta_t^{(i)}(\mathbf{x})\big)\Big)^2, \tag{84}$$

where the last step follows since $\sigma_t^{(i,j)}(\cdot) = \sigma_t^{(i,1)}(\cdot)$ for each $j \in [d_{i+1}]$ by the symmetry of our setup (each function is associated with the same kernel).

Then, by recursion, the diameter of the confidence region of $\mathbf{x}^{(m)}$ is

$$\text{diam}\big(\Delta_t^{(m)}(\mathbf{x})\big) \leq 2\sqrt{d_m}B\sigma_t^{(m-1,1)}(\mathbf{x}^{(m-1)}) + 2\sqrt{d_m}L \cdot \text{diam}\big(\Delta_t^{(m-1)}(\mathbf{x})\big) \tag{85}$$

$$\leq 2\sqrt{d_m}B\sigma_t^{(m-1,1)}(\mathbf{x}^{(m-1)}) + (2\sqrt{d_m}L)(2\sqrt{d_{m-1}}B)\sigma_t^{(m-2,1)}(\mathbf{x}^{(m-2)})$$

$$+ (2\sqrt{d_m}L)(2\sqrt{d_{m-1}}L)\text{diam}\big(\Delta_t^{(m-2)}(\mathbf{x})\big) \tag{86}$$

$$\leq \sum_{i=1}^{m-1} \Big( \prod_{s=i+2}^{m} (2\sqrt{d_s}L)\Big)(2\sqrt{d_{i+1}}B)\sigma_t^{(i,1)}(\mathbf{x}^{(i)}), \tag{87}$$

where we set $\prod_{s=m+1}^{m}(2\sqrt{d_s}L) = 1$.

Similarly to (43)–(49) for the case of chains, with $\overline{\mathbf{x}}_t^{(m)} = \arg\max_{\mathbf{z}\in\Delta_{t-1}^{(m)}(\mathbf{x}_t)} \overline{\text{UCB}}_{t-1}^{(m,1)}(\mathbf{z})$ and $\widetilde{\mathbf{x}}_t^{(m)} = \arg\min_{\mathbf{z}\in\Delta_{t-1}^{(m)}(\mathbf{x}_t)} \overline{\text{LCB}}_{t-1}^{(m,1)}(\mathbf{z})$, the simple regret is upper bounded as follows:

$$r_t = g(\mathbf{x}^*) - g(\mathbf{x}_t) \tag{88}$$

$$\leq \text{UCB}_{t-1}(\mathbf{x}_t) - \overline{\text{LCB}}_{t-1}^{(m,1)}(\widetilde{\mathbf{x}}_t^{(m)}) \tag{89}$$

$$= \overline{\text{UCB}}_{t-1}^{(m,1)}(\overline{\mathbf{x}}_t^{(m)}) - \overline{\text{LCB}}_{t-1}^{(m,1)}(\widetilde{\mathbf{x}}_t^{(m)}) \tag{90}$$

$$\leq \text{UCB}_{t-1}^{(m,1)}(\mathbf{x}_t^{(m)}) + L\|\overline{\mathbf{x}}_t^{(m)} - \mathbf{x}_t^{(m)}\|_2 - \text{LCB}_{t-1}^{(m,1)}(\mathbf{x}_t^{(m)}) + L\|\mathbf{x}_t^{(m)} - \widetilde{\mathbf{x}}_t^{(m)}\|_2 \tag{91}$$

$$\leq 2B\sigma_{t-1}^{(m,1)}(\mathbf{x}_t^{(m)}) + 2L \cdot \text{diam}\big(\Delta_{t-1}^{(m)}(\mathbf{x}_t)\big) \tag{92}$$

$$\leq 2B\sigma_{t-1}^{(m,1)}(\mathbf{x}_t^{(m)}) + 2L \sum_{i=1}^{m-1} \Big( \prod_{s=i+2}^{m} (2\sqrt{d_s}L)\Big)(2\sqrt{d_{i+1}}B)\sigma_{t-1}^{(i,1)}(\mathbf{x}_t^{(i)}) \tag{93}$$

$$\leq \sum_{i=1}^{m} 2^{m-i+1}BL^{m-i}\Big( \prod_{s=i+1}^{m} \sqrt{d_s}\Big)\sigma_{t-1}^{(i,1)}(\mathbf{x}_t^{(i)}), \tag{94}$$

where we use the convention $\prod_{s=m+1}^{m} \sqrt{d_s} = 1$.

Hence, the cumulative regret is

$$R_T = \sum_{t=1}^{T} r_t \tag{95}$$

$$\leq \sum_{i=1}^{m} 2^{m-i+1}BL^{m-i}\Big( \prod_{s=i+1}^{m} (\sqrt{d_s})\Big)\sum_{t=1}^{T}\sigma_{t-1}^{(i,1)}(\mathbf{x}_t^{(i)}) \tag{96}$$

$$\leq 2B\frac{(2L)^m - 1}{2L - 1}\sqrt{D_{2,m}}\Sigma_T \tag{97}$$

$$\leq 2^{m+1}BL^{m-1}\sqrt{D_{2,m}}\Sigma_T, \tag{98}$$

where $D_{2,m} = \prod_{i=2}^{m} d_i$ and $\Sigma_T = \max_{i\in[m]} \max_{\mathbf{z}_1,\ldots,\mathbf{z}_T\in\mathcal{X}^{(i)}} \sum_{t=1}^{T}\sigma_{t-1}^{(i,1)}(\mathbf{z}_t)$. $\qquad\square$

## G   Analysis of Non-Adaptive Sampling (Algorithm 2)

As mentioned in the main body, the analysis in this appendix is restricted to the case that all domains in the network are *hyperrectangular*. This is a somewhat restrictive assumption, but we note that the primary reason for assuming this is to be able to apply Lemma 5 below. Hence, if Lemma 5 can be generalized, then our results also generalize in the same way.

The lemma, stated as follows, provides an upper bound on the posterior standard deviation on a hyper-rectangular domain for the Matérn kernel in terms of the fill distance of the sampled points and the kernel parameter.

**Lemma 5.** (Kanagawa et al., 2018, Theorem 5.4) *For the Matérn kernel with smoothness $\nu$, a hyperrectangular domain $\mathcal{X}$, and any set of $T$ points $\mathcal{X}_T \subset \mathcal{X}$ with fill distance $\delta_T$, define*

$$\bar{\sigma}_T := \max_{\mathbf{x} \in \mathcal{X}} \sigma_T(\mathbf{x}), \tag{99}$$

*where $\sigma_T(\cdot)$ is the posterior standard deviation computed based on $\mathcal{X}_T$ using (2). There exists a constant $\delta_0$ depending only on the kernel parameters such that if $\delta_T \leq \delta_0$ then*

$$\bar{\sigma}_T = O\big((\delta_T)^\nu\big). \tag{100}$$

Recalling that $\{\mathbf{x}_s^{(i)}\}_{s=1}^T$ denote the intermediate outputs of $\{\mathbf{x}_s\}_{s=1}^T$ right after $f^{(i-1)}$, we define the fill distance of the $\mathcal{X}^{(i)}$ with respect to $\{\mathbf{x}_s^{(i)}\}_{s=1}^T$ as

$$\delta_T^{(i)} = \max_{\mathbf{z} \in \mathcal{X}^{(i)}} \min_{s \in [T]} \|\mathbf{z} - \mathbf{x}_s^{(i)}\|_2. \tag{101}$$

We also provide corollaries on the posterior standard deviation of a single layer for each network structure.

**Corollary 1** (Chains). *With $k$ being the Matérn kernel with smoothness $\nu$, consider a chain $g = f^{(m)} \circ f^{(m-1)} \circ \cdots \circ f^{(1)}$ with $f^{(i)} \in \mathcal{H}_k(B) \cap \mathcal{F}(L)$ for each $i \in [m]$. For any set of $T$ points $\mathcal{X}_T = \{\mathbf{x}_s\}_{s=1}^T \subset \mathcal{X}$ with fill distance $\delta_T$, let $\mathcal{X}_T^{(i)} = \{\mathbf{x}_s^{(i)}\}_{s=1}^T$ be the noise-free observations of $f^{(i-1)} \circ \cdots \circ f^{(1)}$. Define*

$$\bar{\sigma}_T^{(i)} := \max_{\mathbf{z} \in \mathcal{X}^{(i)}} \sigma_T^{(i)}(\mathbf{z}), \tag{102}$$

*where $\sigma_T^{(i)}(\cdot)$ is the posterior standard deviation computed based on $\mathcal{X}_T^{(i)}$ using (2). Then, for any $i \in [m]$, there exists a constant $\delta_0$ depending only on the kernel parameters such that if $\delta_T \leq \delta_0$ then*

$$\bar{\sigma}_T^{(i)} = O\big((L^{i-1}\delta_T)^\nu\big). \tag{103}$$

*Proof.* First, it is straightforward that $f^{(i-1)} \circ \cdots \circ f^{(1)}$ has Lipschitz constant $L^{i-1}$. For any input $\mathbf{x}, \mathbf{x}_s \in \mathcal{X}$, $f^{(i-1)} \circ \cdots \circ f^{(1)}$ outputs $\mathbf{x}^{(i)}, \mathbf{x}_s^{(i)}$ such that

$$|\mathbf{x}^{(i)} - \mathbf{x}_s^{(i)}| = |f^{(i-1)} \circ \cdots \circ f^{(1)}(\mathbf{x}) - f^{(i-1)} \circ \cdots \circ f^{(1)}(\mathbf{x}_s)| \leq L^{i-1}\|\mathbf{x} - \mathbf{x}_s\|_2. \tag{104}$$

If $\{\mathbf{x}_s\}_{s=1}^T$ has fill distance $\delta_T$, then the fill distance of $\{\mathbf{x}_s^{(i)}\}_{s=1}^T$ is

$$\delta_T^{(i)} = \max_{\mathbf{z} \in \mathcal{X}^{(i)}} \min_{s \in [T]} |\mathbf{z} - \mathbf{x}_s^{(i)}| = \max_{\mathbf{x} \in \mathcal{X}} \min_{s \in [T]} |\mathbf{x}^{(i)} - \mathbf{x}_s^{(i)}| \leq \max_{\mathbf{x} \in \mathcal{X}} \min_{s \in [T]} L^{i-1}\|\mathbf{x} - \mathbf{x}_s\|_2 = L^{i-1}\delta_T. \tag{105}$$

Then, by Lemma 5 we have

$$\bar{\sigma}_T^{(i)} = \max_{\mathbf{z} \in \mathcal{X}^{(i)}} \sigma_T^{(i)}(\mathbf{z}) = O\big((\delta_T^{(i)})^\nu\big) = O\big((L^{i-1}\delta_T)^\nu\big). \tag{106}$$

$\square$

**Corollary 2** (Multi-output chains). *For $\Gamma(\cdot, \cdot) = k(\cdot, \cdot)\mathbf{I}_n$, with $k$ being the Matérn kernel with smoothness $\nu$ and $\mathbf{I}_n$ being the identity matrix of size $n$, consider a chain $g = f^{(m)} \circ f^{(m-1)} \circ \cdots \circ f^{(1)}$ with $f^{(i)} \in \mathcal{H}_\Gamma(B) \cap \mathcal{F}(L)$ and $\mathcal{X}^{(i)}$ being a hyperrectangle for each $i \in [m]$ and $n = d_{i+1}$. For any set of $T$ points $\mathcal{X}_T = \{\mathbf{x}_s\}_{s=1}^T \subset \mathcal{X}$ with fill distance $\delta_T$, let $\mathcal{X}_T^{(i)} = \{\mathbf{x}_s^{(i)}\}_{s=1}^T$ be the noise-free observations of $f^{(i-1)} \circ \cdots \circ f^{(1)}$. Define*

$$\overline{\Gamma}_T^{(i)} := \max_{\mathbf{z} \in \mathcal{X}^{(i)}} \|\Gamma_T^{(i)}(\mathbf{z}, \mathbf{z})\|_2^{1/2}, \tag{107}$$

*where $\Gamma_T^{(i)}(\mathbf{z}, \mathbf{z})$ is the posterior variance matrix computed based on $\mathcal{X}_T^{(i)}$ using (4), and $\|\cdot\|_2$ denotes the matrix spectral norm. Then, for any $i \in [m]$, there exists a constant $\delta_0$ depending only on the kernel parameters such that if $\delta_T \leq \delta_0$ then*

$$\overline{\Gamma}_T^{(i)} = O\big((L^{i-1}\delta_T)^\nu\big). \tag{108}$$

*Proof.* Recalling that $\|\cdot\|_2$ with a matrix argument denotes the spectral norm, we have

$$\overline{\Gamma}_T^{(i)} = \max_{\mathbf{z} \in \mathcal{X}^{(i)}} \|\Gamma_T^{(i)}(\mathbf{z}, \mathbf{z})\|_2^{1/2} \tag{109}$$

$$= \max_{\mathbf{z} \in \mathcal{X}^{(i)}} \|\left(\sigma_T^{(i)}(\mathbf{z})\right)^2 \mathbf{I}_n\|_2^{1/2} \qquad \text{(by Lemma 4)} \tag{110}$$

$$= \max_{\mathbf{z} \in \mathcal{X}^{(i)}} \sigma_T^{(i)}(\mathbf{z}) \qquad \text{(by } \|\mathbf{I}_n\|_2 = 1) \tag{111}$$

$$= O\left((\delta_T^{(i)})^\nu\right) \qquad \text{(by Lemma 5)} \tag{112}$$

$$= O\left((L^{i-1}\delta_T)^\nu\right). \qquad \text{(by (105))} \tag{113}$$

$\square$

**Corollary 3** (Feed-forward networks). *With $k$ being the Matérn kernel with smoothness $\nu$, consider a feed-forward network $g = f^{(m)} \circ f^{(m-1)} \circ \cdots \circ f^{(1)}$ with $f^{(s)}(\cdot) = [f^{(s,j)}(\cdot)]_{j=1}^{d_{s+1}}$ and $f^{(s,j)} \in \mathcal{H}_k(B) \cap \mathcal{F}(L)$ and $\mathcal{X}^{(s)}$ being a hyperrectangle for each $s \in [m], j \in [d_{s+1}]$. For any set of $T$ points $\mathcal{X}_T = \{\mathbf{x}_s\}_{s=1}^T \subset \mathcal{X}$ with fill distance $\delta_T$, let $\mathcal{X}_T^{(i)} = \{\mathbf{x}_s^{(i)}\}_{s=1}^t$ be the noise-free observations of $f^{(i-1)} \circ \cdots \circ f^{(1)}$. Define*

$$\overline{\sigma}_T^{(i,j)} := \max_{\mathbf{z} \in \mathcal{X}^{(i)}} \sigma_T^{(i,j)}(\mathbf{z}), \tag{114}$$

*where $\sigma_T^{(i,j)}(\cdot)$ is the posterior standard deviation computed based on $\mathcal{X}_T^{(i)}$ using (2). Then, for each $i \in [m]$ and $j \in [d_{i+1}]$, there exists a constant $\delta_0$ depending only on the kernel parameters such that if $\delta_T \leq \delta_0$ then*

$$\overline{\sigma}_T^{(i,j)} = O\left((\sqrt{D_{2,i}}L^{i-1}\delta_T)^\nu\right), \tag{115}$$

*where $D_{2,i} = \prod_{s=2}^i d_s$.*

*Proof.* Since $f^{(s,j)} \in \mathcal{F}(L)$ for each $s \in [m]$, we have for any $\mathbf{z}, \mathbf{z}' \in \mathcal{X}^{(s)}$ that

$$\|f^{(s)}(\mathbf{z}) - f^{(s)}(\mathbf{z}')\|_2 = \sqrt{\sum_{j=1}^{d_{s+1}} \|f^{(s,j)}(\mathbf{z}) - f^{(s,j)}(\mathbf{z}')\|_2^2} \tag{116}$$

$$\leq \sqrt{\sum_{j=1}^{d_{s+1}} L^2\|\mathbf{z} - \mathbf{z}'\|_2^2} \tag{117}$$

$$= \sqrt{d_{s+1}}L\|\mathbf{z} - \mathbf{z}'\|_2, \tag{118}$$

and therefore $f^{(i-1)} \circ \cdots \circ f^{(1)}$ is Lipschitz continuous with constant $\sqrt{D_{2,i}}L^{i-1}$, where $D_{2,i} = \prod_{s=1}^i d_s$. If $\{\mathbf{x}_s\}_{s=1}^T$ has fill distance $\delta_T$, then the fill distance of $\{\mathbf{x}_s^{(i)}\}_{s=1}^T$ is

$$\delta_T^{(i)} = \max_{\mathbf{z} \in \mathcal{X}^{(i)}} \min_{s \in [T]} \|\mathbf{z} - \mathbf{x}_s^{(i)}\|_2 \tag{119}$$

$$= \max_{\mathbf{x} \in \mathcal{X}} \min_{s \in [T]} \|\mathbf{x}^{(i)} - \mathbf{x}_s^{(i)}\|_2 \tag{120}$$

$$\leq \max_{\mathbf{x} \in \mathcal{X}} \min_{s \in [T]} \sqrt{D_{2,i}}L^{i-1}\|\mathbf{x} - \mathbf{x}_s\|_2 \tag{121}$$

$$\leq \sqrt{D_{2,i}}L^{i-1}\delta_T. \tag{122}$$

Hence, by Lemma 5 we have

$$\overline{\sigma}_T^{(i,j)} = \max_{\mathbf{z} \in \mathcal{X}^{(i)}} \sigma_T^{(i,j)}(\mathbf{z}) = O\left((\delta_T^{(i)})^\nu\right) = O\left((\sqrt{D_{2,i}}L^{i-1}\delta_T)^\nu\right). \tag{123}$$

$\square$

**Remark 4.** Corollary 2 and Corollary 3 require $\mathcal{X}^{(i)}$ being a hyperrectangle. This is because they are the consequences of Lemma 5, which only holds for a hyperrectangular domain. For chains, the requirement of hyperrectangular domain is trivial, since $\mathcal{X} = [0,1]^d$ and $\mathcal{X}^{(2)}, \ldots, \mathcal{X}^{(m)}$ have single dimension.

### G.1 Proof of Theorem 4 (Chains)

In this section, we prove Theorem 4, which is restated as follows.

**Theorem 4** (Non-adaptive sampling method for chains)**.** *Under the setup of Section 2, given $B = \Theta(L)$, $k = k_{Matérn}$ with smoothness $\nu$, and a chain $g = f^{(m)} \circ f^{(m-1)} \circ \cdots \circ f^{(1)}$ with $f^{(i)} \in \mathcal{H}_k(B) \cap \mathcal{F}(L)$ for each $i \in [m]$, we have*

- *When $\nu \leq 1$, Algorithm 2 achieves*

$$r_T^* = \widetilde{O}(BL^{m-1}T^{-\nu^m/d});$$

- *When $\nu > 1$, Algorithm 2 achieves*

$$r_T^* = \widetilde{O}\big(\max\big\{BL^{(m-1)\nu}T^{-\nu/d}, B^{1+\nu+\nu^2+\cdots+\nu^{m-2}}L^{\nu^{m-1}}T^{-\nu^2/d}\big\}\big).$$

*Proof.* Since $f^{(i)} \in \mathcal{F}(L)$ implies that $f^{(m)} \circ \cdots \circ f^{(i)}$ is Lipschitz continuous with constant $L^{m-i+1}$, defining

$$\widetilde{\sigma}_T^{(i)}(\mathbf{x}) := (\sigma_T^{(i)} \circ \mu_T^{(i-1)} \circ \cdots \circ \mu_T^{(1)})(\mathbf{x}), \tag{124}$$

we have for all $\mathbf{x} \in \mathcal{X}$ that

$$g(\mathbf{x}) = (f^{(m)} \circ f^{(m-1)} \circ \cdots \circ f^{(2)})\big(f^{(1)}(\mathbf{x})\big) \tag{125}$$

$$\leq (f^{(m)} \circ f^{(m-1)} \circ \cdots \circ f^{(2)})\big(\mu_T^{(1)}(\mathbf{x})\big) + L^{m-1}B\sigma_T^{(1)}(\mathbf{x}) \tag{126}$$

$$= (f^{(m)} \circ f^{(m-1)} \circ \cdots \circ f^{(3)})\big((f^{(2)} \circ \mu_T^{(1)})(\mathbf{x})\big) + L^{m-1}B\sigma_T^{(1)}(\mathbf{x}) \tag{127}$$

$$\leq (f^{(m)} \circ f^{(m-1)} \circ \cdots \circ f^{(3)})\big((\mu_T^{(2)} \circ \mu_T^{(1)})(\mathbf{x})\big) + L^{m-2}B\sigma_T^{(2)}\big(\mu_T^{(1)}(\mathbf{x})\big) + L^{m-1}B\sigma_T^{(1)}(\mathbf{x}) \tag{128}$$

$$\leq (\mu_T^{(m)} \circ \mu_T^{(m-1)} \circ \cdots \circ \mu_T^{(1)})(\mathbf{x}) + B\sigma_T^{(m)}\big((\mu_T^{(m-1)} \circ \cdots \circ \mu_T^{(1)})(\mathbf{x})\big)$$
$$+ LB\sigma_T^{(m-1)}\big((\mu_T^{(m-2)} \circ \cdots \circ \mu_T^{(1)})(\mathbf{x})\big) + \cdots + L^{m-1}B\sigma_T^{(1)}(\mathbf{x}) \tag{129}$$

$$= (\mu_T^{(m)} \circ \mu_T^{(m-1)} \circ \cdots \circ \mu_T^{(1)})(\mathbf{x}) + B\sum_{i=1}^{m} L^{m-i}(\sigma_T^{(i)} \circ \mu_T^{(i-1)} \circ \cdots \circ \mu^{(1)})(\mathbf{x}) \tag{130}$$

$$= (\mu_T^{(m)} \circ \mu_T^{(m-1)} \circ \cdots \circ \mu_T^{(1)})(\mathbf{x}) + B\sum_{i=1}^{m} L^{m-i}\widetilde{\sigma}_T^{(i)}(\mathbf{x}), \tag{131}$$

where the first two inequalities use the confidence bounds and Lipschitz assumption, and the third inequality follows by continuing recursively. Similarly, we also have

$$g(\mathbf{x}) \geq (\mu_T^{(m)} \circ \mu_T^{(m-1)} \circ \cdots \circ \mu_T^{(1)})(\mathbf{x}) - B\sum_{i=1}^{m} L^{m-i}\widetilde{\sigma}_T^{(i)}(\mathbf{x}). \tag{132}$$

Hence, defining $\widetilde{\sigma}_T^{(i)} := \max_{\mathbf{x} \in \mathcal{X}} \widetilde{\sigma}_T^{(i)}(\mathbf{x})$, it follows that

$$r_T^* = g(\mathbf{x}^*) - g(\mathbf{x}_T^*) \tag{133}$$

$$\leq \Big((\mu_T^{(m)} \circ \mu_T^{(m-1)} \circ \cdots \circ \mu_T^{(1)})(\mathbf{x}^*) + B\sum_{i=1}^{m} L^{m-i}\widetilde{\sigma}_T^{(i)}(\mathbf{x}^*)\Big)$$

$$- \Big((\mu_T^{(m)} \circ \mu_T^{(m-1)} \circ \cdots \circ \mu_T^{(1)})(\mathbf{x}_T^*) - B\sum_{i=1}^{m} L^{m-i}\widetilde{\sigma}_T^{(i)}(\mathbf{x}_T^*)\Big) \tag{134}$$

$$\leq 2B\sum_{i=1}^{m} L^{m-i}\max\{\widetilde{\sigma}_T^{(i)}(\mathbf{x}^*), \widetilde{\sigma}_T^{(i)}(\mathbf{x}_T^*)\} \tag{135}$$

$$\leq 2B\sum_{i=1}^{m} L^{m-i}\widetilde{\sigma}_T^{(i)}, \tag{136}$$

where (135) uses the fact that $\mathbf{x}_T^*$ maximizes the composite posterior mean function. It remains to derive an upper bound on $\widetilde{\sigma}_T^{(i)}$ for $i \in [m]$. Firstly, by Lemma 5, we have

$$\widetilde{\sigma}_T^{(1)}(\mathbf{x}) = \sigma_T^{(1)}(\mathbf{x}) = O\big((\delta_T)^\nu\big) = O(T^{-\nu/d}). \tag{137}$$

We now represent the upper bound on $\widetilde{\sigma}_T^{(i+1)}$ in terms of $\{\widetilde{\sigma}_T^{(j)} : j \in [i]\}$ for $i \in [m-1]$. For all $\mathbf{x} \in \mathcal{X}$, the distance between $(\mu_T^{(i)} \circ \cdots \circ \mu_T^{(1)})(\mathbf{x})$ and its closest point in $\mathcal{X}^{(i+1)}$ is

$$\delta_{i+1}(\mathbf{x}) = \min_{\mathbf{z} \in \mathcal{X}^{(i+1)}} |(\mu_T^{(i)} \circ \cdots \circ \mu_T^{(1)})(\mathbf{x}) - \mathbf{z}| \tag{138}$$

$$\leq |\mu_T^{(i)} \circ \cdots \circ \mu_T^{(1)}(\mathbf{x}) - f^{(i)} \circ \cdots \circ f^{(1)}(\mathbf{x})| \tag{139}$$

$$= B \sum_{j=1}^i L^{i-j} \widetilde{\sigma}_T^{(j)}(\mathbf{x}), \tag{140}$$

where the last step follows from (131). Then, recalling the definition of $\delta_T^{(i)}$ in (101), we have that the distance between $(\mu_T^{(i)} \circ \cdots \circ \mu_T^{(1)})(\mathbf{x})$ and its closest point in $\mathcal{X}_T^{(i+1)}$ is

$$\delta'_{i+1}(\mathbf{x}) = \min_{s \in [T]} |(\mu_T^{(i)} \circ \cdots \circ \mu_T^{(1)})(\mathbf{x}) - \mathbf{x}_s^{(i+1)}| \tag{141}$$

$$\leq \delta_T^{(i+1)} + \delta_{i+1}(\mathbf{x}) \tag{142}$$

$$\leq O(L^i T^{-1/d}) + B \sum_{j=1}^i L^{i-j} \widetilde{\sigma}_T^{(j)}(\mathbf{x}) \tag{143}$$

$$= O\Big( \max \Big\{ L^i T^{-1/d}, iB \cdot \max_{j \in [i]} L^{i-j} \widetilde{\sigma}_T^{(j)}(\mathbf{x}) \Big\} \Big), \tag{144}$$

where (142) applies the triangle inequality, and (143) uses (105) and (140).

Now, we extend $\mathcal{X}^{(i+1)}$ to $\widetilde{\mathcal{X}}^{(i+1)}$, the shortest interval that covers both $\{\mu_T^{(i)} \circ \cdots \circ \mu_T^{(1)}(\mathbf{x}) : \mathbf{x} \in \mathcal{X}\}$ and $\mathcal{X}^{(i+1)}$. Then, the fill distance of the extended domain with respect to $\mathcal{X}_T^{(i+1)}$ is

$$\delta'_{i+1} := \max_{\mathbf{x} \in \mathcal{X}} \delta'_{i+1}(\mathbf{x}) = O\Big( \max \Big\{ L^i T^{-1/d}, iB \cdot \max_{j \in [i]} L^{i-j} \widetilde{\sigma}_T^{(j)} \Big\} \Big), \tag{145}$$

and by Lemma 5, we have

$$\widetilde{\sigma}_T^{(i+1)} = \max_{\mathbf{x} \in \mathcal{X}} \widetilde{\sigma}_T^{(i+1)}(\mathbf{x}) = O\big((\delta'_{i+1})^\nu\big) = O\Big( \Big( \max \Big\{ L^i T^{-1/d}, iB \cdot \max_{j \in [i]} L^{i-j} \widetilde{\sigma}_T^{(j)} \Big\} \Big)^\nu \Big). \tag{146}$$

We now split the analysis into two cases.

The case that $\nu \leq 1$: Recall that we assume $B = \Theta(L)$, and using the fact that $\widetilde{O}(L^{c\nu})$ is equivalent to $\overline{O\big((L^{c\nu} \mathrm{poly}(c \log L))\big)}$ for fixed $\nu$, it follows from (137) and (146) that when $\nu \leq 1$, we have

$$\widetilde{\sigma}_T^{(2)} = O\Big( \max \Big\{ L^\nu T^{-\nu/d}, B^\nu T^{-\nu^2/d} \Big\} \Big) = O(L^\nu T^{-\nu^2/d}), \tag{147}$$

$$\widetilde{\sigma}_T^{(3)} = O\Big( \max \Big\{ L^{2\nu} T^{-\nu/d}, 2^\nu B^\nu L^\nu T^{-\nu^2/d}, 2^\nu B^\nu L^{2} T^{-\nu^3/d} \Big\} \Big) = O(2^\nu B^\nu L^\nu T^{-\nu^3/d}) = \widetilde{O}(B^\nu L^\nu T^{-\nu^3/d}), \tag{148}$$

$$\vdots$$

$$\widetilde{\sigma}_T^{(i)} = \widetilde{O}\big((i-1)^\nu B^\nu L^{(i-2)\nu} T^{-\nu^i/d}\big) = \widetilde{O}(B^\nu L^{(i-2)\nu} T^{-\nu^i/d}), \tag{149}$$

and the simple regret is

$$r_T^* \leq 2B \sum_{i=1}^m L^{m-i} \widetilde{\sigma}_T^{(i)} = \widetilde{O}(BL^{m-1} T^{-\nu^m/d}). \tag{150}$$

The case that $\nu \leq 1$: With $B = \Theta(L)$, using the property of $\widetilde{O}(\cdot)$ similarly to the first case, it follows from (137) and (146) that when $\nu > 1$, we have

$$\widetilde{\sigma}_T^{(2)} = O\Big( \max\Big\{ L^\nu T^{-\nu/d}, B^\nu T^{-\nu^2/d} \Big\}\Big) = O(L^\nu T^{-\nu/d}), \tag{151}$$

$$\widetilde{\sigma}_T^{(3)} = O\Big( \max\Big\{ L^{2\nu} T^{-\nu/d}, 2^\nu B^\nu L^{\nu^2} T^{-\nu^2/d} \Big\}\Big) = O(2^\nu B^\nu L^{\nu^2} T^{-\nu/d}) = \widetilde{O}(B^\nu L^{\nu^2} T^{-\nu/d}), \tag{152}$$

$$\vdots$$

$$\widetilde{\sigma}_T^{(i)} = \widetilde{O}\Big( \max\Big\{ L^{(i-1)\nu} T^{-\nu/d}, (i-1)^\nu B^{\nu + \cdots + \nu^{i-2}} L^{\nu^{i-1}} T^{-\nu^2/d} \Big\}\Big)$$
$$= \widetilde{O}\Big( \max\Big\{ L^{(i-1)\nu} T^{-\nu/d}, B^{\nu + \cdots + \nu^{i-2}} L^{\nu^{i-1}} T^{-\nu^2/d} \Big\}\Big), \tag{153}$$

and the simple regret is

$$r_T^* \leq 2B \sum_{i=1}^m L^{m-i} \widetilde{\sigma}_T^{(i)} = \widetilde{O}\Big( \max\Big\{ BL^{(m-1)\nu} T^{-\nu/d}, B^{1+\nu+\nu^2+\cdots+\nu^{m-2}} L^{\nu^{m-1}} T^{-\nu^2/d} \Big\}\Big). \tag{154}$$

$$\square$$

## G.2 Two More Restrictive Cases for Chains

Recall the following two more restrictive cases introduced in the main body:

- **Case 1**: We additionally assume that $(\mu_T^{(i)} \circ \cdots \circ \mu_T^{(1)})(\mathbf{x}^*) \in \mathcal{X}^{(i+1)}$ and $(\mu_T^{(i)} \circ \cdots \circ \mu_T^{(1)})(\mathbf{x}_T^*) \in \mathcal{X}^{(i+1)}$ for $i \in [m-1]$.

- **Case 2**: We additionally assume that all the $\mathcal{X}^{(i)}$'s are known. Defining

$$\widetilde{\mu}_T^{(i)}(\mathbf{z}) = \underset{\mathbf{z}' \in \mathcal{X}^{(i+1)}}{\arg\min} |\mu_T^{(i)}(\mathbf{z}) - \mathbf{z}'|, \tag{155}$$

we let the algorithm return

$$\mathbf{x}_T^* = \underset{\mathbf{x} \in \mathcal{X}}{\arg\max}(\widetilde{\mu}_T^{(m)} \circ \cdots \circ \widetilde{\mu}_T^{(1)})(\mathbf{x}). \tag{156}$$

In Case 1, it follows from Corollary 1 (along with (115) and $\delta_T = \Theta(T^{-\frac{1}{d}})$) that for each $\mathbf{x}' \in \{\mathbf{x}^*, \mathbf{x}_T^*\}$

$$\widetilde{\sigma}_T^{(i+1)}(\mathbf{x}') = (\sigma_T^{(i+1)} \circ \mu_T^{(i)} \circ \cdots \circ \mu_T^{(1)})(\mathbf{x}') \leq \overline{\sigma}_T^{(i+1)} = O\big((L^i \delta_T)^\nu\big) = O\big((L^i T^{-1/d})^\nu\big). \tag{157}$$

By substituting these upper bounds into (135), it holds that

$$r_T^* \leq 2B \sum_{i=1}^m L^{m-i} \max\{\widetilde{\sigma}_T^{(i)}(\mathbf{x}^*), \widetilde{\sigma}_T^{(i)}(\mathbf{x}_T^*)\} \tag{158}$$

$$\leq 2B \sum_{i=1}^m L^{m-i} O\big((L^i T^{-1/d})^\nu\big) \tag{159}$$

$$= \begin{cases} O(BL^{m-1} T^{-\nu/d}) & \text{when } \nu \leq 1, \\ O(BL^{(m-1)\nu} T^{-\nu/d}) & \text{when } \nu > 1. \end{cases} \tag{160}$$

In Case 2, for all $\mathbf{z} \in \mathcal{X}^{(i)}$, we have $\widetilde{\mu}_T^{(i)}(\mathbf{z}) \in \mathcal{X}^{(i+1)}$, and

$$|f^{(i)}(\mathbf{z}) - \widetilde{\mu}_T^{(i)}(\mathbf{z})| \leq |f^{(i)}(\mathbf{z}) - \mu_T^{(i)}(\mathbf{z})| + |\mu_T^{(i)}(\mathbf{z}) - \widetilde{\mu}_T^{(i)}(\mathbf{z})| \tag{161}$$

$$\leq |f^{(i)}(\mathbf{z}) - \mu_T^{(i)}(\mathbf{z})| + |\mu_T^{(i)}(\mathbf{z}) - f^{(i)}(\mathbf{z})| \tag{162}$$

$$\leq 2B\sigma_T^{(i)}(\mathbf{z}), \tag{163}$$

where we used (155) and the confidence bounds. By recursion, we also have $(\widetilde{\mu}_T^{(i)} \circ \cdots \circ \widetilde{\mu}_T^{(1)})(\mathbf{x}) \in \mathcal{X}^{(i+1)}$ for all $\mathbf{x} \in \mathcal{X}$, and therefore

$$(\sigma_T^{(i+1)} \circ \widetilde{\mu}_T^{(i)} \circ \cdots \circ \widetilde{\mu}_T^{(1)})(\mathbf{x}) \leq \bar{\sigma}_T^{(i+1)}. \tag{164}$$

Hence, replacing $\mu$ with $\widetilde{\mu}$ in (125)–(135), it follows from Corollary 1 that

$$r_T^* \leq 4B \sum_{i=1}^m L^{m-i} \bar{\sigma}_T^{(i)} = \begin{cases} O(BL^{m-1}T^{-\nu/d}) & \text{when } \nu \leq 1, \\ O(BL^{(m-1)\nu}T^{-\nu/d}) & \text{when } \nu > 1. \end{cases} \tag{165}$$

### G.3 Non-Adaptive Sampling Method for Multi-Output Chains

For multi-output chains, the composite posterior mean of $g(\mathbf{x})$ is

$$\mu_T^g(\mathbf{x}) = (\mu_T^{(m)} \circ \mu_T^{(m-1)} \circ \cdots \circ \mu_T^{(1)})(\mathbf{x}), \tag{166}$$

where $\mu_T^{(i)}$ denotes the posterior mean of $f^{(i)}$ computed using (3) based on $\{(\mathbf{x}_s^{(i)}, \mathbf{x}_s^{(i+1)})\}_{s=1}^T$ for each $i \in [m]$.

We again assume that each $\mathcal{X}^{(i)}$ is a hyperrectangle of dimension $d_i$. Then, the upper bound on simple regret of Algorithm 2 using (166) is provided in the following theorem.

**Theorem 7** (Non-adaptive sampling method for multi-output chains). *Under the setup of Section 2, given* $B = \Theta(L)$, $k = k_{Matérn}$, $\Gamma(\cdot, \cdot) = k(\cdot, \cdot)\mathbf{I}$, *and a multi-output chain* $g = f^{(m)} \circ f^{(m-1)} \circ \cdots \circ f^{(1)}$ *with* $f^{(i)} \in \mathcal{H}_\Gamma(B) \cap \mathcal{F}(L)$ *and* $\mathcal{X}^{(i)}$ *being a hyperrectangle for each* $i \in [m]$,

- *when* $\nu \leq 1$, *Algorithm 2 achieves*

$$r_T^* = \widetilde{O}(BL^{m-1}T^{-\nu^m/d});$$

- *when* $\nu > 1$, *Algorithm 2 achieves*

$$r_T^* = \widetilde{O}\big(\max\big\{BL^{(m-1)\nu}T^{-\nu/d}, B^{1+\nu+\nu^2+\cdots+\nu^{m-2}}L^{\nu^{m-1}}T^{-\nu^2/d}\big\}\big).$$

*Proof.* The analysis is similar to the case of chains, so we omit some details and focus on the main differences. Defining $\widetilde{\Gamma}_T^{(i)}(\mathbf{x}) = \|\Gamma_T^{(i)} \circ \mu_T^{(i-1)} \circ \cdots \circ \mu_T^{(1)}(\mathbf{x})\|_2^{1/2}$, similarly to the case of chains, we have

$$|g(\mathbf{x}) - (\mu_T^{(m)} \circ \mu_T^{(m-1)} \circ \cdots \circ \mu_T^{(1)})(\mathbf{x})| \leq B \sum_{i=1}^m L^{m-i} \widetilde{\Gamma}_T^{(i)}(\mathbf{x}), \tag{167}$$

and

$$r_T^* = g(\mathbf{x}^*) - g(\mathbf{x}_T^*) \leq 2B \sum_{i=1}^m L^{m-i} \max\{\widetilde{\Gamma}_T^{(i)}(\mathbf{x}^*), \widetilde{\Gamma}_T^{(i)}(\mathbf{x}_T^*)\}. \tag{168}$$

For each $\mathbf{x}' \in \{\mathbf{x}^*, \mathbf{x}_T^*\}$, we extend $\mathcal{X}^{(i+1)}$ to the smallest hyperrectangle $\widetilde{\mathcal{X}}^{(i+1)}$ that covers $(\mu_T^{(i)} \circ \cdots \circ \mu^{(1)})(\mathbf{x}')$, and the original domain $\mathcal{X}^{(i+1)}$ (see Figure 5). Recalling the definition of $\delta_T^{(i+1)}$ in (101), with $\delta_{i+1}(\mathbf{x}) = \min_{\mathbf{z} \in \mathcal{X}^{(i+1)}} \|(\mu_T^{(i)} \circ \cdots \circ \mu^{(1)})(\mathbf{x}) - \mathbf{z}\|_2$, the fill distance of $\widetilde{\mathcal{X}}^{(i+1)}$ with respect to $\{\mathbf{x}_s^{(i+1)}\}_{s=1}^T$ is

$$\delta_{i+1}' = \max_{\mathbf{z}' \in \widetilde{\mathcal{X}}^{(i+1)}} \min_{s \in [T]} \|\mathbf{z}' - \mathbf{x}_s^{(i+1)}\|_2 \tag{169}$$

$$\leq \delta_T^{(i+1)} + \max_{\mathbf{z}' \in \widetilde{\mathcal{X}}^{(i+1)}} \min_{\mathbf{z} \in \mathcal{X}^{(i+1)}} \|\mathbf{z}' - \mathbf{z}\|_2 \tag{170}$$

$$\leq \delta_T^{(i+1)} + \delta_{i+1}(\mathbf{x}') \tag{171}$$

$$\leq \delta_T^{(i+1)} + B \sum_{j=1}^i L^{i-j} \widetilde{\Gamma}_T^{(i)}(\mathbf{x}'). \tag{172}$$

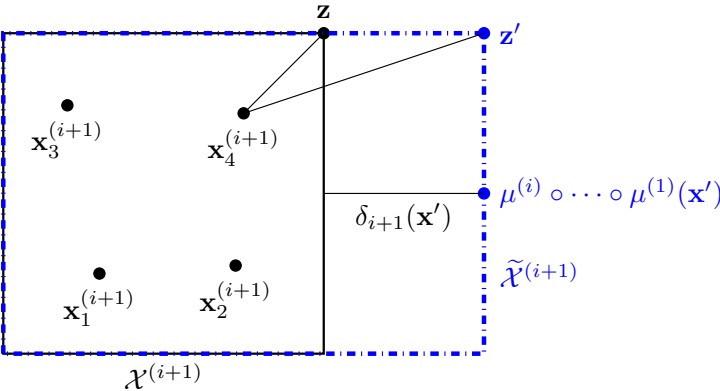

Figure 5: Extended domain for multi-output chains.

Since $\widetilde{\mathcal{X}}^{(i+1)}$ is a hyperrectangle and $\mu_T^{(i)} \circ \cdots \circ \mu^{(1)}(\mathbf{x}') \in \widetilde{\mathcal{X}}^{(i+1)}$, it follows from (112) that

$$\widetilde{\Gamma}_T^{(i+1)}(\mathbf{x}') = O\big((\delta_{i+1}')^\nu\big) \tag{173}$$

$$= O\left( \left( \max\left\{ L^i T^{-1/d}, iB \cdot \max_{j \in [i]} L^{i-j} \widetilde{\Gamma}_T^{(j)}(\mathbf{x}') \right\} \right)^\nu \right). \tag{174}$$

By Lemma 4, we also have

$$\widetilde{\Gamma}_T^{(1)}(\mathbf{x}') = \|\Gamma_T^{(1)}(\mathbf{x}')\|_2^{1/2} = \sigma_T^{(1)}(\mathbf{x}') = O\big((\delta_T)^\nu\big) = O(T^{-\nu/d}). \tag{175}$$

This recursive relation is exactly the same as that of chains in (146), showing Theorem 4 extends to multi-output chains.

For the two more restrictive cases, Corollary 2, with the same posterior standard deviation upper bound as Corollary 1, implies that the results in Appendix G.2 also extend to multi-output chains.

$\square$

## G.4 Non-Adaptive Sampling Method for Feed-Forward Networks

For feed-forward networks of scalar-valued functions, let $\mu_T^{(i,j)}$ denote the posterior mean of $f^{(i,j)}$ computed using (1) based on $\{(\mathbf{x}_s^{(i)}, \mathbf{x}_s^{(i+1,j)})\}_{s=1}^T$. The composite posterior mean of $g(\mathbf{x})$ with feed-forward network structure is

$$\mu_T^g(\mathbf{x}) = \mu_T^{(m,1)}(\mathbf{z}^{(m)}) \tag{176}$$

with

$$\mathbf{z}^{(1)} = \mathbf{x}, \tag{177}$$

$$\mathbf{z}^{(i+1,j)} = \mu_T^{(i,j)}(\mathbf{z}^{(i)}) \qquad\qquad \text{for } i \in [m-1], j \in [d_{i+1}], \tag{178}$$

$$\mathbf{z}^{(i+1)} = \mu_T^{(i)}(\mathbf{z}^{(i)}) = [\mathbf{z}^{(i+1,1)}, \ldots, \mathbf{z}^{(i+1,d_i)}] \qquad\qquad \text{for } i \in [m-1]. \tag{179}$$

Assuming each $\mathcal{X}^{(i)}$ is a hyperrectangle of dimension $d_i$, the following theorem provides the upper bound on simple regret of Algorithm 2 using (176) for feed-forward networks.

**Theorem 8** (Non-adaptive sampling method for feed-forward networks)**.** *Under the setup of Section 2, given $B = \Theta(L)$, $k = k_{Matérn}$, and a feed-forward network $g = f^{(m)} \circ f^{(m-1)} \circ \cdots \circ f^{(1)}$ with $f^{(i)}(\mathbf{z}) = [f^{(i,j)}(\mathbf{z})]_{j=1}^{d_{i+1}}$, $f^{(i,j)} \in \mathcal{H}_k(B) \cap \mathcal{F}(L)$, and $\mathcal{X}^{(i)}$ being a hyperrectangle for each $i \in [m], j \in [d_{i+1}]$,*

- *when $\nu \le 1$, Algorithm 2 achieves*

$$r_T^* = \widetilde{O}(\sqrt{D_{2,m}} B L^{m-1} T^{-\nu^m/d}); \tag{180}$$

- *when $\nu > 1$, Algorithm 2 achieves*

$$r_T^* = \widetilde{O}\Big( \max\Big\{ (D_{2,m})^{\nu/2} B L^{(m-1)\nu} T^{-\nu/d}, \widetilde{D}_{2,m}^{\nu} B^{1+\nu+\nu^2+\cdots+\nu^{m-2}} L^{\nu^{m-1}} T^{-\nu^2/d} \Big\} \Big), \tag{181}$$

*where $D_{2,m} = \prod_{i=2}^m d_i$ and $\widetilde{D}_{2,m}^{\nu} = \prod_{i=2}^m (d_i)^{\nu^{m+1-i}/2}$.*

*Proof.* With $\mu_T^{(i)}(\cdot) = [\mu_T^{(i,j)}(\cdot)]_{j=1}^{d_{i+1}}$, we have

$$\|f^{(i)}(\mathbf{z}) - \mu_T^{(i)}(\mathbf{z})\|_2 = \sqrt{\sum_{j=1}^{d_{i+1}} \left( f^{(i,j)}(\mathbf{z}) - \mu_T^{(i,j)}(\mathbf{z}) \right)^2} \tag{182}$$

$$\le \sqrt{\sum_{j=1}^{d_{i+1}} \left( B \sigma_T^{(i,j)}(\mathbf{z}) \right)^2} \tag{183}$$

$$= \sqrt{d_{i+1}} B \sigma_T^{(i,1)}(\mathbf{z}), \tag{184}$$

where we use the confidence bounds and the fact that $\sigma_T^{(i,j)}(\mathbf{z}) = \sigma_T^{(i,1)}(\mathbf{z})$ by the symmetry of our setup.

Since each $f^{(i,j)}$ is Lipschitz continuous with parameter $L$, we have that $f^{(i)}$ is Lipschitz continuous with parameter $\sqrt{d_{i+1}}L$, and $f^{(m)} \circ \cdots \circ f^{(i)}$ is Lipschitz continuous with parameter $\sqrt{D_{i+1,m+1}} L^{m-i+1}$, where $D_{i,j} = \prod_{s=i}^j d_s$ and $d_{m+1} = 1$. Then, defining $\widetilde{\sigma}_T^{(i,1)}(\mathbf{x}) = (\sigma_T^{(i,1)} \circ \mu_T^{(i-1)} \circ \cdots \circ \mu_T^{(1)})(\mathbf{x})$, we can follow (125)–(131) to obtain

$$g(\mathbf{x}) = (f^{(m)} \circ f^{(m-1)} \circ \cdots \circ f^{(2)})\big(f^{(1)}(\mathbf{x})\big) \tag{185}$$

$$\le (f^{(m)} \circ f^{(m-1)} \circ \cdots \circ f^{(2)})\big(\mu_T^{(1)}(\mathbf{x})\big) + \sqrt{D_{3,m+1}} L^{m-1} \|f^{(1)}(\mathbf{x}) - \mu_T^{(1)}(\mathbf{x})\|_2 \tag{186}$$

$$= (f^{(m)} \circ f^{(m-1)} \circ \cdots \circ f^{(2)})\big(\mu_T^{(1)}(\mathbf{x})\big) + \sqrt{D_{2,m+1}} L^{m-1} B \sigma_T^{(1,1)}(\mathbf{x}) \tag{187}$$

$$\le (f^{(m)} \circ f^{(m-1)} \circ \cdots \circ f^{(3)})\big((\mu_T^{(2)} \circ \mu_T^{(1)})(\mathbf{x})\big) + \sqrt{D_{3,m+1}} L^{m-2} B \sigma_T^{(2,1)}\big(\mu_T^{(1)}(\mathbf{x})\big)$$
$$+ \sqrt{D_{2,m+1}} L^{m-1} B \sigma_T^{(1,1)}(\mathbf{x}) \tag{188}$$

$$\le (\mu_T^{(m)} \circ \mu_T^{(m-1)} \circ \cdots \circ \mu_T^{(1)})(\mathbf{x}) + B \sigma^{(m,1)}\big((\mu_T^{(m-1)} \circ \cdots \circ \mu_T^{(1)})(\mathbf{x})\big)$$
$$+ L B \sigma^{(m-1)}\big((\mu_T^{(m-2)} \circ \cdots \circ \mu_T^{(1)})(\mathbf{x})\big) + \cdots + \sqrt{D_{2,m+1}} L^{m-1} B \sigma_T^{(1,1)}(\mathbf{x}) \tag{189}$$

$$\le (\mu_T^{(m)} \circ \mu_T^{(m-1)} \circ \cdots \circ \mu_T^{(1)})(\mathbf{x}) + B \sum_{i=1}^m \sqrt{D_{i+1,m+1}} L^{m-i} \widetilde{\sigma}_T^{(i,1)}(\mathbf{x}). \tag{190}$$

Hence, we have

$$r_T^* = g(\mathbf{x}^*) - g(\mathbf{x}_T^*) \le 2B \sum_{i=1}^m \sqrt{D_{i+1,m}} L^{m-i} \max\{\widetilde{\sigma}_T^{(i,1)}(\mathbf{x}^*), \widetilde{\sigma}_T^{(i,1)}(\mathbf{x}_T^*)\}. \tag{191}$$

It remains to upper bound $\max\{\widetilde{\sigma}_T^{(i,1)}(\mathbf{x}^*), \widetilde{\sigma}_T^{(i,1)}(\mathbf{x}_T^*)\}$ for $i \in [m]$. Firstly, by Lemma 5, we have

$$\widetilde{\sigma}_T^{(1,1)}(\mathbf{x}) = \sigma_T^{(1,1)}(\mathbf{x}) = O\big((\delta_T)^{\nu}\big) = O(T^{-\nu/d}). \tag{192}$$

We now represent the upper bound on $\widetilde{\sigma}_T^{(i+1,1)}(\cdot)$ in terms of $\{\widetilde{\sigma}_T^{(j,1)}(\cdot) : j \in [i]\}$ for $i \in [m-1]$. For all $\mathbf{x} \in \mathcal{X}$, the distance between $\mu_T^{(i)} \circ \cdots \circ \mu_T^{(1)}(\mathbf{x})$ and its closest point in $\mathcal{X}^{(i+1)}$ is

$$\delta_{i+1}(\mathbf{x}) = \min_{\mathbf{z} \in \mathcal{X}^{(i+1)}} |(\mu_T^{(i)} \circ \cdots \circ \mu_T^{(1)})(\mathbf{x}) - \mathbf{z}| \tag{193}$$

$$\leq |(\mu_T^{(i)} \circ \cdots \circ \mu_T^{(1)})(\mathbf{x}) - f^{(i)} \circ \cdots \circ f^{(1)}(\mathbf{x})| \tag{194}$$

$$\leq B \sum_{j=1}^{i} \sqrt{D_{j+1,i+1}} L^{i-j} \widetilde{\sigma}_T^{(j,1)}(\mathbf{x}), \tag{195}$$

where the last step follows from (190). Then, the distance between $(\mu_T^{(i)} \circ \cdots \circ \mu_T^{(1)})(\mathbf{x})$ and its closest point in $\mathcal{X}_T^{(i+1)}$ is

$$\delta_{i+1}'(\mathbf{x}) = \min_{s \in [T]} |(\mu_T^{(i)} \circ \cdots \circ \mu_T^{(1)})(\mathbf{x}) - \mathbf{x}_s^{(i+1)}| \tag{196}$$

$$\leq \delta_T^{(i+1)} + \delta_{i+1}(\mathbf{x}) \tag{197}$$

$$\leq O(\sqrt{D_{2,i+1}} L^i T^{-1/d}) + B \sum_{j=1}^{i} \sqrt{D_{j+1,i+1}} L^{i-j} \widetilde{\sigma}_T^{(j,1)}(\mathbf{x}) \tag{198}$$

$$= O\left( \max\left\{ \sqrt{D_{2,i+1}} L^i T^{-1/d}, iB \cdot \max_{j \in [i]} \sqrt{D_{j+1,i+1}} L^{i-j} \widetilde{\sigma}_T^{(j,1)}(\mathbf{x}) \right\} \right), \tag{199}$$

where (197) applies the triangle inequality, and (198) uses (122) and (195).

Now, for each $\mathbf{x}' \in \{\mathbf{x}^*, \mathbf{x}_T^*\}$, we extend $\mathcal{X}^{(i+1)}$ to $\widetilde{\mathcal{X}}^{(i+1)}$, the smallest hyperrectangle that covers $(\mu_T^{(i)} \circ \cdots \circ \mu_T^{(1)})(\mathbf{x}')$ and $\mathcal{X}^{(i+1)}$. Then, the fill distance of the extended domain with regard to $\mathcal{X}_T^{(i+1)}$ is

$$\delta_{i+1}' = O\left( \max\left\{ \sqrt{D_{2,i+1}} L^i T^{-1/d}, iB \cdot \max_{j \in [i]} \sqrt{D_{j+1,i+1}} L^{i-j} \widetilde{\sigma}_T^{(j,1)}(\mathbf{x}') \right\} \right), \tag{200}$$

and by Lemma 5, we have

$$\widetilde{\sigma}_T^{(i+1,1)}(\mathbf{x}') = O((\delta_{i+1}')^\nu) = O\left( \left( \max\left\{ \sqrt{D_{2,i+1}} L^i T^{-1/d}, iB \cdot \max_{j \in [i]} \sqrt{D_{j+1,i+1}} L^{i-j} \widetilde{\sigma}_T^{(j,1)}(\mathbf{x}') \right\} \right)^\nu \right). \tag{201}$$

We now consider two cases;

The case that $\nu \leq 1$: Recalling the assumption $B = \Theta(L)$, and using the fact that $\widetilde{O}(L^{c\nu})$ is equivalent to $O((L^{c\nu} \mathrm{poly}(c \log L)))$ for fixed $\nu$, it follows from (192) and (201) that when $\nu \leq 1$, we have

$$\widetilde{\sigma}_T^{(2,1)}(\mathbf{x}') = O\left( \max\left\{ (d_2)^{\nu/2} L^\nu T^{-\nu/d}, (d_2)^{\nu/2} B^\nu T^{-\nu^2/d} \right\} \right) = O((d_2)^{\nu/2} L^\nu T^{-\nu^2/d}), \tag{202}$$

$$\widetilde{\sigma}_T^{(3,1)}(\mathbf{x}') = O\left( \max\left\{ (D_{2,3})^{\nu/2} L^{2\nu} T^{-\nu/d}, 2^\nu (D_{2,3})^{\nu/2} B^\nu L^\nu T^{-\nu^2/d}, 2^\nu (d_3)^{\nu/2} B^\nu (d_2)^{\nu^2/2} L^{\nu^2} T^{-\nu^3/d} \right\} \right)$$

$$= O(2^\nu (D_{2,3})^{\nu/2} B^\nu L^\nu T^{-\nu^3/d}) = \widetilde{O}((D_{2,3})^{\nu/2} B^\nu L^\nu T^{-\nu^3/d}), \tag{203}$$

$$\vdots$$

$$\widetilde{\sigma}_T^{(i,1)}(\mathbf{x}') = \widetilde{O}((i-1)^\nu (D_{2,i})^{\nu/2} B^\nu L^{(i-2)\nu} T^{-\nu^i/d}) = \widetilde{O}((D_{2,i})^{\nu/2} B^\nu L^{(i-2)\nu} T^{-\nu^i/d}), \tag{204}$$

and the simple regret is

$$r_T^* \leq 2B \sum_{i=1}^{m} \sqrt{D_{i+1,m}} L^{m-i} \max\{\widetilde{\sigma}_T^{(i,1)}(\mathbf{x}^*), \widetilde{\sigma}_T^{(i,1)}(\mathbf{x}_T^*)\} \tag{205}$$

$$= \widetilde{O}(\sqrt{D_{2,m}} B L^{m-1} T^{-\nu^m/d}). \tag{206}$$

The case that $\nu > 1$: With $B = \Theta(L)$, and using the property of $\widetilde{O}(\cdot)$ similar to the first case, it follows from (192) and (201) that when $\nu > 1$, we have

$$\widetilde{\sigma}_T^{(2,1)}(\mathbf{x}') = O\Big(\max\Big\{(d_2)^{\nu/2}L^\nu T^{-\nu/d}, (d_2)^{\nu/2}B^\nu T^{-\nu^2/d}\Big\}\Big) = O((d_2)^{\nu/2}L^\nu T^{-\nu/d}), \tag{207}$$

$$\widetilde{\sigma}_T^{(3,1)}(\mathbf{x}') = O\Big(\max\Big\{(D_{2,3})^{\nu/2}L^{2\nu}T^{-\nu/d}, (d_3)^{\nu/2}B^\nu(d_2)^{\nu^2/2}L^{\nu^2}T^{-\nu^2/d}\Big\}\Big)$$
$$= O\big(2^\nu(d_3)^{\nu/2}(d_2)^{\nu^2/2}B^\nu L^{\nu^2}T^{-\nu/d}\big) = \widetilde{O}\big((d_3)^{\nu/2}(d_2)^{\nu^2/2}B^\nu L^{\nu^2}T^{-\nu/d}\big), \tag{208}$$

$$\vdots$$

$$\widetilde{\sigma}_T^{(i,1)}(\mathbf{x}') = \widetilde{O}\Big(\max\Big\{(D_{2,i})^{\nu/2}L^{(i-1)\nu}T^{-\nu/d}, (i-1)^\nu(d_i)^{\nu/2}(d_{i-1})^{\nu^2/2}\cdots(d_2)^{\nu^{i-1}/2}B^{\nu+\cdots+\nu^{i-2}}L^{\nu^{i-1}}T^{-\nu^2/d}\Big\}\Big)$$
$$= \widetilde{O}\Big(\max\Big\{(D_{2,i})^{\nu/2}L^{(i-1)\nu}T^{-\nu/d}, (d_i)^{\nu/2}(d_{i-1})^{\nu^2/2}\cdots(d_2)^{\nu^{i-1}/2}B^{\nu+\cdots+\nu^{i-2}}L^{\nu^{i-1}}T^{-\nu^2/d}\Big\}\Big), \tag{209}$$

and the simple regret is

$$r_T^* \le 2B\sum_{i=1}^m \sqrt{D_{i+1,m}}L^{m-i}\max\{\widetilde{\sigma}_T^{(i,1)}(\mathbf{x}^*), \widetilde{\sigma}_T^{(i,1)}(\mathbf{x}_T^*)\} \tag{210}$$

$$= \widetilde{O}\Big(\max\Big\{(D_{2,m})^{\nu/2}BL^{(m-1)\nu}T^{-\nu/d}, \widetilde{D}_{2,m}^\nu B^{1+\nu+\nu^2+\cdots+\nu^{m-2}}L^{\nu^{m-1}}T^{-\nu^2/d}\Big\}\Big), \tag{211}$$

where $\widetilde{D}_{2,m}^\nu = \prod_{i=2}^m (d_i)^{\nu^{m+1-i}/2}$. $\qquad\square$

Next, we recall the following two restrictive cases introduced in the main body:

- **Case 1**: We additionally assume that $(\mu_T^{(i)} \circ \cdots \circ \mu_T^{(1)})(\mathbf{x}^*) \in \mathcal{X}^{(i+1)}$ and $(\mu_T^{(i)} \circ \cdots \circ \mu_T^{(1)})(\mathbf{x}_T^*) \in \mathcal{X}^{(i+1)}$ for $i \in [m-1]$.

- **Case 2**: We additionally assume that all the $\mathcal{X}^{(i)}$'s are known. Defining

$$\widetilde{\mu}_T^{(i)}(\mathbf{z}) = \underset{\mathbf{z}' \in \mathcal{X}^{(i+1)}}{\arg\min} |\mu_T^{(i)}(\mathbf{z}) - \mathbf{z}'|, \tag{212}$$

  we let the algorithm return

$$\mathbf{x}_T^* = \underset{\mathbf{x} \in \mathcal{X}}{\arg\max}(\widetilde{\mu}_T^{(m)} \circ \cdots \circ \widetilde{\mu}_T^{(1)})(\mathbf{x}). \tag{213}$$

Similarly to chains and multi-output chains, in either case, it follows that

$$r_T^* \le 2B\sum_{i=1}^m \sqrt{D_{i+1,m}}L^{m-i}\max\{\widetilde{\sigma}_T^{(i,1)}(\mathbf{x}^*), \widetilde{\sigma}_T^{(i,1)}(\mathbf{x}_T^*)\} = O\Big(B\sum_{i=1}^m \sqrt{D_{i+1,m}}L^{m-i}\overline{\sigma}_T^{(i,1)}\Big). \tag{214}$$

Moreover, by Corollary 3, we can further upper bound this by

$$r_T^* \le O\Big(B\sum_{i=1}^m \sqrt{D_{i+1,m}}L^{m-i}\overline{\sigma}_T^{(i,1)}\Big) = \begin{cases} O(\sqrt{D_{2,m}}BL^{m-1}T^{-\nu/d}) & \text{when } \nu \le 1, \\ O\big((D_{2,m})^{\nu/2}BL^{(m-1)\nu}T^{-\nu/d}\big) & \text{when } \nu > 1. \end{cases} \tag{215}$$

## H   Lower Bound on Simple Regret (Proof of Theorem 5)

Recall the high-level intuition behind our lower bound outlined in Section 5: We use the idea from (Bull, 2011) of having a small bump that is difficult to locate, but unlike (Bull, 2011), we exploit Lipschitz functions at the intermediate layers to "amplify" that bump (which means the original bump can have a much narrower width for a given RKHS norm).

We first show that, for fixed $\epsilon > 0$, we can construct a base function $\bar{g}$ with height $2\epsilon$ and support radius $w = \Theta\big((\frac{\epsilon}{BL^{m-1}})^{1/\nu}\big)$ for each network structure.

### H.1 Hard Function for Chains

To construct a base function with chain structure, we define the following two scalar-valued functions.

Considering the "bump" function $h(\mathbf{x}) = \exp\left(\frac{-1}{1-\|\mathbf{x}\|_2^2}\right)\mathbb{1}\{\|\mathbf{x}\|_2 < 1\}$, for some $\epsilon_1 > 0$ and width $w > 0$, we define (Bull, 2011)

$$\widetilde{h}(\mathbf{x}, \epsilon_1, w) = \frac{2\epsilon_1}{h(\mathbf{0})} h\left(\frac{\mathbf{x}}{w}\right), \tag{216}$$

which is a scaled bump function with height $2\epsilon_1$ and compact support $\{\mathbf{x} \in \mathcal{X} : \|\mathbf{x}\|_2 < w\}$.

For a fixed $u > 0$ and $\widetilde{L} = \sqrt{2}B/\sqrt{k(0) - k(2u)}$, where $k$ is the Matérn kernel on $\mathbb{R}$ with smoothness $\nu$ and $k(|x - x'|) = k(x, x')$, we define

$$\widetilde{g}_k(\cdot) = \frac{\widetilde{L}}{2}\left(k(\cdot, u) - k(\cdot, -u)\right). \tag{217}$$

**Theorem 9** (Hard function for chains). *For the Matérn kernel $k$ with smoothness $\nu \geq 1$, sufficiently small $\epsilon > 0$, and sufficiently large $B$, there exists $\epsilon_1 > 0, L = \Theta(B), c = \Theta(1), w = \Theta\left(\left(\frac{\epsilon}{B(cL)^{m-1}}\right)^{1/\nu}\right)$, and $u > 0$ such that $\bar{g} := \bar{f}^{(m)} \circ \bar{f}^{(m-1)} \circ \cdots \circ \bar{f}^{(1)}$ with $\bar{f}^{(1)}(\cdot) = \widetilde{h}(\cdot, \epsilon_1, w)$ and $\bar{f}^{(s)}(\cdot) = \widetilde{g}_k(\cdot)$ for $s \in [2, m]$ has the following properties:*

- $\bar{f}^{(i)} \in \mathcal{H}_k(B) \cap \mathcal{F}(L)$ *for each* $i \in [m]$,
- $\max_{\mathbf{x}} \bar{g}(\mathbf{x}) = 2\epsilon$,
- $\bar{g}(\mathbf{x}) > 0$ *when* $\|\mathbf{x}\|_2 < w$ *, and* $\bar{g}(\mathbf{x}) = 0$ *otherwise.*

*Proof.* For $k$ being the Matérn kernel with smoothness $\nu$, recalling that

$$\widetilde{h}(\mathbf{x}, \epsilon_1, w) = \frac{2\epsilon_1}{h(\mathbf{0})} h\left(\frac{\mathbf{x}}{w}\right) \tag{218}$$

with $h$ being the bump function, (Bull, 2011, Section A.2) has shown that for some constant $C_1$,

$$\|\widetilde{h}\|_k \leq C_1 \frac{2\epsilon_1}{h(\mathbf{0})}\left(\frac{1}{w}\right)^\nu \|h\|_k. \tag{219}$$

Hence, we have $\|\widetilde{h}\|_k \leq B$ when $w = \left(\frac{2C_1\|h\|_k\epsilon_1}{h(\mathbf{0})B}\right)^{1/\nu}$. When $\frac{\epsilon_1}{B}$ is sufficiently small, the diameter of the support satisfies $2w \ll 1$. Since the bump function is infinitely differentiable, $\widetilde{h}$ is Lipschitz continuous with some constant $L' = \Theta(\frac{\epsilon_1}{w})$, and our assumption of $\nu \geq 1$ further implies $L' = O(B)$.

Recall that for a fixed $u > 0$ and the Matérn kernel $k$ on a one-dimension domain, we define

$$\widetilde{g}_k(\cdot) = \frac{\widetilde{L}}{2}\left(k(\cdot, u) - k(\cdot, -u)\right) \tag{220}$$

with $\widetilde{L} = \frac{\sqrt{2}B}{\sqrt{k(0) - k(2u)}}$. Applying (23) with $\widetilde{g}_k$ replacing both $f$ and $f'$, we have

$$\|\widetilde{g}_k\|_k = \frac{\widetilde{L}}{2}\sqrt{k(u, u) + 2k(u, -u) + k(-u, -u)} = \frac{\widetilde{L}}{2}\sqrt{2k(0) - 2k(2u)} = B, \tag{221}$$

where $k(d_{\mathbf{x},\mathbf{x}'}) = k(\mathbf{x}, \mathbf{x}')$ with $d_{\mathbf{x},\mathbf{x}'} = \|\mathbf{x} - \mathbf{x}'\|_2$.

We choose the two constants $u > \widetilde{u} > 0$ to satisfy the following:

1. $k(d_{\mathbf{x},\mathbf{x}'})$ is non-increasing when $d_{\mathbf{x},\mathbf{x}'} \in [u - \widetilde{u}, u + \widetilde{u}]$.

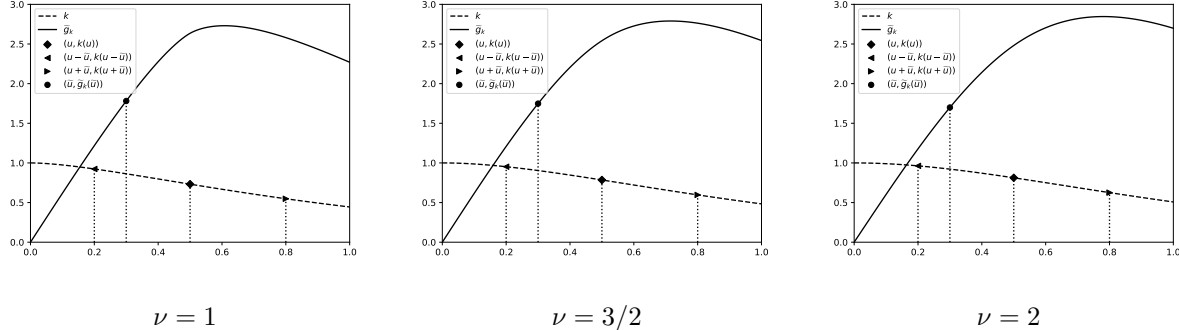

$$\nu = 1 \qquad\qquad \nu = 3/2 \qquad\qquad \nu = 2$$

Figure 6: $k = k_{\text{Matérn}}$ with lengthscale $l = 1$ and smoothness $\nu$, and $\widetilde{g}_k$ with $B = 5$, $u = 0.5$, and $\widetilde{u} = 0.3$.

2. $k(u - \widetilde{u}) - k(u + \widetilde{u}) \geq \frac{2}{L} = \frac{\sqrt{2}\sqrt{k(0) - k(2u)}}{B}$.

3. Defining $r_{\max} = \sup_{z \in (0, \widetilde{u}]} \frac{k(u-z) - k(u+z)}{2z}$ and $r_{\min} = \inf_{z \in (0, \widetilde{u}]} \frac{k(u-z) - k(u+z)}{2z}$, it holds that $r_{\max} = \Theta(1)$ and $r_{\min} = \Theta(1)$.

Whenever $k(d_{\mathbf{x}, \mathbf{x}'})$ is continuous and non-increasing in $d_{\mathbf{x}, \mathbf{x}'}$ on $\mathbb{R}^+$ (e.g., the Matérn kernel), condition 1 is satisfied. This condition guarantees that $k(\cdot, u)$ is non-decreasing on $[0, \widetilde{u}]$, and $k(\cdot, -u)$ is non-increasing on $[0, \widetilde{u}]$, which together implies that $\widetilde{g}_k$ is non-decreasing on $[0, \widetilde{u}]$. Under condition 1, condition 2 is satisfied as long as $B$ is sufficiently large, and it implies that $\widetilde{g}_k(\widetilde{u}) \geq 1$. Examples of $\widetilde{g}_k$ for $k = k_{\text{Matérn}}$ are given in Figure 6.

Let $L = \max\{L', \sup_{z \in (0, \widetilde{u}]} \frac{\widetilde{g}_k(z)}{z}\} = \max\{L', r_{\max} \widetilde{L}\}$ and $\alpha = \inf_{z \in (0, \widetilde{u}]} \frac{\widetilde{g}_k(z)}{zL} = \frac{r_{\min} \widetilde{L}}{L}$. Then, it holds for all $z \in (0, \widetilde{u}]$ that

$$1 < \alpha L \leq \frac{\widetilde{g}_k(z)}{z} \leq L. \tag{222}$$

Clearly $\alpha$ cannot exceed 1, and moreover, condition 3 above along with $L' = O(B)$ implies that $\alpha = \min\left\{r_{\min}\sqrt{\frac{2}{k(0) - k(2u)}} \frac{B}{L'}, \frac{r_{\min}}{r_{\max}}\right\} = \Theta(1)$.

Defining $\bar{g}_i = \overline{f}^{(i)} \circ \overline{f}^{(i-1)} \circ \cdots \circ \overline{f}^{(1)}$, we denote the height of $\bar{g}_i$ by $h_i = \max_{\mathbf{x}} \bar{g}_i(\mathbf{x})$. If $h_{i-1} \leq \widetilde{u}$, (222) implies

$$h_{i-1} < \alpha L h_{i-1} \leq h_i \leq L h_{i-1}. \tag{223}$$

Then, given $\epsilon \in (0, \widetilde{g}_k(\widetilde{u})/2]$ (recalling that we assume $\epsilon \in (0, 1/2]$ and noting that condition 2 implies $\widetilde{g}_k(\widetilde{u}) \geq 1$), we choose $\epsilon_1$ to satisfy

$$(\overline{f}^{(m)} \circ \cdots \circ \overline{f}^{(2)})(2\epsilon_1) = (\widetilde{g}_k \circ \cdots \circ \widetilde{g}_k)(2\epsilon_1) = 2\epsilon, \tag{224}$$

By $h_1 = 2\epsilon_1$ and (223), this choice of $\epsilon_1$ must also satisfy

$$2(\alpha L)^{m-1}\epsilon_1 = (\alpha L)^{m-1} h_1 \leq h_m = 2\epsilon \leq L^{m-1} h_1 = 2L^{m-1}\epsilon_1, \tag{225}$$

implying (via (222)) that

$$\frac{\epsilon}{L^{m-1}} \leq \epsilon_1 \leq \frac{\epsilon}{(\alpha L)^{m-1}} < \epsilon. \tag{226}$$

Since $\widetilde{h}$ has a compact support with radius $w$ and $\widetilde{g}(0) = 0$, $\bar{g}$ also has a compact support with radius

$$w = \Theta\left(\left(\frac{\epsilon_1}{B}\right)^{1/\nu}\right) = \Theta\left(\left(\frac{\epsilon}{B(cL)^{m-1}}\right)^{1/\nu}\right), \tag{227}$$

for some constant $c = \Theta(1)$.

Lastly, (222) implies $\widetilde{g}_k$ on $[0, \widetilde{u}]$ is a member of $\mathcal{F}(L)$, and $L \geq L'$ guarantees $\widetilde{h} \in \mathcal{F}(L)$. $\qquad\square$

## H.2 Hard Function for Multi-Output Chains

In this section, we consider the operator-valued Matérn kernel $\Gamma^{(i)}(\mathbf{x}, \mathbf{x}') = k^{(i)}(\mathbf{x}, \mathbf{x}')\mathbf{I}^{(i+1)}$, where $k^{(i)}$ is the scalar-valued Matérn kernel on $\mathbb{R}^{d_i}$ and $\mathbf{I}^{(i+1)}$ is the identity matrix of size $d_{i+1}$. Then, for a fixed $\mathbf{u}^{(i)} \in \mathbb{R}^{d_i}$ and $\widetilde{L}^{(i)} = \sqrt{2}B/\sqrt{k^{(i)}(0) - k^{(i)}(2\|\mathbf{u}^{(i)}\|_2)}$ where $k^{(i)}(\|\mathbf{x} - \mathbf{x}'\|_2) = k^{(i)}(\mathbf{x}, \mathbf{x}')$, we define

$$\widetilde{g}^{(i)}(\cdot) = \frac{\widetilde{L}^{(i)}}{2}\big(\Gamma^{(i)}(\cdot, \mathbf{u}^{(i)}) - \Gamma^{(i)}(\cdot, -\mathbf{u}^{(i)})\big). \tag{228}$$

**Theorem 10** (Hard function for multi-output chains). *Let $\mathbf{I}^{(i)}$ denote the identity matrix of size $d_i$, let $\mathbf{e}_1^{(i)}$ denote the first column of $\mathbf{I}^{(i)}$, and let $k^{(i)}$ denote the scalar-valued Matérn kernel on $\mathbb{R}^{d_i}$ with smoothness $\nu \geq 1$. For $\Gamma^{(i)}(\mathbf{x}, \mathbf{x}') = k^{(i)}(\mathbf{x}, \mathbf{x}')\mathbf{I}^{(i+1)}$, sufficiently small $\epsilon > 0$, and sufficiently large $B$, there exists $\epsilon_1 > 0, L = \Theta(B), c = \Theta(1), w = \Theta\big((\frac{\epsilon}{B(cL)^{m-1}})^{1/\nu}\big)$, and $u \in \mathbb{R}$ such that $\overline{g} := \overline{f}^{(m)} \circ \overline{f}^{(m-1)} \circ \cdots \circ \overline{f}^{(1)}$ with $\overline{f}^{(1)}(\cdot) = \widetilde{h}(\cdot, \epsilon_1, w)\mathbf{e}_1^{(2)}$ and $\overline{f}^{(s)}(\cdot) = \widetilde{g}^{(s)}(\cdot)\mathbf{e}_1^{(s+1)}$ with $\mathbf{u}^{(s)} = u\mathbf{e}_1^{(s)}$ for $s \in [2, m]$ has the following properties:*

- *$\overline{f}^{(i)} \in \mathcal{H}_{\Gamma^{(i)}}(B) \cap \mathcal{F}(L)$ for each $i \in [m]$,*

- *$\max_{\mathbf{x}} \overline{g}(\mathbf{x}) = 2\epsilon$,*

- *$\overline{g}(\mathbf{x}) > 0$ when $\|\mathbf{x}\|_2 < w$ , and $\overline{g}(\mathbf{x}) = 0$ otherwise.*

*Proof.* The rough idea is to reduce to the case of regular chains by only making use of a single coordinate throughout the network. We leave open the question as to whether the lower bound can be improved by utilizing all coordinates.

With $\mathbf{I}$ denoting the identity matrix of size $d_2$ and $\mathbf{e}_1$ denoting the first column of $\mathbf{I}$, we aim to show that $\overline{f}^{(1)} \in \mathcal{H}_{\Gamma^{(1)}}(B)$ by showing that if $\Gamma(\cdot, \cdot) = k(\cdot, \cdot)\mathbf{I} \in \mathbb{R}^{d_2 \times d_2}$ and $\widetilde{h} \in \mathcal{H}_k(B)$ for some scalar-valued kernel $k$ and some constant $B$, then the function $\widetilde{h}'(\cdot) = \widetilde{h}(\cdot)\mathbf{e}_1$ satisfies $\widetilde{h}' \in \mathcal{H}_\Gamma(B)$. Since $\widetilde{h} \in \mathcal{H}_k(B)$, there exists a sequence $\{(a_i, \mathbf{x}_i)\}_{i=1}^\infty$ such that $\widetilde{h}(\cdot) = \sum_{i=1}^\infty a_i k(\cdot, \mathbf{x}_i)$. Then, using the definition of RKHS norm for vector-valued functions in (24), we have

$$\|\widetilde{h}'\|_\Gamma^2 = \lim_{n\to\infty} \Big\| \sum_{i=1}^n a_i k(\cdot, \mathbf{x}_i)\mathbf{e}_1 \Big\|_\Gamma^2 \tag{229}$$

$$= \lim_{n\to\infty} \Big\| \sum_{i=1}^n a_i \Gamma(\cdot, \mathbf{x}_i)\mathbf{e}_1 \Big\|_\Gamma^2 \tag{230}$$

$$= \sum_{i,j=1}^\infty \langle \Gamma(\mathbf{x}_i, \mathbf{x}_j)(a_i\mathbf{e}_1), a_j\mathbf{e}_1 \rangle \tag{231}$$

$$= \sum_{i,j=1}^\infty a_i a_j k(\mathbf{x}_i, \mathbf{x}_j) \tag{232}$$

$$= \|\widetilde{h}\|_k^2 \tag{233}$$

$$\leq B^2, \tag{234}$$

and therefore $\widetilde{h}' \in \mathcal{H}_\Gamma(B)$. Next, for $i \in [2, m]$, with $\mathbf{u}^{(i)} = u\mathbf{e}_1^{(i)}$ we have

$$\|\overline{f}^{(i)}\|_\Gamma = \frac{\widetilde{L}^{(i)}}{2}\sqrt{\langle\Gamma^{(i)}(\mathbf{u}^{(i)}, \mathbf{u}^{(i)})\mathbf{e}_1, \mathbf{e}_1\rangle - 2\langle\Gamma^{(i)}(\mathbf{u}^{(i)}, -\mathbf{u}^{(i)})\mathbf{e}_1, \mathbf{e}_1\rangle + \langle\Gamma^{(i)}(-\mathbf{u}^{(i)}, -\mathbf{u}^{(i)})\mathbf{e}_1, \mathbf{e}_1\rangle} \tag{235}$$

$$= \frac{\widetilde{L}^{(i)}}{2}\sqrt{k^{(i)}(\mathbf{u}^{(i)}, \mathbf{u}^{(i)}) - 2k^{(i)}(\mathbf{u}^{(i)}, -\mathbf{u}^{(i)}) + k^{(i)}(-\mathbf{u}^{(i)}, -\mathbf{u}^{(i)})} \tag{236}$$

$$= \frac{\widetilde{L}^{(i)}}{2}\sqrt{2k^{(i)}(0) - 2k^{(i)}(2u)} \tag{237}$$

$$= B. \tag{238}$$

Since $\widetilde{h} \in \mathcal{F}(L')$ for some $L' > 1$, we also have $\widetilde{h}' \in \mathcal{F}(L')$.

We reuse the choice of $u$ and $\widetilde{u}$ in the previous case. Then, for any $z \in \mathbb{R}$, with $\mathbf{z}^{(i)} = z\mathbf{e}_1^{(i)}$ and $\mathbf{u}^{(i)} = u\mathbf{e}_1^{(i)}$, we have

$$\overline{f}^{(i)}(\mathbf{z}) = \widetilde{g}^{(i)}(\mathbf{z}^{(i)}, \mathbf{u}^{(i)})\mathbf{e}_1^{(i+1)} \tag{239}$$

$$= \frac{\widetilde{L}^{(i)}}{2}\big(\Gamma^{(i)}(\mathbf{z}^{(i)}, \mathbf{u}^{(i)}) - \Gamma^{(i)}(\mathbf{z}^{(i)}, -\mathbf{u}^{(i)})\big)\mathbf{e}_1^{(i+1)} \tag{240}$$

$$= \frac{\widetilde{L}^{(i)}}{2}\big(k^{(i)}(\mathbf{z}^{(i)}, \mathbf{u}^{(i)}) - k^{(i)}(\mathbf{z}^{(i)}, -\mathbf{u}^{(i)})\big)\mathbf{e}_1^{(i+1)} \tag{241}$$

$$= \frac{\widetilde{L}^{(i)}}{2}\big(k(\|\mathbf{z}^{(i)} - \mathbf{u}^{(i)}\|) - k(\|\mathbf{z}^{(i)} + \mathbf{u}^{(i)}\|)\big)\mathbf{e}_1^{(i+1)} \tag{242}$$

$$= \frac{\widetilde{L}}{2}\big(k(|z - u|) - k(|z + u|)\big)\mathbf{e}_1^{(i+1)} \tag{243}$$

$$= \widetilde{g}_k(z)\mathbf{e}_1^{(i+1)}, \tag{244}$$

where $\widetilde{g}_k(\cdot)$ depending on $u$ is defined in (217). Hence, as illustrated in Figure 7, for any input $\mathbf{x} \in [0, 1]^d$ of $\overline{g}$, we have

$$\mathbf{x}^{(2,1)} = \widetilde{h}(\mathbf{x}), \tag{245}$$

$$\mathbf{x}^{(2,j)} = 0 \qquad\qquad \text{for } j \geq 2, \tag{246}$$

$$\mathbf{x}^{(i+1,1)} = \widetilde{g}_k(\mathbf{x}^{(i,1)}) \qquad \text{for } i \geq 2, \tag{247}$$

$$\mathbf{x}^{(i+1,j)} = 0 \qquad\qquad \text{for } i \geq 2 \text{ and } j \geq 2. \tag{248}$$

By a similar argument to the case of single-output chains, there exists $L \geq L'$ and $\alpha = \Theta(1)$ such that for all $z \in [0, \widetilde{u}]$,

$$1 < \alpha L \leq \frac{\widetilde{g}_k(z)}{z} \leq L. \tag{249}$$

For $2\epsilon \leq \widetilde{g}_k(\widetilde{u})$, we choose $\epsilon_1$ to satisfy

$$(\overline{f}^{(m)} \circ \overline{f}^{(m-1)} \circ \cdots \circ \overline{f}^{(2)})(2\epsilon_1\mathbf{e}_1^{(2)}) = (\widetilde{g}_k \circ \cdots \circ \widetilde{g}_k)(2\epsilon_1) = 2\epsilon, \tag{250}$$

and $\overline{g}$ has a compact support with radius

$$w = \Theta\left(\left(\frac{\epsilon_1}{B}\right)^{1/\nu}\right) = \Theta\left(\left(\frac{\epsilon}{B(cL)^{m-1}}\right)^{1/\nu}\right), \tag{251}$$

for some constant $c = \Theta(1)$.

Lastly, since $L \geq L'$, we immediately deduce that $\overline{f}^{(i)} \in \mathcal{F}(L)$ on its domain for each $i \in [m]$. $\qquad\square$

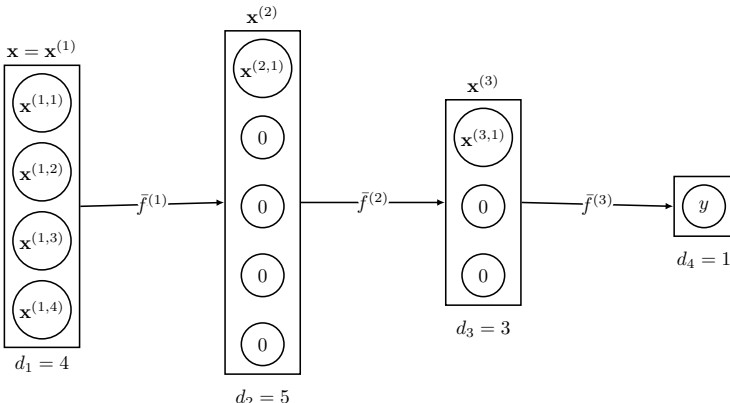

Figure 7: Illustration of $\bar{g}$ for multi-output chain.

### H.3 Hard Function for Feed-Forward Networks

For a fixed $\mathbf{u}^{(i)} \in \mathbb{R}^{d_i}$ and $\widetilde{L}^{(i)} = \sqrt{2}B/\sqrt{k^{(i)}(0) - k^{(i)}(2\|\mathbf{u}^{(i)}\|_2)}$, where $k^{(i)}(\|\mathbf{x} - \mathbf{x}'\|_2) = k^{(i)}(\mathbf{x}, \mathbf{x}')$, we define

$$\widetilde{g}_k^{(i)}(\cdot) = \frac{\widetilde{L}^{(i)}}{2}\big(k^{(i)}(\cdot, \mathbf{u}^{(i)}) - k^{(i)}(\cdot, -\mathbf{u}^{(i)})\big). \tag{252}$$

**Theorem 11** (Hard function for feed-forward networks). *Let $\mathbf{e}_1^{(i)}$ denote the first column of identity matrix of size $d_i$. For the Matérn kernel $k$ with smoothness $\nu \geq 1$, sufficiently small $\epsilon > 0$, and sufficiently large $B$, there exists $\epsilon_1 > 0, L = \Theta(B), c = \Theta(1), w = \Theta\big(\big(\frac{\epsilon}{B(cL)^{m-1}}\big)^{1/\nu}\big)$, and $u > 0$ such that $\bar{g} := \overline{f}^{(m)} \circ \overline{f}^{(m-1)} \circ \cdots \circ \overline{f}^{(1)}$ with $\overline{f}^{(i)}(\cdot) = [\overline{f}^{(i,j)}(\cdot)]_{j=1}^{d_{i+1}}$ for $i \in [m]$, $\overline{f}^{(1,1)}(\cdot) = \widetilde{h}(\cdot, \epsilon_1, w)$, $\overline{f}^{(s,1)}(\cdot) = \widetilde{g}_k^{(s)}(\cdot)$ with $\mathbf{u}^{(s)} = u\mathbf{e}_1^{(s)}$, and $\overline{f}^{(s,r)}(\cdot) = 0$ for $s \in [2, m]$ and $r \neq 1$ has the following properties:*

- *$\overline{f}^{(i,j)} \in \mathcal{H}_k(B) \cap \mathcal{F}(L)$ for each $i \in [m], j \in [d_{i+1}]$;*

- *$\max_{\mathbf{x}} \bar{g}(\mathbf{x}) = 2\epsilon$;*

- *$\bar{g}(\mathbf{x}) > 0$ when $\|\mathbf{x}\|_2 < w$ , and $\bar{g}(\mathbf{x}) = 0$ otherwise.*

*Proof.* We adopt a similar general approach to the case of chains and multi-output chains, but with some different details.

As noted in our analysis of chains, we have $\widetilde{h} \in \mathcal{H}_k(B)$ and $\widetilde{h} \in \mathcal{F}(L')$ for some constant $L' > 1$, and we also have

$$\|\widetilde{g}_k^{(i)}\|_k = \frac{\widetilde{L}^{(i)}}{2}\sqrt{k^{(i)}(\mathbf{u}^{(i)}, \mathbf{u}^{(i)}) - 2k^{(i)}(\mathbf{u}^{(i)}, -\mathbf{u}^{(i)}) + k^{(i)}(-\mathbf{u}^{(i)}, -\mathbf{u}^{(i)})} \tag{253}$$

$$= \frac{\widetilde{L}^{(i)}}{2}\sqrt{2k^{(i)}(0) - 2k^{(i)}(2\|\mathbf{u}^{(i)}\|_2)} \tag{254}$$

$$= B. \tag{255}$$

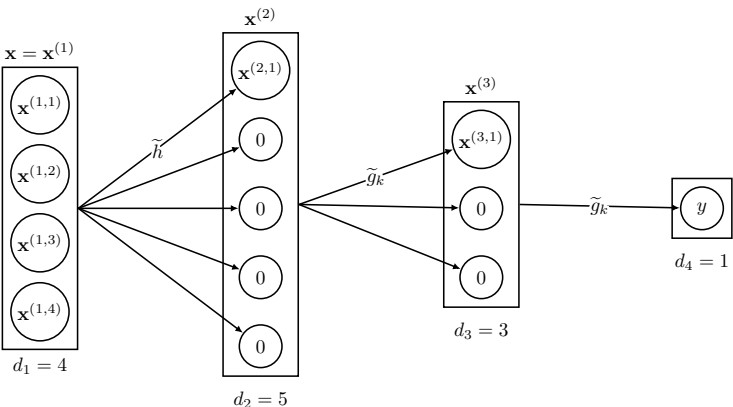

Figure 8: Illustration of $\bar{g}$ for feed-forward network.

Reusing the choice of $u$ and $\widetilde{u}$ in the case of chains, with $\mathbf{z}^{(i)} = z\mathbf{e}_1^{(i)}$ and $\mathbf{u}^{(i)} = u\mathbf{e}_1^{(i)}$, we have

$$\widetilde{g}_k^{(i)}(\mathbf{z}^{(i)}) = \frac{\widetilde{L}^{(i)}}{2}\left(k^{(i)}(\mathbf{z}^{(i)}, \mathbf{u}^{(i)}) - k^{(i)}(\mathbf{z}^{(i)}, -\mathbf{u}^{(i)})\right) \tag{256}$$

$$= \frac{\widetilde{L}^{(i)}}{2}\left(k^{(i)}(\|\mathbf{z}^{(i)} - \mathbf{u}^{(i)}\|_2) - k^{(i)}(\|\mathbf{z}^{(i)} + \mathbf{u}^{(i)}\|_2)\right) \tag{257}$$

$$= \frac{\widetilde{L}^{(i)}}{2}\left(k^{(i)}(|z - u|) - k^{(i)}(|z + u|)\right) \tag{258}$$

$$= \widetilde{g}_k(z) \tag{259}$$

Hence, as illustrated in Figure 8, for any input $\mathbf{x} \in [0,1]^d$ of $\bar{g}$, we have

$$\mathbf{x}^{(2,1)} = \widetilde{h}(\mathbf{x}), \tag{260}$$

$$\mathbf{x}^{(2,j)} = 0 \qquad \text{for } j \geq 2, \tag{261}$$

$$\mathbf{x}^{(i+1,1)} = \widetilde{g}_k(\mathbf{x}^{(i,1)}) \qquad \text{for } i \geq 2, \tag{262}$$

$$\mathbf{x}^{(i+1,j)} = 0 \qquad \text{for } i \geq 2 \text{ and } j \geq 2. \tag{263}$$

By a similar argument to the previous cases, there exists $L \geq L'$ and $\alpha = \Theta(1)$ such that for all $z \in [0, \widetilde{u}]$,

$$1 < \alpha L \leq \frac{\widetilde{g}_k(z)}{z} \leq L. \tag{264}$$

For $2\epsilon \leq \widetilde{g}_k(\widetilde{u})$, we choose $\epsilon_1$ satisfying

$$(\bar{f}^{(m)} \circ \bar{f}^{(m-1)} \circ \cdots \circ \bar{f}^{(2)})(2\epsilon_1\mathbf{e}_1^{(2)}) = (\widetilde{g}_k \circ \cdots \circ \widetilde{g}_k)(2\epsilon_1) = 2\epsilon, \tag{265}$$

and $\bar{g}$ has a compact support with radius

$$w = \Theta\left(\left(\frac{\epsilon_1}{B}\right)^{1/\nu}\right) = \Theta\left(\left(\frac{\epsilon}{B(cL)^{m-1}}\right)^{1/\nu}\right), \tag{266}$$

for some constant $c = \Theta(1)$.

Lastly, due to $L \geq L'$, $\bar{f}^{(i,j)} \in \mathcal{F}(L)$ on its domain for each $i \in [m], j \in [d_{i+1}]$. $\qquad\square$

### H.4 Lower Bound on Simple Regret

With the preceding "hard functions" established, the final step is to essentially follow that of (Bull, 2011). We provide the details for completeness.

By splitting the domain $\mathcal{X} = [0,1]^d$ into a grid of dimension $d$ with spacing $2w$, we construct $M = (\lfloor \frac{1}{2w} \rfloor)^d$ functions with disjoint supports by shifting the origin of $\bar{g}$ to the center of each cell and cropping the shifted function into $[0,1]^d$, which is denoted by $\mathcal{G} = \{g_1, \ldots, g_M\}$. For $g$ sampled uniformly from $\mathcal{G}$, we first show that the expected simple regret $\mathbb{E}[r_T^*]$ of an arbitrary algorithm is lower bounded, then there must exist a function in $\mathcal{G}$ that has the same lower bound.

Now, we prove Theorem 5, which is restated as follows.

**Theorem 5** (Lower bound on simple regret). *Fix $\epsilon \in (0, \frac{1}{2}]$, sufficiently large $B > 0$, $k = k_{Matérn}$, and $\Gamma = \Gamma_{Matérn}$ with smoothness $\nu \geq 1$. Suppose that there exists an algorithm (possibly randomized) that achieves average simple regret $\mathbb{E}[r_T^*] \leq \epsilon$ after $T$ rounds for any $m$-layer chain, multi-output chain, or feed-forward network on $[0,1]^d$ with some $L = \Theta(B)$. Then, provided that $\frac{\epsilon}{B}$ is sufficiently small, it is necessary that*

$$T = \Omega\left( \left( \frac{B(cL)^{m-1}}{\epsilon} \right)^{d/\nu} \right)$$

*for some $c = \Theta(1)$.*

*Proof.* Since the theorem concerns worst-case RKHS functions, it suffices to establish the same result when the function $g$ is drawn uniformly from the hard subset $\mathcal{G}$ introduced above. Given that $g$ is random, Yao's minimax principle implies that it suffices to consider deterministic algorithms.

For any given deterministic algorithm, let $\mathcal{X}_T' = \{\mathbf{x}_1', \mathbf{x}_2', \cdots, \mathbf{x}_T'\}$ be the set of points that it would sample if the function were zero everywhere. We observe that if $g = g_i$ for some $g_i$ satisfying $g(\mathbf{x}_1') = g(\mathbf{x}_2') = \ldots = g(\mathbf{x}_T') = 0$ (i.e., the bump in $g_i$ does not cover any of the points in $\mathcal{X}_T'$), then the algorithm will precisely sample $\mathbf{x}_1', \mathbf{x}_2', \cdots, \mathbf{x}_T'$. Moreover, if there are multiple such functions $g_i$, then the final returned point $\mathbf{x}_T^*$ can only be below $2\epsilon$ (i.e., the bump height) for at most one of those functions, since their supports are disjoint by construction.

Now suppose that $T \leq \frac{M}{2} - 1$, where $M$ is the number of functions in $\mathcal{G}$. This means that regardless of the values $\mathbf{x}_1', \mathbf{x}_2', \cdots, \mathbf{x}_T'$, there are at least $\frac{M}{2} + 1$ functions in $\mathcal{G}$ such that the sampled values are all zero. Hence, there are at least $\frac{M}{2}$ functions where a simple regret of at least $2\epsilon$ is incurred, meaning that the average simple regret is at least $\epsilon$.

With this result in place, Theorem 5 immediately follows by substituting $M = \Theta\left((1/w)^d\right)$ and $w = \Theta\left(\left(\frac{\epsilon}{B(cL)^{m-1}}\right)^{1/\nu}\right)$. $\qquad\square$

### H.5 Behavior of $c^{m-1}$

We readily find from (226) (and the counterparts in the other analyses) that the constant $c$ lies in the range $[\alpha, 1]$. The closer $c$ is to 1, the higher our lower bound is. To understand how close $c$ can be to 1, we recall from Section H.1 that $\alpha = \frac{r_{\min}\widetilde{L}}{L}$ with $L = \max\{L', r_{\max}\widetilde{L}\}$, $L' = \Theta\left(\frac{\epsilon_1}{w}\right)$, and $\widetilde{L} = \frac{\sqrt{2}B}{\sqrt{k(0)-k(2u)}}$ (as well as $r_{\max} = \sup_{z \in (0,\widetilde{u}]} \frac{k(u-z)-k(u+z)}{2z}$ and $r_{\min} = \inf_{z \in (0,\widetilde{u}]} \frac{k(u-z)-k(u+z)}{2z}$).

Observe that when $\frac{\epsilon_1}{B} \ll 1$ and $\nu > 1$, we have $L' = \Theta\left(\frac{\epsilon_1}{w}\right) = \Theta\left(\frac{\epsilon_1}{(\epsilon_1/B)^{1/\nu}}\right) \ll B$. Since $\widetilde{L} = \Theta(B)$, we conclude that $L = r_{\max}\widetilde{L}$, and hence

$$\alpha = \frac{r_{\min}}{r_{\max}} = \frac{\inf_{z \in (0,\widetilde{u}]} \frac{k(u-z)-k(u+z)}{2z}}{\sup_{z \in (0,\widetilde{u}]} \frac{k(u-z)-k(u+z)}{2z}}. \tag{267}$$

In our analysis, we need to choose $u$ and $\widetilde{u}$ such that $\widetilde{g}_k(\widetilde{u}) \geq 2\epsilon$. Thus, as $\epsilon$ decreases towards zero, $u$ and $\widetilde{u}$ can also be chosen arbitrarily small, which in turn implies that $r_{\min}$ and $r_{\max}$ are arbitrarily close to each other as long as the kernel function has a finite slope near zero.

In fact, in Figure 6 we observe that $u$ and $\widetilde{u}$ do not even need to be particularly small to have $\frac{r_{\min}}{r_{\max}}$ close to 1; with $(u, \widetilde{u}) = (0.5, 0.3)$ we get this ratio being 0.958, 0.939, and 0.934 for $\nu = 1, 1.5$, and 2 respectively.

# I    Lower Bound on Cumulative Regret (Proof of Theorem 6)

In this section, we prove Theorem 6, which is restated as follows.

**Theorem 6** (Lower bound on cumulative regret)**.** *Fix sufficiently large $B > 0$, $k = k_{Matérn}$, and $\Gamma = \Gamma_{Matérn}$ with smoothness $\nu \geq 1$. Suppose that there exists an algorithm (possibly randomized) that achieves average cumulative regret $\mathbb{E}[R_T]$ after $T$ rounds for any $m$-layer chain, multi-output chain, or feed-forward network on $[0, 1]^d$ with some $L = \Theta(B)$. Then, it is necessary that*

$$\mathbb{E}[R_T] = \begin{cases} \Omega\big( \min\{T, B(cL)^{m-1}T^{1-\nu/d}\}\big) & when\ d > \nu, \\ \Omega\big( \min\{T, \big(B(cL)^{m-1}\big)^{d/\nu}\}\big) & when\ d \leq \nu, \end{cases}$$

*for some $c = \Theta(1)$.*

*Proof.* By rearranging Theorem 5, we have $\epsilon = \Omega(B(cL)^{m-1}T^{-\nu/d})$, which implies that the lower bound on cumulative regret is

$$\mathbb{E}[R_T] = \Omega(\epsilon T) = \Omega(B(cL)^{m-1}T^{1-\nu/d}). \tag{268}$$

However, this lower bound is loose when $d < \nu$. Theorem 5 implies that to have simple regret at most $\epsilon = \Theta(1)$ requires $T = \Omega\big((B(cL)^{m-1})^{d/\nu}\big)$. When $T = \Theta\big((B(cL)^{m-1})^{d/\nu}\big)$, we have $\mathbb{E}[r_T^*] = \Omega(1)$ and $\mathbb{E}[R_T] = \Omega\big((B(cL)^{m-1})^{d/\nu}\big)$. Since cumulative regret is always non-decreasing in $T$, when $T = \Omega\big((B(cL)^{m-1})^{d/\nu}\big)$, we also have $\mathbb{E}[R_T] = \Omega\big((B(cL)^{m-1})^{d/\nu}\big)$.

For $T = o\big((B(cL)^{m-1})^{d/\nu}\big)$ (which is equivalent to $T = o(B(cL)^{m-1}T^{1-\nu/d})$), we show by contradiction that $\mathbb{E}[R_T] = \Omega(T)$. Suppose on the contrary that there exists an algorithm guaranteeing $\mathbb{E}[R_T] = o(T)$ when $T = T_0$ for some $T_0 = o\big((B(cL)^{m-1})^{d/\nu}\big)$. Then, by repeatedly selecting the best point among the first $T_0$ time steps, the algorithm attains $\mathbb{E}[R_T] = o\big((B(cL)^{m-1})^{d/\nu}\big)$ when $T = \Theta\big((B(cL)^{m-1})^{d/\nu}\big)$, which contradicts the lower bound for $T = \Omega\big((B(cL)^{m-1})^{d/\nu}\big)$.

Hence, Theorem 6 follows by combining the two cases.

$\square$

# J    Summary of Regret Bounds

A detailed summary of our regret bounds for the Matérn kernel is given in Table 2.

# K    Comparison to Related Works

## K.1    Comparison to (Kusakawa et al., 2022)

In this section, we compare our theoretical result to two confidence bound based algorithms cascade UCB (cUCB) and optimistic improvement (OI) proposed by (Kusakawa et al., 2022). Both algorithms utilize a novel posterior standard deviation defined using the Lipschitz constant.

**Problem Setup.**

---

[6]For bounds not requiring this conjecture, see Section 3.4.

| Algorithm-Independent Cumulative Regret Lower Bound | |
|---|---|
| Chains/Multi-Output Chains/Feed-Forward Networks | $\begin{cases} \Omega\big(\min\{T, B(cL)^{m-1}T^{1-\nu/d}\}\big) & \text{when } d > \nu \ge 1, \\ \Omega\big(\min\{T, (B(cL)^{m-1})^{d/\nu}\}\big) & \text{when } d \le \nu. \end{cases}$ |
| **Algorithmic Cumulative Regret Upper Bound** **(If Conjecture of (Vakili, 2022) Holds)**[6] | |
| Chains | $\begin{cases} O(2^m BL^{m-1}T^{1-\nu/d}) & \text{when } d > \nu, \\ \widetilde{O}(2^m BL^{m-1}) & \text{when } d \le \nu. \end{cases}$ |
| Multi-Output Chains | $\begin{cases} O(5^m BL^{m-1}T^{1-\nu/d_{\max}}) & \text{when } d_{\max} > \nu, \\ \widetilde{O}(5^m BL^{m-1}) & \text{when } d_{\max} \le \nu. \end{cases}$ |
| Feed-Forward Networks | $\begin{cases} O(2^m \sqrt{D_{2,m}} BL^{m-1}T^{1-\nu/d_{\max}}) & \text{when } d_{\max} > \nu, \\ \widetilde{O}(2^m \sqrt{D_{2,m}} BL^{m-1}) & \text{when } d_{\max} \le \nu. \end{cases}$ |
| **Algorithm-Independent Simple Regret Lower Bound** | |
| Chains/Multi-Output Chains/Feed-Forward Networks | $\Omega(B(cL)^{m-1}T^{-\nu/d})$ \qquad when $\nu \ge 1.$ |
| **Algorithmic Simple Regret Upper Bound** | |
| Chains/Multi-Output Chains | $\begin{cases} \widetilde{O}(BL^{m-1}T^{-\nu^m/d}) & \text{when } \nu \le 1, \\ \widetilde{O}(B^{1+\nu+\nu^2+\nu^{m-2}}L^{\nu^{m-1}}T^{-\nu/d}) & \text{when } \nu > 1. \end{cases}$ |
| Chains/Multi-Output Chains (Restrictive Cases) | $\begin{cases} O(BL^{m-1}T^{-\nu/d}) & \text{when } \nu \le 1, \\ O(BL^{(m-1)\nu}T^{-\nu/d}) & \text{when } \nu > 1. \end{cases}$ |
| Feed-Forward Networks | $\begin{cases} \widetilde{O}(\sqrt{D_{2,m}} BL^{m-1}T^{-\nu^m/d}) & \text{when } \nu \le 1, \\ \widetilde{O}(\widetilde{D}_{2,m}^{\nu} B^{1+\nu+\nu^2+\nu^{m-2}}L^{\nu^{m-1}}T^{-\nu/d}) & \text{when } \nu > 1. \end{cases}$ |
| Feed-Forward Networks (Restrictive Cases) | $\begin{cases} O(\sqrt{D_{2,m}} BL^{m-1}T^{-\nu/d}) & \text{when } \nu \le 1, \\ O\big((D_{2,m})^{\nu/2} BL^{(m-1)\nu}T^{-\nu/d}\big) & \text{when } \nu > 1. \end{cases}$ |

Table 2: Summary of regret bounds for the Matérn kernel. (The algorithmic simple regret upper bounds are valid when the domain of each layer is a hyperrectangle.) $T$ denotes the time horizon; $m$ denotes the number of layers; $B$ denotes the RKHS norm upper bound of each layer; $L$ denotes the Lipschitz constant upper bound of each layer; $\nu$ denotes the smoothness of the Matérn kernel; $d$ denotes the domain dimension of $g$; $d_{\max} = \max_{i\in[m]} d_i$ denotes the maximum dimension among all the $m$ layers; $D_{2,m} = \prod_{i=2}^{m} d_i$ denotes the product of the dimensions from the second to the last layer; $\widetilde{D}_{2,m}^{\nu} = \prod_{i=2}^{m} (d_i)^{\nu^{m+1-i}/2}$. The lower bounds hold for some $c = \Theta(1)$.

- In each layer, (Kusakawa et al., 2022) assumes the entries in the same layer are mutually independent, which is equivalent to our feed-forward network structure.

- Different from our assumption of $f^{(i,j)} \in \mathcal{H}_k(B) \cap \mathcal{F}(L, \|\cdot\|_2)$ for each $i \in [m]$ and $j \in [d_{i+1}]$, where the Lipschitz continuity is with respect to $\|\cdot\|_2$, (Kusakawa et al., 2022) assumes $f^{(i,j)} \in \mathcal{H}_k(B) \cap \mathcal{F}(L_f, \|\cdot\|_1)$, where the Lipschitz continuity is with respect to $\|\cdot\|_1$. Therefore, for a fixed scalar-valued function $f^{(i,j)} \in \mathcal{F}(L, \|\cdot\|_2) \cap \mathcal{F}(L_f, \|\cdot\|_1)$ with the smallest possible $L$ and $L_f$, it is satisfied that $L_f \le L \le \sqrt{d_i} L_f$.

- (Kusakawa et al., 2022) has an additional assumption that $\sigma_t^{(i,j)} \in \mathcal{F}(L_\sigma, \|\cdot\|_1)$ for all $t \ge 1$ for some constant $L_\sigma$. The constant $L_\sigma$ is not used in the algorithms, and only appears in the regret bounds.

**Regret Bounds.**

- With $D_{2,m} = \prod_{i=2}^{m} d_i$ and $D_{2,m}^+ = \sum_{i=2}^{m} d_i$, noise-free cUCB achieves cumulative regret

$$R_T = O\big(B(BL_\sigma + L_f)^{m-1} D_{2,m} D_{2,m}^+ \sqrt{T\gamma_T}\big),$$

with a $T$-independent factor no smaller than that in $R_T = O(2^m B L^{m-1}\sqrt{D_{2,m} T \gamma_T})$ of our GPN-UCB (Algorithm 1) since $BL^{m-1}\sqrt{D_{2,m}} \leq BL_f^{m-1} D_{2,m} \leq B(BL_\sigma + L_f)^{m-1} D_{2,m}$.

- Noise-free OI achieves simple regret

$$r_T^* = O\big(m^{m^2+m+\frac{1}{2}} B^{m+1} L_f^{m^3} (BL_\sigma + L_f)^{3m^3} (D_{2,m})^{2m^2} (D_{2,m}^+)^{m^2+1} T^{-\frac{\nu}{2\nu+d}}\big)$$

for the Matérn kernel with smoothness $\nu$. When $\nu \leq 1$, our non-adaptive sampling (Algorithm 2) achieves $r_T^* = \widetilde{O}(\sqrt{D_{2,m}} BL^{m-1} T^{-\nu^m/d})$, which has a significantly smaller $T$-independent factor. When $\nu > 1$, our Algorithm 2 achieves $r_T^* = \widetilde{O}(\widetilde{D}_{2,m}^\nu B^{1+\nu+\nu^2+\nu^{m-2}} L^{\nu^{m-1}} T^{-\nu/d})$, which has a smaller $T$-dependent factor. The $T$-independent factors are more difficult to compare, though ours certainly become preferable under the more restrictive scenarios discussed leading up to (22).

**Generality.** (Kusakawa et al., 2022) allows additional (multi-dimensional) input independent of previous layers for each layer. GPN-UCB and non-adaptive sampling can also be adapted to accept additional input. Let $\widehat{\mathcal{X}}^{(i)} \subset \mathbb{R}^{\widehat{d}_i}$ denote the domain of the additional input of $f^{(i)}$, and let $\widehat{\mathbf{x}}_{[i:j]} = [\widehat{\mathbf{x}}^{(i)}, \ldots, \widehat{\mathbf{x}}^{(j)}]$ denote the concatenation of the addition inputs from $f^{(i)}$ to $f^{(j)}$.

- For GPN-UCB (Algorithm 1), we simply modify the upper confidence bound to be

$$\mathrm{UCB}_t(\mathbf{x}, \widehat{\mathbf{x}}) = \max_{\mathbf{z} \in \Delta_t^{(m)}(\mathbf{x}, \widehat{\mathbf{x}}_{[2:m-1]})} \overline{\mathrm{UCB}}_t^{(m,1)}([\mathbf{z}, \widehat{\mathbf{x}}^{(m)}]), \tag{269}$$

where

$$\Delta_t^{(1)}(\mathbf{x}) = \{\mathbf{x}\}, \tag{270}$$

$$\Delta_t^{(i+1,j)}(\mathbf{x}, \widehat{\mathbf{x}}_{[2:i]}) = \left[\min_{\mathbf{z} \in \Delta_t^{(i)}(\mathbf{x}, \widehat{\mathbf{x}}_{[2:i-1]})} \overline{\mathrm{LCB}}_t^{(i,j)}([\mathbf{z}, \widehat{\mathbf{x}}^{(i)}]), \max_{\mathbf{z} \in \Delta_t^{(i)}(\mathbf{x}, \widehat{\mathbf{x}}_{[2:i-1]})} \overline{\mathrm{UCB}}_t^{(i,j)}([\mathbf{z}, \widehat{\mathbf{x}}^{(i)}])\right]$$
$$\text{for } i \in [m-1], j \in [d_{i+1}], \tag{271}$$

$$\Delta_t^{(i)}(\mathbf{x}, \widehat{\mathbf{x}}_{[2:i-1]}) = \Delta_t^{(i,1)}(\mathbf{x}, \widehat{\mathbf{x}}_{[2:i-1]}) \times \cdots \times \Delta_t^{(i,d_i)}(\mathbf{x}, \widehat{\mathbf{x}}_{[2:i-1]})$$
$$\text{for } i \in [m]. \tag{272}$$

The cumulative regret upper bound will remain the same, since $\mathrm{diam}\big(\Delta_t^{(i)}(\mathbf{x}, \widehat{\mathbf{x}}_{[2:i-1]})\big)$ remains the same.

- For non-adaptive sampling (Algorithm 2), with $\widehat{d} = d + \sum_{i=2}^m \widehat{d}_i$, we choose $\{\mathbf{x}_s, \widehat{\mathbf{x}}_s^{(2)}, \ldots, \widehat{\mathbf{x}}_s^{(m)}\}_{s=1}^T$ such that

$$\max_{[\mathbf{x}, \widehat{\mathbf{x}}^{(2)}, \ldots, \widehat{\mathbf{x}}^{(m)}] \in \mathcal{X} \times \widehat{\mathcal{X}}^{(2)} \times \cdots \times \widehat{\mathcal{X}}^{(m)}} \min_{s \in [T]} \|[\mathbf{x}, \widehat{\mathbf{x}}^{(2)}, \ldots, \widehat{\mathbf{x}}^{(m)}] - [\mathbf{x}_s, \widehat{\mathbf{x}}_s^{(2)}, \ldots, \widehat{\mathbf{x}}_s^{(m)}]\|_2 = O(T^{-1/\widehat{d}}). \tag{273}$$

Then, modify the composite mean to be

$$\mu_T^g(\mathbf{x}, \widehat{\mathbf{x}}) = \mu_T^{(m,1)}([\mathbf{z}^{(m)}, \widehat{\mathbf{x}}^{(m)}]) \tag{274}$$

with

$$\mathbf{z}^{(1)} = \mathbf{x}, \tag{275}$$

$$\mathbf{z}^{(i+1,j)} = \mu_T^{(i,j)}([\mathbf{z}^{(i)}, \widehat{\mathbf{x}}^{(i)}]) \qquad \text{for } i \in [m-1], j \in [d_{i+1}], \tag{276}$$

$$\mathbf{z}^{(i+1)} = \mu_T^{(i)}([\mathbf{z}^{(i)}, \widehat{\mathbf{x}}^{(i)}]) = [\mathbf{z}^{(i+1,1)}, \ldots, \mathbf{z}^{(i+1,d_i)}] \qquad \text{for } i \in [m-1]. \tag{277}$$

Then, the only change in simple regret upper bound is that $d$ is replaced by $\widehat{d}$.

### K.2 Comparison to (Sussex et al., 2023)

Another related work (Sussex et al., 2023) seeks the best intervention action for a given causal graph structure. Focusing on DAGs, they have a similar setup to (Kusakawa et al., 2022):

- $f^{(i)} \in \mathcal{H}_k(B) \cap \mathcal{F}(L, \|\cdot\|_2)$ for each node $i$, where the Lipschitz continuity is with respect to $\|\cdot\|_2$.

- $\sigma_t^{(i)} \in \mathcal{F}(L_\sigma, \|\cdot\|_2)$ for each node $i$ and all $t \geq 1$ for some constant $L_\sigma$.

When the causal graph structure is a feed-forward network, their expected improvement based method achieves $R_T = O(B^m L_\sigma^m L^m d_{\max}^m D_{2,m}^+ \sqrt{T\gamma_T})$ when specialized to the noise-free setting, thus containing significantly larger $T$-independent terms compared to GPN-UCB similarly to the above discussion. For fairness, we note that (Sussex et al., 2023) is concerned mainly with the noisy setting, which we do not handle in this paper, instead leaving analogous improvements as potential future work.

## L Experiments

In this section, we experimentally evaluate the performance of our proposed algorithms on chains composed of synthetic functions, and compare to three grey-box algorithms (cascaded UCB (cUCB), optimistic improvement (OI), and cascaded expected improvement (cEI) (Kusakawa et al., 2022)) and to two classic black-box algorithms. We note that with our main contributions all being theoretical, these experiments are only intended to suggest the potential plausibility of our algorithms for practical use, rather than being comprehensive or definitive.

### L.1 Synthetic Chains

We generate two chains $g_1 = f_3 \circ f_2 \circ f_1$ and $g_2 = h_3 \circ h_2 \circ h_1$ (see Figure 9) with $d = 2$ and $m = 3$ by sampling $\{f_1, f_2, f_3, h_1, h_2, h_3\}$ from GP prior with zero mean and squared exponential kernel with lengthscale $l = 1$. $g_1$ and $g_2$ share the same domain $\mathcal{X}$, which contains 2500 points obtained by evenly splitting $[-5, 5]^2$ into a $50 \times 50$ grid. We set $B = 2$ and $L = 2$ for both chains.

### L.2 Algorithms

We consider our proposed algorithms GPN-UCB and NonAda, three grey-box algorithms cUCB, OI, and cEI from (Kusakawa et al., 2022), and two black-box algorithms GP-UCB and EI.

**GPN-UCB.** To reduce the computational cost of GPN-UCB, with $\mathcal{D}$ denoting 100 points evenly selected from $[-5, 5]$, we make the following adjustments for each $i \in \{2, \ldots, m\}$ in the implementation:

- When computing $\overline{\mathrm{UCB}}_t^{(i)}(\mathbf{z})$ (resp. $\underline{\mathrm{LCB}}_t^{(i)}(\mathbf{z})$), we only take minimum (resp. maximum) over $\{\mathbf{z}' \in \mathcal{D} : \|\mathbf{z} - \mathbf{z}'\| \leq 1\}$

- When computing $\Delta_t^{(i+1)}(\mathbf{x})$, we always replace $\Delta_t^{(i)}(\mathbf{x})$ with $\Delta_t^{(i)}(\mathbf{x}) \cap \mathcal{D}$. If $\Delta_t^{(i)}(\mathbf{x})$ is too small making the intersection empty, we use the two endpoints of $\Delta_t^{(i)}(\mathbf{x})$ instead.

**NonAda.** Since the non-adaptive sampling strategy depends on the value of $T$, we run NonAda with $T \in \{2^2, 3^2, \ldots, 14^2\}$ in parallel, and compute the simple regret of the returning point for each $T$.

**cUCB and OI.** (Kusakawa et al., 2022) introduces confidence bounds as follows:

$$\mathrm{UCB}_t(\mathbf{x}) = \widetilde{\mu}_t^{(m)}(\mathbf{x}) + B\widetilde{\sigma}_t^{(m)}(\mathbf{x}), \tag{278}$$

$$\mathrm{LCB}_t(\mathbf{x}) = \widetilde{\mu}_t^{(m)}(\mathbf{x}) - B\widetilde{\sigma}_t^{(m)}(\mathbf{x}), \tag{279}$$

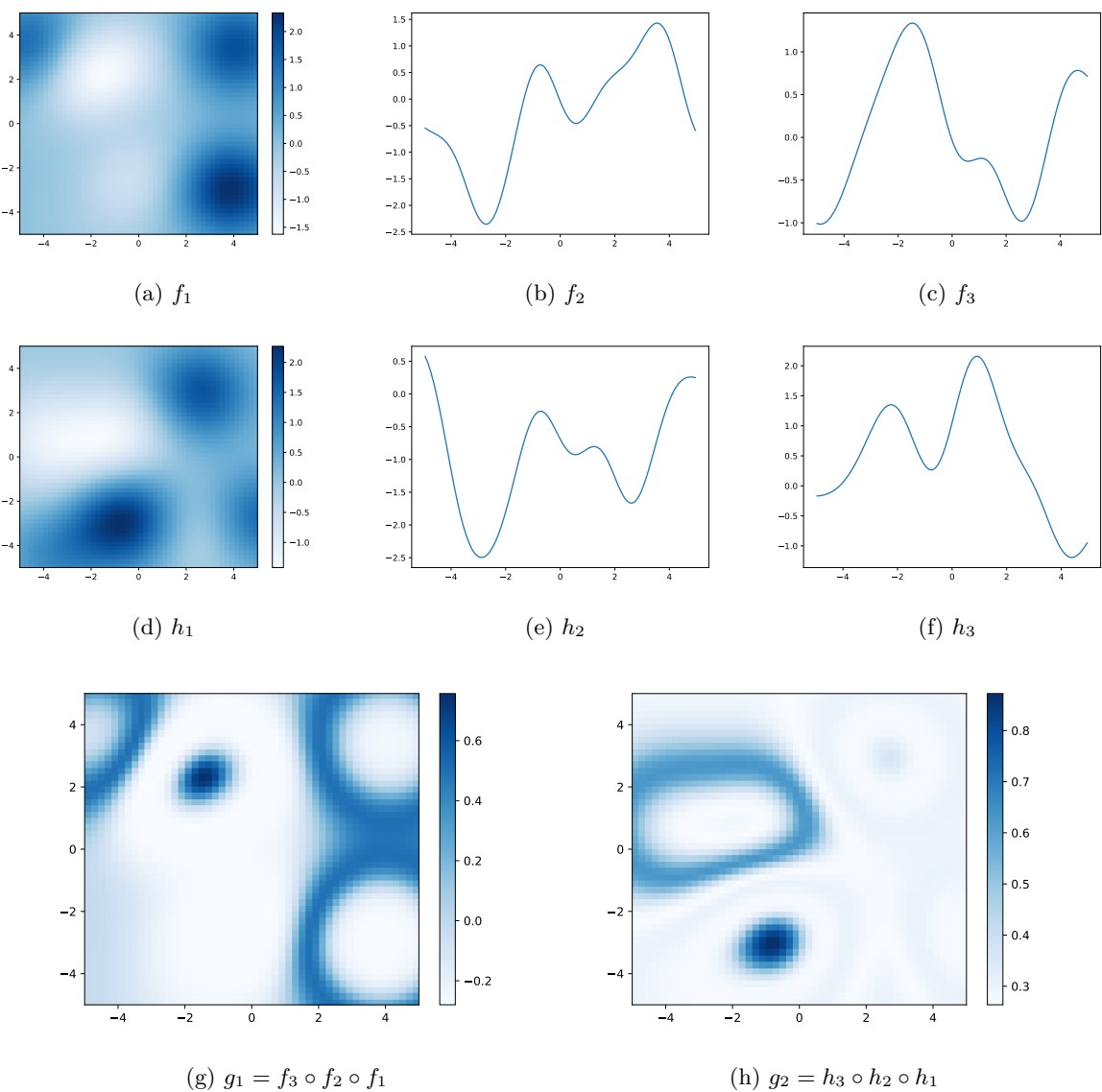

(a) $f_1$

(b) $f_2$

(c) $f_3$

(d) $h_1$

(e) $h_2$

(f) $h_3$

(g) $g_1 = f_3 \circ f_2 \circ f_1$

(h) $g_2 = h_3 \circ h_2 \circ h_1$

Figure 9: Synthetic chains $g_1 = f_3 \circ f_2 \circ f_1$ and $g_2 = h_3 \circ h_2 \circ h_1$

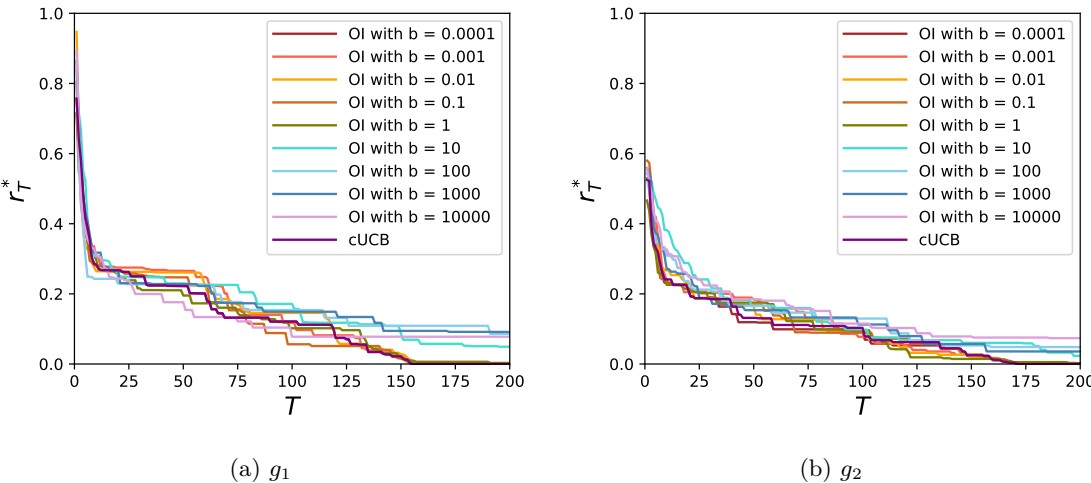

(a) $g_1$             (b) $g_2$

Figure 10: Simple regret of OI with different values of $b$ and cUCB. (The curves of OI with $b = 0.0001$ and cUCB for $g_1$ are indistinguishable.)

where

$$\widetilde{\mu}_t^{(1)}(\mathbf{x}) = \mu_t^{(1)}(\mathbf{x}), \tag{280}$$

$$\widetilde{\mu}_t^{(i)}(\mathbf{x}) = \mu_t^{(i)} \circ \cdots \circ \mu_t^{(1)}(\mathbf{x}) \qquad\qquad \text{for } i = 2, \ldots, m, \tag{281}$$

$$\widetilde{\sigma}_t^{(1)}(\mathbf{x}) = \sigma_t^{(1)}(\mathbf{x}), \tag{282}$$

$$\widetilde{\sigma}_t^{(i)}(\mathbf{x}) = \sigma_t^{(i)}\big(\widetilde{\mu}_t^{(i-1)}(\mathbf{x})\big) + L\widetilde{\sigma}_t^{(i-1)}(\mathbf{x}) \qquad\qquad \text{for } i = 2, \ldots, m. \tag{283}$$

Then, cUCB iteratively selects

$$\mathbf{x}_t^{\mathrm{cUCB}} = \arg\max \mathrm{UCB}_{t-1}(\mathbf{x}). \tag{284}$$

OI iteratively selects

$$\mathbf{x}_t^{\mathrm{OI}} = \arg\max\{\mathrm{UCB}_{t-1}(\mathbf{x}) - \max \mathrm{LCB}_{t-1}, \eta_{t-1}\widetilde{\sigma}_{t-1}(\mathbf{x})\}, \tag{285}$$

where $\eta_t = (1 + \log t)^{-1}$ is an additional parameter.

In our experiments, we introduce a parameter $b$ and set $\eta_t = b \cdot (1 + \log t)^{-1}$. To find the best choice of $b$ for OI, we conduct experiments for OI with $b \in \{10^{-4}, 10^{-3}, \ldots, 10^4\}$, as well as cUCB. We conduct 10 independent trials and plot (see Figure 10) the average simple regret of the best observed points $r_T^* = g(\mathbf{x}^*) - \max_{t \leq T} y_t$. The experimental results show that OI with $b \leq 1$ has similar performance to cUCB, and we will use OI with $b = 1$ as one of the baselines in the following section.

**cEI.** The exact cascaded expected improvement is

$$\mathrm{cEI}_t(\mathbf{x}) = \mathbb{E}_{f^{(1)}, \ldots, f^{(m)}}[\max\{(f^{(m)} \circ \cdots \circ f^{(1)})(\mathbf{x}) - y_{\max}, 0\}], \tag{286}$$

where $f^{(i)}$ follows the posterior distribution based on $t$ observations and $y_{\max}$ is the highest observed value.

Similar to (Kusakawa et al., 2022), we approximate this acquisition function by sampling. Specifically, given $\mathbf{x}$, to generate a sample of $g(\mathbf{x})$, we do the following:

- draw a sample $y_t^1(\mathbf{x})$ from $N\big(\mu_t^{(1)}(\mathbf{x}), \sigma_t^{(1)}(\mathbf{x})^2\big)$;

- recursively draw a sample $y_t^{i+1}(\mathbf{x})$ from $N\big(\mu_t^{(i+1)}(\mathbf{z}), \sigma_t^{(i+1)}(\mathbf{z})^2\big)$, where $\mathbf{z} = y_t^i(\mathbf{x})$.

Repeating this process $S$ times, we obtain $Y_t(\mathbf{x})$, containing $S$ samples of $g(\mathbf{x})$ from the cascaded posterior distribution, and $\text{cEI}_t(\mathbf{x})$ can be approximated by

$$\widehat{\text{cEI}}_t(\mathbf{x}) = \frac{1}{S} \sum_{s \in Y_t(\mathbf{x})} \max\{s - y_{\max}, 0\}. \tag{287}$$

The algorithm selects

$$\mathbf{x}_t^{\text{cEI}} = \arg\max \widehat{\text{cEI}}_{t-1}(\mathbf{x}). \tag{288}$$

In the experiments, we set the sample size as $S = 1000$.

**GP-UCB and EI.** Both algorithms consider $g$ as a black-box function and ignore the intermediate observations. With $\mu_t(\mathbf{x})$ and $\sigma_t(\mathbf{x})$ denoting the posterior mean and standard deviation of the overall function $g(\mathbf{x})$, GP-UCB selects

$$\mathbf{x}_t^{\text{GP-UCB}} = \arg\max \mu_{t-1}(\mathbf{x}) + B\sigma_{t-1}(\mathbf{x}), \tag{289}$$

and EI selects

$$\mathbf{x}_t^{\text{EI}} = \arg\max \mathbb{E}_g[\max\{g(\mathbf{x}) - y_{\max}, 0\}] \tag{290}$$

$$= \big(\mu_{t-1}(\mathbf{x}) - y_{\max}\big)\Phi\Big(\frac{\mu_{t-1}(\mathbf{x}) - y_{\max}}{\sigma_{t-1}(\mathbf{x})}\Big) + \sigma_{t-1}(\mathbf{x})\phi\Big(\frac{\mu_{t-1}(\mathbf{x}) - y_{\max}}{\sigma_{t-1}(\mathbf{x})}\Big), \tag{291}$$

where $\phi$ and $\Phi$ are the pdf and cdf of the standard Gaussian distribution, and $y_{\max}$ is the highest observed value.

### L.3  Experimental Results

We let $T = 200$, and compute posterior mean and variance with $\lambda = 10^{-7}$ to avoid numerical error. The experimental results are displayed in Figure 11 and Figure 12, where the average regret is computed based on 10 independent trials with error bars indicating one standard deviation. Note that the randomness of cEI mainly comes from sampling Gaussian random variables, and the randomness of other algorithms comes from random tie-breaking in the $\arg\max$ operation. In Figure 12, except for NonAda, the reported point for computing simple regret is the best observed point, i.e., $r_T^* = g(\mathbf{x}^*) - \max_{t \leq T} y_t$. The experimental results show that both algorithms are able to locate a near-optimal point in a short time (by $T = 50$).

In terms of cumulative regret, GPN-UCB outperforms all of the baselines. In terms of simple regret, for $g_2$ with an "obvious" maximizer, NonAda outperforms all the baselines, and GPN-UCB has very similar performance to cEI. For $g_1$ with several potential maximizers, our algorithms are slightly outperformed by cEI, but still surpass other baselines. However, it should be noted that the slightly better performance of cEI requires significantly higher computation resources due to the large sample size. Perhaps unexpectedly, we found that the grey-box OI and cUCB algorithms (but not GPN-UCB and cEI) are sometimes outperformed by the black-box GP-UCB and EI algorithms, at least in this simple setting with relatively small $m$. This may be caused by the large uncertainty in (283) when $L > 1$, which makes the confidence bounds loose. This observation, on the other hand, provides some justification for our envelope technique in constructing confidence bounds. Comparing the cumulative and simple regret, we observe that although some trials of cEI and cUCB query a good point at an early time (due to randomness), this does not always help them find the near-optimal point earlier.

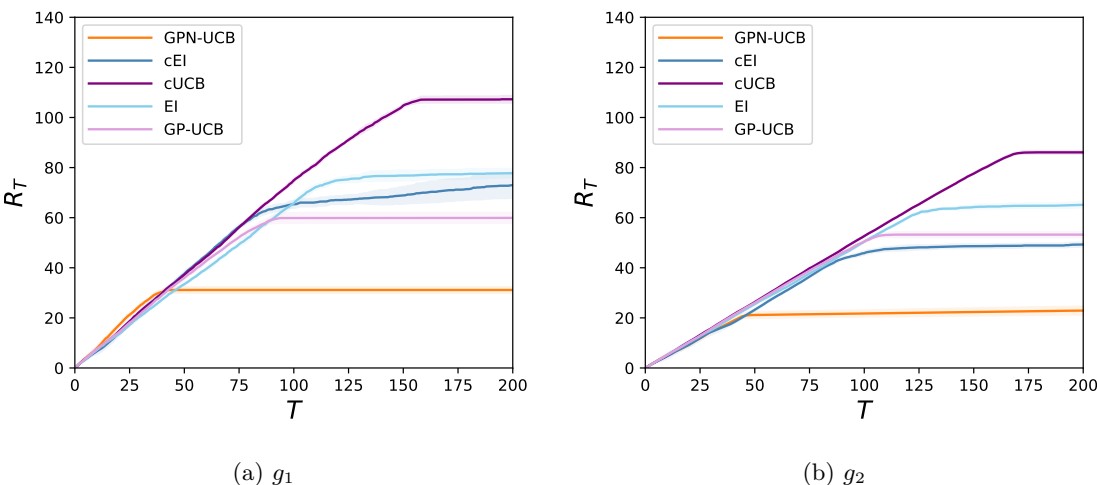

(a) $g_1$       (b) $g_2$

Figure 11: Cumulative regret of GPN-UCB, cEI, cUCB, EI, and GP-UCB .

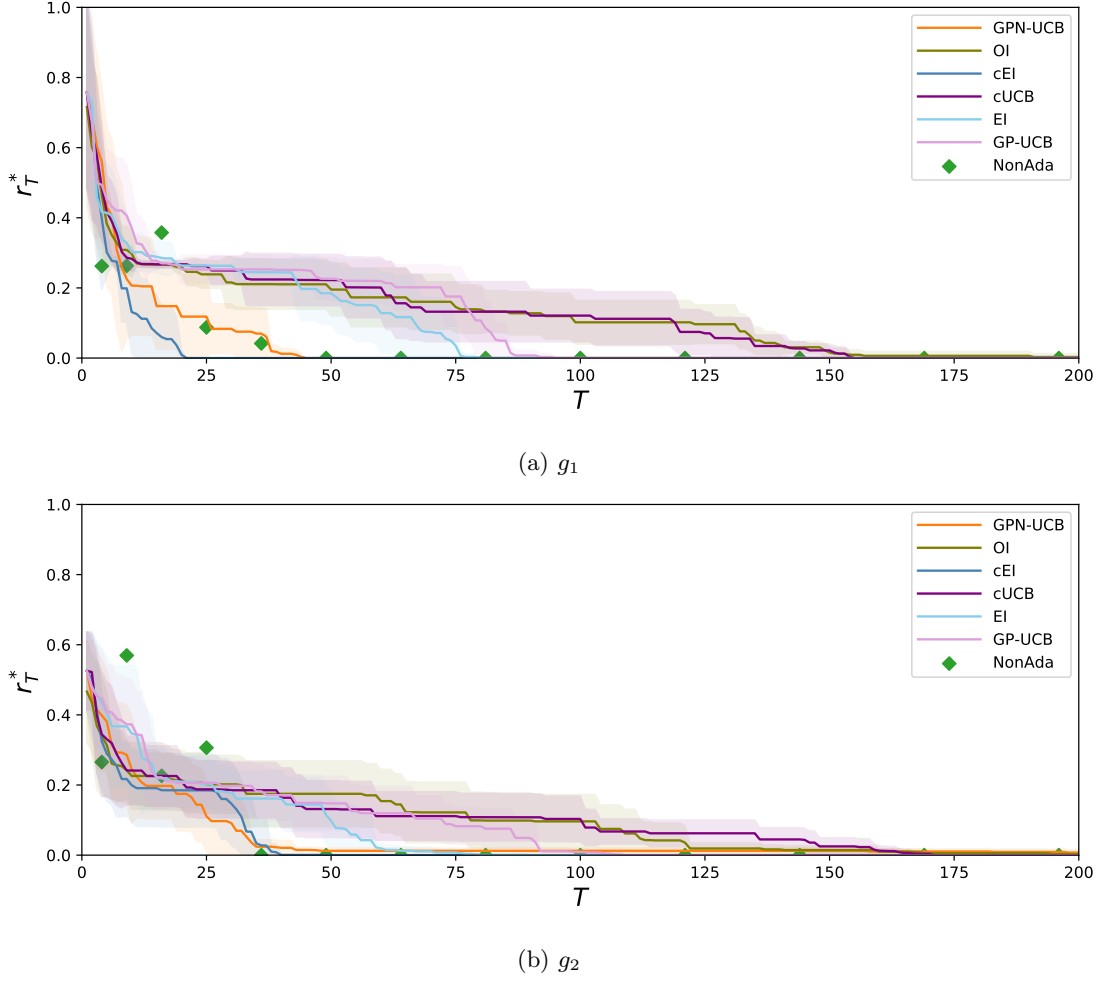

(a) $g_1$

(b) $g_2$

Figure 12: Simple regret of GPN-UCB, NonAda, OI, cEI, cUCB, EI, and GP-UCB.