# OpenReview forum: "Regret Bounds for Noise-Free Cascaded Kernelized Bandits"
_TMLR — Accepted by TMLR_

### Review · Reviewer_Sw3i · 2024-03-25

**Summary Of Contributions:**

The paper studies the problem of optimizing an m-layer function network in the noise free grey-box setting. The authors study different types of networks. They propose a new algorithm that improves the regret bound with respect to the state of the art. Moreover, they complement this result with nearly-matching lower bounds. Finally, the paper shows the empirical performance of the algorithm in simple experimental scenarios.

**Audience:**

Yes

**Claims And Evidence:**

Yes

**Requested Changes:**

None

**Strengths And Weaknesses:**

The paper is well-written and easy to follows. The results are satisfactory and provide improved regret bounds and new lower bounds.

---

### Review · Reviewer_Zmb1 · 2024-04-05

**Summary Of Contributions:**

This paper considers noise-free cascaded kernelized bandit problem in which all the exact intermediate results are observable.
The contribution of this paper includes:
* An upper confidence bound-based algorithm, referred to as GPN-UCB, that achieves an upper bound on the cumulative regret.
* A non-adaptive sampling based method that admits a simple-regret bound when the corresponding RKHS is associated with the Matern kernel.
* An algorithm-independent lower bounds on the simple regret and cumulative regret.

**Audience:**

Yes

**Claims And Evidence:**

No

**Requested Changes:**

I have a concern about the way this paper refers to the conjecture of (Vakili, 2022).
I would appreciate it if you could review the following concerns and either respond or correct the statements.

In this paper,
$\Sigma_T$ is defined as
$$
\Sigma_T = \max_{i \in [m]} \max_{z_1, \ldots, z_T \in \mathcal{X}^{(i)}} \sum_{t=1}^T \sigma_{t-1}^{(i)} (z_t),
$$
i.e.,
this is defined as the **worst-case value** for the layer $i$ and the sample sequence $(z\_t )\_{t=1}^{T}$.
On the other hand,
(Vakili, 2022) seems to conjecture that **there exists an algorithm** that achieves ideal regret bounds.
In particular,
(Vakili, 2022) consider bounds on
$$
\Theta_T^* = \sum_{t=1}^{T} \sigma_{t-1}(x_t^*)
$$
for $( x_t^* )\_{t=1}^{T}$ that is **chosen by UCB-type policies**
(Some notation has been changed for consistency)
as mentioned in Section 4.4. of (Vakili, 2022).
In fact,
(Vakili, 2022) describes
"We however conjecture that the following regret bound is **achievable**"
and
"We conjecture that under certain mild regularity of the domain,
$( x_t^* )\_{t=1}^{T}$ are distributed nearly uniformly across the domain, in which case $\sigma_{i-1}(x_i^*)=O(i^{-\frac{\nu}{d}})$"
which appears to impose some assumptions on $( x_t^* )\_{t=1}^{T}$.

From the above, there seems to be some gap between the "conjecture of (Vakili, 2022)" in the paper submitted to TMLR and the actual conjecture that (Vakili, 2022) had in mind.
Thus, it is questionable whether the original conjecture in (Vakili, 2022) implies the rightmost bounds in Table 1.

**Strengths And Weaknesses:**

Strength:
* This paper provides lower bounds for regret, which would be useful in subsequent studies.
* The models and analyses discussed in the paper are clearly explained through figures and intuitive explanations.

Weakness:
* The regret upper bound of GPN-UCB includes a parameter $\Sigma_T$ defined in Section 1.1, of which (non-trivial) upper bounds are still open.
* It is doubtful that the way of citing the conjecture of (Vakili, 2022) is fair. (Please refer to "Requested Changes" for details.)
* There is a lack of explanation of the motivation and application of considering noise-free (Although the motivation for setting up the cascade is explained).

---

> ### Author Response · Authors · 2024-04-16
> **Response to Reviewer Zmb1**
>
> We thank the reviewer for the helpful feedback.
>
> **Conjecture**
>
> We have added Footnote 4 on Page 9 addressing this, and we summarize as follows.
>
> First, note that Vakili’s paper uses the word “conjecture” twice.  In the first use “...regret bound is achievable…”, we agree that this could be interpreted as being achieved by some algorithm that may or may not be GP-UCB.  However, the second use is much more specific and we believe that it justifies the presentation in our paper.
>
> Specifically, (Vakili, 2022) shows that the analysis of GP-UCB can be reduced to understanding $\max_{x_1,\dots, x_T} \sum \sigma_{t-1}(x_t)$.  By this point, the maximum is over all possible $x_1,…,x_T$ in the domain, not necessarily those of GP-UCB or any other algorithm.  Vakili’s conjecture is that **this maximum is achieved by points roughly uniformly distributed throughout the domain**.  Denoting this maximum value by $h^{(i)}_T$ for layer $i$, our $\Sigma_T$ is exactly $\max_i h^{(i)}_T$. Hence, the conjecture is directly applicable to our setting.
>
> **Noise-Free Setting**
>
> As mentioned in the response to the other reviewer, we view the noise-free setting as an important starting point towards the noisy setting (and already very challenging), and note that early works on noise-free standard Bayesian optimization were highly influential in this sense [1], [2].  In addition, in some applications, the observations could be noise-free, e.g., hyper-parameter tuning, simulation [3], goal-driven dynamics learning [4], and density map alignment [5]. We now mention this in the paper’s first paragraph.
>
> [1] Bull, A. D. (2011). Convergence rates of efficient global optimization algorithms. Journal of Machine Learning Research, 12(10).
>
> [2] Grünewälder, S. et al. (2010). Regret bounds for Gaussian process bandit problems. AISTATS.
>
> [3] Nguyen, T. D., et al. (2016). Cascade Bayesian optimization. In AI 2016: Advances in Artificial Intelligence: 29th Australasian Joint Conference.
>
> [4] Bansal, S., et al. (2017). Goal-driven dynamics learning via Bayesian optimization. In 2017 IEEE 56th Annual Conference on Decision and Control (CDC) (pp. 5168-5173). IEEE.
>
> [5] Singer, A., & Yang, R. (2023). Alignment of density maps in Wasserstein distance. arXiv preprint arXiv:2305.12310.

---

### Review · Reviewer_2SE6 · 2024-04-05

**Summary Of Contributions:**

This work examines noise-free cascaded kernelized bandits across several different network structures, including chains, multi-output chains, and feed-forward networks. For these cascaded structures, the authors propose a novel algorithm by sequentially constructing the confidence region for the output of each layer and ultimately selecting the action with the highest upper confidence bound. They provide both upper and lower bounds for the regret in cascaded kernelized bandits and introduce a Non-Adaptive Sampling Based Method to calculate the confidence region. Additionally, experimental results support the efficiency of the proposed method.

**Audience:**

Yes

**Broader Impact Concerns:**

I do not find any potential negative societal impact.

**Claims And Evidence:**

Yes

**Requested Changes:**

See Weakness.

**Strengths And Weaknesses:**

Strengths:

1. For these cascaded structures, the authors propose a novel algorithm by sequentially constructing the confidence region for the output of each layer and provide a theoretical guarantee for the performance.

2. The author also provides a lower bound, suggesting that the upper bound is near-optimal.

3. The paper is well-written and easy to follow.

4. Experimental results also support the efficiency of the algorithm.

Weakness:

1. For bandit problems, most existing research focuses on scenarios with noise feedback (e.g., Sub-Gaussian noise), which are significantly more challenging than the noise-free case discussed in this work. In a noise-free environment, it appears that any agent could identify the optimal action by simply performing each action once. Therefore, it is crucial to provide an explanation of why a noise-free environment is relevant and important in the study of bandit problems.

2. Regarding the lower bound, it is important to note the existence of an exponential dependency on $m$, represented as $O(c^m)$. In such scenarios, even a minor constant factor in $c$ can result in an exponential increase in the final regret. Hence, it is essential to specify the constant $c$ in the lower bound to facilitate a fair comparison with the upper bound.

---

> ### Author Response · Authors · 2024-04-16
> **Response to Reviewer 2SE6**
>
> We thank the reviewer for the helpful feedback.
>
> **Noise-Free Setting**
>
> We agree that allowing noise would be ideal, but studying the noise-free setting is already a very challenging theoretical problem and it is an important stepping stone towards studying the noisy setting. Some of the earliest Bayesian optimization works were also noise-free and ended up being very influential [1], [2].  In the first paragraph of the paper, we have now added some potential applications that can be noise-free.
>
> Please note that at least as $T$ grows large, our bounds amount to significantly better efficiency than playing every action once (since they depend on the function complexity but not the number of domain points), and they even apply to continuous action spaces.  From a more practical viewpoint, we see in Appendix L that our algorithms are able to locate the optimal action within 50 queries, which is much more effective than performing each of the 2500 actions once.
>
>
> [1] Bull, A. D. (2011). Convergence rates of efficient global optimization algorithms. Journal of Machine Learning Research, 12(10).
>
> [2] Grünewälder, S. et al. (2010). Regret bounds for Gaussian process bandit problems. AISTATS.
>
> **$c^m$ in Lower Bounds**
>
> Thanks for raising this important point.  We have updated the paper to discuss this in Remark 3 and Appendix H.5.
>
> A brief summary is as follows: Our proof (Appendix H, equations (222)-(227)) directly reveals that $c < 1$, but ideally we would like $c$ to be close to 1 in order to match the upper bound.  We argue that this is indeed the case:
> - Under the mild condition $\nu > 1$, we show that when $\epsilon$ is small enough, we get $L’$ being insignificant and $c = \frac{r_{\min}}{r_{\max}}$, and in addition, we can take $\widetilde{u}$ arbitrarily small in order to get $c$ arbitrarily close to one (for small enough $\epsilon$).
> - Even if we fix $u$ and $\widetilde{u}$ to moderate values, we find in our examples that $\frac{r_{\min}}{r_{\max}}$ is close to one.  Specifically, in Figure 6, we obtain values of $0.958, 0.939,$ and $0.934$ for $\nu = 1, 1.5, 2$ respectively.

---

### Decision · Action_Editor_nVPJ · 2024-05-06

**Recommendation:** Accept as is

**Comment:**

The reviewers agree the claims in the paper are backed by sufficient evidence and the results are sound. The paper provides lower bounds and algorithm guarantees and the proofs appear correct. Most of the reviewer concerns have been addressed in the author discussion, hence I believe the paper can be accepted as is.

**Audience:**

The paper is likely to be of interest to at least one sub-community of TMLR readers, optimization over functions that are expensive to evaluate being a fairly widespread problem.

**Claims And Evidence:**

The reviewers agree that the claims in the paper are appropriately backed by theoretical guarantees (mathematical proofs).